# NAD+ regulates nucleotide metabolism and genomic DNA replication

Sebastian Howen Nesgaard Munk [1], Joanna Maria Merchut-Maya[1],
Alba Adelantado Rubio[1], Arnaldur Hall[2], George Pappas[2], Giacomo Milletti [1],
MyungHee Lee[1,2], Lea Giørtz Johnsen [3], Per Guldberg[4,5], Jiri Bartek [2,6] &
Apolinar Maya-Mendoza [1]

The intricate orchestration of enzymatic activities involving nicotinamide adenine dinucleotide (NAD+) is essential for maintaining metabolic homeostasis and preserving genomic integrity. As a co-enzyme, NAD+ plays a key role in regulating metabolic pathways, such as glycolysis and Kreb's cycle. ADP-ribosyltransferases (PARPs) and sirtuins rely on NAD+ to mediate post-translational modifications of target proteins. The activation of PARP1 in response to DNA breaks leads to rapid depletion of cellular NAD+ compromising cell viability. Therefore, the levels of NAD+ must be tightly regulated. Here we show that exogenous NAD+, but not its precursors, has a direct effect on mitochondrial activity. Short-term incubation with NAD+ boosts Kreb's cycle and the electron transport chain and enhances pyrimidine biosynthesis. Extended incubation with NAD+ results in depletion of pyrimidines, accumulation of purines, activation of the replication stress response and cell cycle arrest. Moreover, a combination of NAD+ and 5-fluorouridine selectively kills cancer cells that rely on de novo pyrimidine synthesis. We propose an integrated model of how NAD+ regulates nucleotide metabolism, with relevance to healthspan, ageing and cancer therapy.

NAD+, an essential metabolite and co-enzyme, assumes two distinct forms within the cellular milieu: oxidized (NAD+) and reduced (NADH). NAD+ and NADH are necessary for glycolysis, the tricarboxylic acid (TCA) cycle (also known as Kreb's cycle) and the electron transport chain (ETC). NAD+ is also required as a co-factor for enzymatic processes involved in post-translational modifications. Sirtuins are a family of proteins that use NAD+ as a co-factor. SIRT1, the most studied mammalian sirtuin, controls deacetylation of important regulators of basal metabolism[1,2]. ADP-ribosyltransferases (PARPs) also employ NAD+ as co-factor for catalysing the attachment of ADP-ribose to target molecules. In unperturbed S phase, the most abundant member of the PARP family, PARP1, regulates the speed of DNA synthesis[3], while upon DNA damage, PARP1 is recruited to DNA breaks to catalyse ADP-ribosylation

of itself and target proteins. Sustained PARP1 activity consumes NAD+ and leads to depletion of ATP, ultimately compromising cell viability[4,5].

Ageing may induce chronic DNA damage and PARP activation, thereby resulting in NAD+ depletion and mitochondrial dysfunction[6]. In old mice, NAD+ levels can be restored by feeding them with the NAD+ precursor nicotinamide mononucleotide (NMN)[7]. Inhibition of PARP1 leads to increased cellular NAD+ and improves mitochondrial function in skeletal muscle[8]. Thus, the use of PARP1 inhibitors has been proposed to modulate NAD+ and energy levels[9].

Three synthesis pathways maintain cellular NAD+ (Extended Data Fig. 1a): (1) the de novo pathway, which occurs only in the liver and generates NAD+ from tryptophan; (2) the Preiss–Handler pathway, which uses nicotinic acid as a precursor; and (3) the salvage pathway,

[1]DNA Replication and Cancer Group, Danish Cancer Institute, Copenhagen, Denmark. [2]Genome Integrity Group, Danish Cancer Institute, Copenhagen, Denmark. [3]MS-Omics, Vedbæk, Denmark. [4]Molecular Diagnostics Group, Danish Cancer Institute, Copenhagen, Denmark. [5]Department of Cancer and Inflammation Research, Institute for Molecular Medicine, University of Southern Denmark, Odense, Denmark. [6]Division of Genome Biology, Department of Medical Biochemistry and Biophysics, Karolinska Institutet, SciLifeLab, Stockholm, Sweden. ✉e-mail: jb@cancer.dk; apomm@cancer.dk

which synthesizes NAD+ from nicotinamide riboside (NR) or nicotinamide (NAM)[10]. In the salvage pathway, nicotinamide phosphoribosyltransferase (NAMPT) catalyses the rate-limiting reaction, and its inhibition depletes intracellular NAD+(ref. 11). A minor fluctuation of NAD+ levels may have profound effects on cellular physiology, given the reported cytoplasmic concentration of NAD+ at ~100 μM and the mitochondrial concentration at ~230 μM (ref. 12). The prevailing consensus that NAD+ as an intact molecule cannot cross the cellular membrane[13,14] has been challenged by recent observations that NAD+ can enter the cell via non-specific channel proteins, including the connexin 43 hemichannels[15–17]. Moreover, cytoplasmic NAD+ can be taken directly into the mitochondrial matrix[18] by the SLC25A51 transporter[19]. Whether the level of exogenous NAD+ has a direct impact on the DNA damage response (DDR) and genomic DNA synthesis is a matter of controversy. In this Article, we report that a concentration ≥80 μM of exogenous NAD+ initially boosts mitochondrial activity and DNA synthesis, however, long exposure to NAD+ leads to a reduction of pyrimidine biosynthesis and cell cycle arrest. Lastly, we propose an integrated model to reconcile and interpret past and newly emerging evidence for the role of NAD+ in controlling mitochondrial function, DNA replication and cell proliferation.

## Results

### Exogenous NAD+ impairs genomic DNA replication

To alter intracellular levels of NAD+, we inhibited the activity of PARP1, SIRT1 and NAMPT in various human cell models and quantified the levels of genomic DNA synthesis through incorporation of 5-ethynyl-2′-deoxyuridine (EdU) (Fig. 1a and Extended Data Fig. 1b). While PARP inhibition reduced DNA synthesis in all tested models, inhibition of NAMPT caused this effect specifically in HeLa cells. The inhibition of NAMPT resulted in a >90% reduction of intracellular NAD(H), whereas inhibition of PARP1 increased it by ~50% (Fig. 1b). Inhibition of SIRT1 had a statistically non-significant impact on NAD(H) levels. To extend our findings, we used quantitative image-based cytometry (QIBC)[20] to measure the level of genomic DNA synthesis together with DDR activation. For the analysis of DDR, we quantified foci formation of γH2AX. In HeLa cells, incorporation of EdU was reduced after inhibition of NAMPT while U2OS cells were unaffected (Fig. 1c and Extended Data Fig. 1c). PARP1 inhibition resulted in a global reduction of DNA synthesis with foci formation of γH2AX, whereas inhibition of SIRT1 had no discernible impact on either of the parameters. Next, we analysed replication fork speed and fork progression using the DNA fibre technique[3]. Cells were subjected to two consecutive pulses of 5-chloro-2′-deoxyuridine (CldU) and 5-iodo-2′-deoxyuridine (IdU), each lasting 20 min, and the total fork speed was quantified by assessing the length of both pulses, while fork progression was evaluated by calculating the ratio of CldU/IdU. CldU/IdU ratios of <0.5 or >1.5 indicate asymmetric forks, a readout of compromised fork progression. In U2OS cells, the results showed no differences in fork speed or fork progression after the inhibition of NAMPT and only a small reduction in fork speed after the inhibition of SIRT1 (Extended Data Fig. 1d,e). Consistent with our previous observations[3], inhibition of PARP1 accelerated the speed of fork progression and triggered DDR.

Because inhibition of NAMPT resulted in depletion of NAD+ and impaired genomic DNA synthesis in HeLa cells, we sought to rescue these deficiencies with exogenous NAD+(refs. 19,21,22). Unexpectedly, treatment with NAD+ hindered genomic DNA synthesis independently of NAMPT and PARP1 activity (Fig. 1d and Extended Data Fig. 1f). Analysis of replication forks revealed a slight rescue of fork speed by the combined treatment with NAD+ and NAMPT inhibitor in HeLa but not in U2OS cells (Fig. 1e and Extended Data Fig. 1g–i). To investigate the cell-type dependency of replication sensitivity to NAD+, we treated several other cell models and observed that both fork speed and progression were negatively affected, with primary normal cell types exhibiting lower sensitivity to NAD+ (Fig. 1f and Extended Data Fig. 1j).

### Exogenous NAD+ induces DNA replication stress

Next, we treated cells with different concentrations of NAD+. When added for 24 h at 80 μM in HeLa and 400 μM in U2OS cells, NAD+ impaired the incorporation of EdU (Fig. 2a and Extended Data Fig. 1k,l). Moreover, 16 μM of NAD+ was sufficient to reduce fork speed by ~40% in HeLa cells without affecting the total level of DNA synthesis. This apparent discrepancy between the EdU incorporation and the results from DNA fibres may be related to the fact that a reduction of fork speed by ~30–40% can be compensated by the activation of dormant origins to maintain global levels of DNA synthesis[23]. In U2OS cells, 16 μM NAD+ slightly accelerated fork progression, while higher concentrations impaired it (Fig. 2b,c and Extended Data Fig. 1m,n). In time course experiments, we found that 1 h incubation with NAD+ was sufficient to increase the intracellular levels of NAD+ and NADH by ~4-fold (Extended Data Fig. 2a,b) and enhance fork speed without activating DDR, while after 3 h fork speed started to decrease and asymmetric forks accumulate. As a control to reduce the speed and integrity of replication forks, cells were treated with hydroxyurea (HU), an inhibitor of ribonucleotide reductase (RNR) that induces depletion of purines with an increase in thymidine 5′-triphosphate (TTP) and deoxycytidine triphosphate pools[24]. After 24 h of NAD+ treatment, the reduction in fork speed was similar to that in cells treated with HU (Fig. 2d and Extended Data Fig. 2c–e). Immunoblot analysis revealed a progressive activation of the replication stress response, as evidenced by the accumulation of phosphorylated CHK1 and H2AX (Fig. 2e and Extended Data Fig. 2f). Furthermore, 24 h treatment with NAD+ had a cytostatic effect with little or no induction of cell death (Extended Data Fig. 2g,h). Notably, this cytostatic effect was fully reversible upon removal of NAD+ (Extended Data Fig. 2i).

NAD+ as well as its precursors NR and NMN can be used to increase intracellular NAD(H) levels[16,25,26]. We therefore treated cells with NR, NMN or NAD+ and investigated their effects. While NR increased NAD(H) levels (Extended Data Fig. 2j), only NAD+ impaired the speed and progression of replication forks (Fig. 2f and Extended Data Fig. 2k). QIBC analysis showed that, even after 72 h of treatment, neither NR nor NMN affected EdU incorporation (Fig. 2g and Extended Data Fig. 2l), and only a slight increase in phosphorylation of H2AX, but not CHK1, was detected in cells treated with NR for 48 h (Extended Data Fig. 2m). These findings indicate that genomic DNA synthesis exhibits tolerance towards a lower level of intracellular NAD(H). Conversely, exogenous NAD+ elicits a dual effect on genomic DNA synthesis, initially augmenting this process followed by a subsequent cessation.

NAD+ can also be metabolized by the CD38 ectoenzyme to produce cyclic ADP-ribose and activate calcium-regulated signalling pathways[27], which potentially could explain the differential effects observed between treatment with NAD+ and its precursors. The chemical inhibition of CD38 altered neither the effect of NAD+ on genomic DNA synthesis, nor was this associated with altered Ca2+ signalling (Extended Data Fig. 2n,o).

### Exogenous NAD+ depletes pyrimidine nucleotides

NAD(H) is necessary for ATP production during glycolysis and oxidative phosphorylation (OXPHOS), and NAMPT inhibition causes depletion of ATP and altered cellular bioenergetics[28]. To test how NAD+ affects ATP levels and cell proliferation, we treated various cell types with increasing concentrations of the NAMPT inhibitor or NAD+. Inhibition of NAMPT caused a decrease in ATP and cell proliferation, and a small increase in cell death, with HeLa cells being the most sensitive. NAD+ treatment led to elevated ATP levels in a concentration-dependent manner and correlated inversely with cell proliferation, which was most pronounced in HeLa cells. Aligned with the effect on DNA synthesis, cell proliferation was halted in HeLa cells by treatment with exogenous NAD+ from 80 μM (Fig. 3a and Extended Data Fig. 3a).

To investigate the impact of NAD+ and its precursors on mitochondrial NAD(H) levels, we measured the concentration of NAD(H)

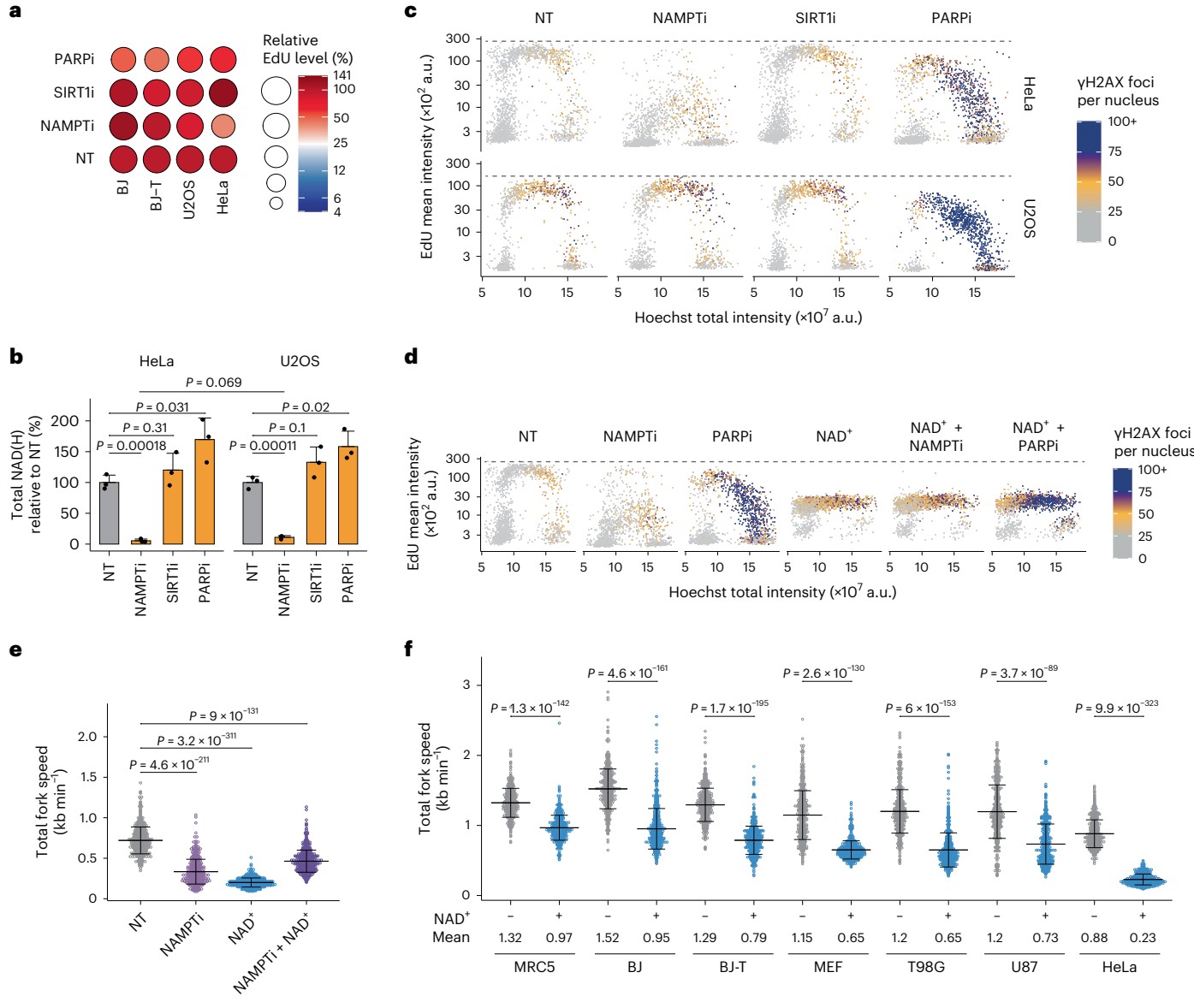

**Fig. 1 | Exogenous NAD⁺ impairs DNA replication. a**, Flow cytometry analysis of EdU incorporation in S-phase cells relative to the non-treated (NT) condition. Cells were treated for 24 h with 10 nM NAMPTi, 2 µM SIRT1i or 10 µM PARPi. The experiment was performed twice with similar results. **b**, NAD(H) quantification in HeLa and U2OS cells treated as in **a**. Data are mean + s.d., $n = 3$ biological replicates, two-tailed Student's $t$-test. **c**, QIBC analysis of γH2AX foci, EdU intensity and DNA content in HeLa and U2OS cells treated as in **a**. Representative experiment from three independent experiments. **d**, QIBC analysis of γH2AX foci, EdU intensity and DNA content in HeLa cells treated for 24 h without (NT) or with 10 nM NAMPTi, 10 µM PARPi and/or 2 mM NAD⁺. Representative experiment from three (NT, PARPi and NAD⁺) and two (all other conditions) independent

experiments. **e**, Replication fork speed in HeLa cells treated for 24 h without (NT) or with 10 nM NAMPTi and/or 2 mM NAD⁺. Data are mean ± s.d., means are indicated. Scored forks: NT, 553; NAMPTi, 515; NAD⁺, 552; NAMPTi+NAD⁺, 557. Two-tailed Welch's $t$-test. **f**, Replication fork speed in different cell lines treated for 24 h without or with 2 mM NAD⁺. Data are mean ± s.d., and mean values in kb min⁻¹ of the fork speed are indicated at the bottom of the plot. Scored forks: MRC5 NT, 525; MRC5 NAD⁺, 535; BJ NT, 543; BJ NAD⁺, 523; BJ-T NT, 530; BJ-T NAD⁺, 542; MEF NT, 529; MEF NAD⁺, 612; T98G NT, 519; T98G NAD⁺, 519; U87 NT, 522; U87 NAD⁺, 539; HeLa NT, 543; HeLa NAD⁺, 582. Two-tailed Welch's $t$-test. Replication forks were scored for two biological replicates. Numerical data are available in source data files.

in mitochondria isolated from cells treated with either NAMPT inhibitor or different concentrations of NAD⁺ and NR. Inhibition of NAMPT depleted mitochondrial NAD(H), while >16 µM of exogenous NAD⁺ led to a two-fold increase in the mitochondrial level of NAD(H) (Fig. 3b and Extended Data Fig. 3b). In a time course experiment, mitochondrial NAD(H) exhibited a nearly two-fold increase after 1 h of treatment with either 80 µM or 2 mM NAD⁺, while no discernible effect was observed in cells treated with NR (Extended Data Fig. 3c). After 24 h of treatment with NAD⁺ or NR, the levels of mitochondrial NAD(H) were elevated;

nonetheless, only NAD⁺ yielded concurrent elevations in ATP levels and compromised cell proliferation (Extended Data Fig. 3d,e).

Considering the potential impact of NAD⁺ on various cellular metabolic pathways, we conducted a comprehensive analysis of the HeLa cell metabolome using ultraperformance liquid chromatography (UPLC)/mass spectrometry (MS). Following treatment with 80 µM of NAD⁺, we observed a distinct metabolic transition characterized by reduced pyrimidine nucleotide levels and concurrent elevation in purine nucleotide levels (Fig. 3c). This transition coincided with a

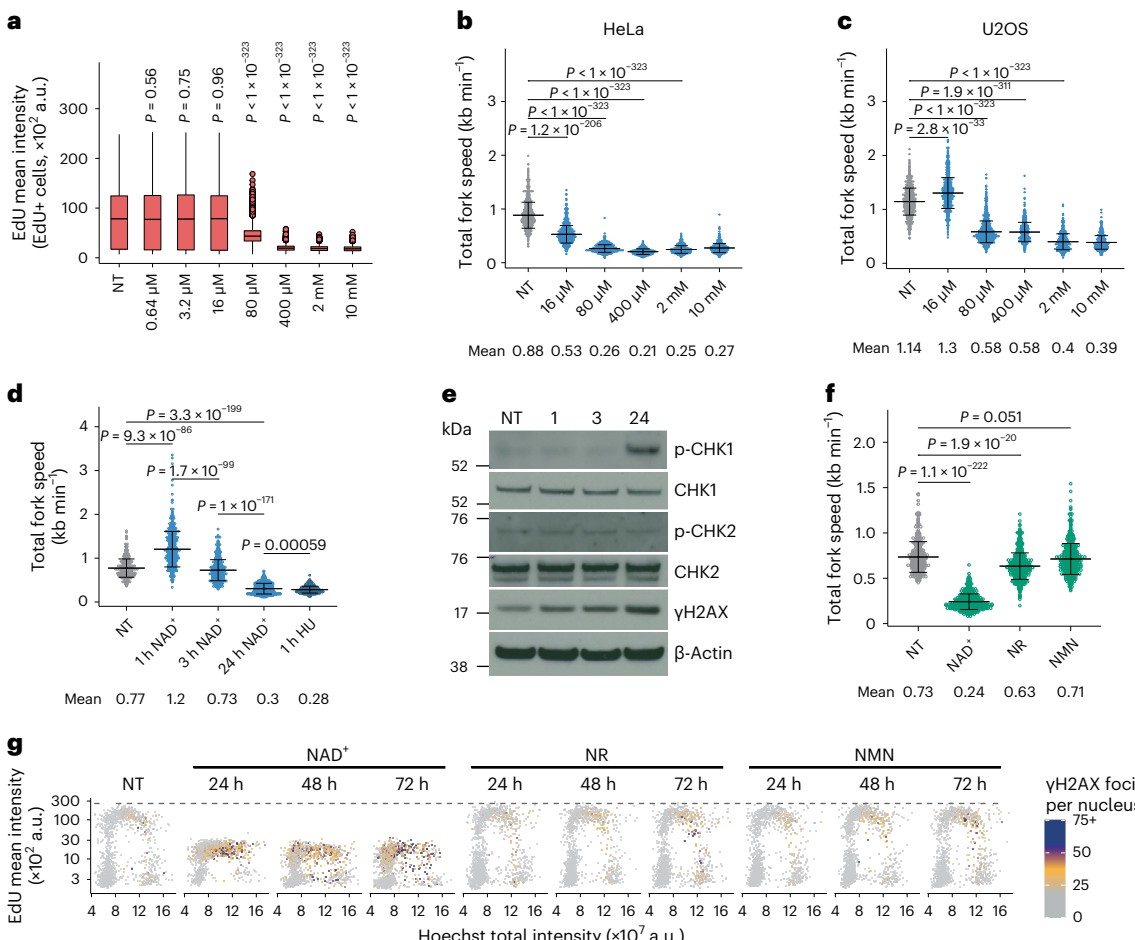

**Fig. 2 | Exogenous NAD⁺ induces replication stress. a**, QIBC analysis of EdU-positive HeLa cells treated for 24 h with NAD⁺. Cells quantified: non-treated (NT), 7,540; 0.64 μM, 7,557; 3.2 μM, 7,472; 16 μM, 6,935; 80 μM, 12,107; 400 μM, 13,322; 2 mM, 13,068; 10 mM, 12,874, from three biological replicates. Two-tailed Welch's *t*-test compared with NT. **b,c**, Replication fork speed from HeLa (**b**) and U2OS (**c**) cells treated for 24 h with NAD⁺. Scored forks in **b**: non-treated (NT), 874; 16 μM, 879; 80 μM, 821; 400 μM, 526; 2 mM, 513; 10 mM, 543. Scored forks in **c**: NT, 1,022; 16 μM, 787; 80 μM, 1,088; 400 μM, 507; 2 mM, 592; 10 mM, 561. **d**, Replication fork speed from HeLa cells treated with 2 mM NAD⁺ or HU. Scored forks: non-treated (NT), 464; 1 h NAD⁺, 565; 3 h NAD⁺, 539; 24 h NAD⁺, 577; 1 h HU, 380. **e**, Immunoblot from HeLa cells treated without (NT) or with 2 mM NAD⁺ for the indicated hours. The experiment was performed twice with similar results. **f**, Replication fork speed from HeLa cells treated for 24 h with NAD⁺, NR or NMN at 2 mM. Scored forks: non-treated (NT), 406; NAD⁺, 560; NR, 518; NMN, 526. **g**, QIBC analysis of γH2AX foci, EdU intensity and DNA content in HeLa cells treated without (NT) or with NAD⁺, NR or NMN at 2 mM. Representative experiment from two (24 h NAD⁺, 24 h NMN, 48 h NMN, 72 h NMN, 72 h NR) and three (all other conditions) independent experiments. Replication fork data are mean ± s.d., means are indicated, forks were scored for two biological replicates, two-tailed Welch's *t*-test. For boxplots in **a**, centre line indicates the median; box limits indicate the 25th and 75th percentiles; minima and maxima of whiskers extend 1.5 times the inter-quartile range from 25th and 75th percentiles, respectively. Numerical data and uncropped immunoblots are available in source data files.

decrease in replication fork speed (Fig. 3d). To get more insight into the effect of exogenous NAD⁺, we analysed the transcriptomes of four cell types following treatment with NAD⁺. At the transcriptional level, HeLa and U2OS cells showed the most changes, with upregulation of genes involved in oxidative metabolism and the ETC (Extended Data Fig. 4a–f). To further characterize these two cell types, we analysed the metabolome of cells treated with NAD⁺ for 1 h and 24 h (Extended Data Fig. 5a). It showed that U2OS cells maintain a larger pool of nucleotides than HeLa cells. Following 1 h of treatment, a modest increase in some purine and pyrimidine nucleotides was observed in both cell lines, coinciding with an increase in replication fork speed. However, after 24 h, a notable depletion of pyrimidine nucleotides alongside an accumulation of purine nucleotides manifested in both cell lines. Particularly in HeLa cells, the levels of AMP and GMP were greatly increased. It is plausible that depletion of pyrimidines and accumulation of purines serve as a mechanism underlying the impediment of genomic DNA synthesis induced by NAD⁺ (Fig. 3e).

The accumulation of AMP has been reported to inhibit the growth of fibroblasts, presumably by inducing pyrimidine starvation[29], and AMP as well as IMP, a key intermediate in purine biosynthesis, can impair the activity of uridine monophosphate synthase (UMPS) in the pyrimidine biosynthesis pathway[30]. The enzymes adenine phosphoribosyltransferase (APRT) and hypoxanthine-guanine phosphoribosyltransferase (HPRT1) are responsible for converting adenine to AMP and hypoxanthine to IMP, respectively. These enzymes were expressed at higher levels in HeLa than in U2OS cells (Fig. 3f and Extended Data Fig. 5b). We also detected higher levels of both AMP and IMP in HeLa than in U2OS cells after NAD⁺ treatment, while hypoxanthine was greatly elevated in U2OS cells (Fig. 3g and Extended Data Fig. 5b). To explore whether HeLa and U2OS cells were differentially sensitive to AMP, we tested the effect of exogenous AMP on DNA synthesis. Following treatment with 100 μM AMP, we observed impaired EdU incorporation only in HeLa cells. However, the effect of 2 mM AMP was evident in both cell lines, with a more pronounced sensitivity in HeLa cells

(Fig. 3h). Together, the metabolome and transcriptome analyses revealed differences between the two cell types in the response to NAD[+], and the lower expression of *HPRT1* and *APRT* in U2OS cells could prevent AMP accumulation following purine degradation and, hence, the impairments induced by AMP accumulation.

HeLa cells seem to be particularly dependent on the de novo pyrimidine synthesis pathway given their higher levels of the dihydroorotate dehydrogenase (DHODH) and UMPS enzymes. Moreover, pointing to different mechanisms of pyrimidine regulation triggered by exogenous NAD[+], U2OS cells accumulated carbamoylaspartate and dihydroorotic acid (Extended Data Fig. 6a). In addition, we observed differences in AMPK regulation between HeLa and U2OS cells, with AMPK being constitutively phosphorylated in U2OS cells, which in turn could influence the regulation of the recently characterized pyrimidinosome[31], favouring increased dihydroorotate synthesis in U2OS cells after NAD[+] treatment (Extended Data Fig. 6b). Consistent with pyrimidine depletion, after 3 h treatment with NAD[+], global transcription started to be affected and structural changes in the nucleolus were seen later[32] (Extended Data Fig. 6c–e). Together, these results showed that exogenous NAD[+] induces depletion of pyrimidine nucleotides and concomitant accumulation of purine nucleotides with functional consequences for genomic DNA replication.

## The effects of exogenous NAD[+] are dependent on mitochondrial activity

The metabolome analysis showed that in HeLa cells, many of the TCA cycle intermediates were depleted upon treatment with NAD[+] (Fig. 4a). The levels of glutamine, glutamate and aspartate were maintained at higher levels in U2OS than in HeLa cells, suggesting a better capacity of U2OS cells to replenish TCA cycle intermediates. Moreover, under untreated conditions, the levels of the intermediates of the TCA cycle upstream of complex II-mediated succinate oxidation were lower in HeLa than in U2OS cells, and fumarate and malate were rapidly depleted in HeLa cells upon NAD[+] treatment. These differences indicated the potential impairment of complex II activity in HeLa cells, leading to increased sensitivity to complex I inhibition and NAD(H) depletion.

Given that pyrimidine biosynthesis is linked to ETC activity via DHODH, the transient accumulation of pyrimidines after NAD[+] treatment alluded to an increase in TCA cycle flux. To restrict anaplerosis of the TCA cycle and reduce the flux in the cycle, we used the glutaminase inhibitor bis-2-(5-phenylacetamido-1,3,4-thiadiazol-2-yl)ethyl sulfide (BPTES), which impairs the conversion of glutamine to glutamate in mitochondria, thereby preventing anaplerosis of α-ketoglutarate[33]. To further assess the dependence on complex I, cells were co-treated with NAD[+] and either BPTES or metformin, followed by measurements of EdU incorporation, DDR activation, and DNA fibre analysis. BPTES

rescued EdU incorporation and prevented accumulation of DNA damage in both NAD[+]-treated HeLa and U2OS cells (Fig. 4b). However, HeLa cells showed only partial rescue of fork speed with BPTES, while fork speed in U2OS cells was fully restored. Metformin impaired fork speed in HeLa but not in U2OS cells, and the effect of NAD[+] treatment was dominant over that of metformin in both cell lines (Fig. 4c,d and Extended Data Fig. 7a,b).

We next focused on the role of the ETC and NAD(H) in the regulation of DNA synthesis. We blocked the ATP synthase with oligomycin, and to investigate the importance of the glycolytic rate, we inhibited glycolysis using 2-deoxy-D-glucose (2-DG) (Fig. 4e,f and Extended Data Fig. 7c,d). Inhibition of glycolysis altered neither replication fork progression nor the effects of NAD[+] treatment, suggesting that glycolysis is not a limiting factor for NAD[+]-induced effects on DNA synthesis. Phenocopying the effect of BPTES, oligomycin rescued the inhibitory effect of NAD[+] treatment on fork speed partially in HeLa and fully in U2OS cells.

Since treatment with NAD[+] induced depletion of pyrimidines and accumulation of purines, we analysed the metabolome of cells co-treated with NAD[+] and oligomycin or BPTES. Oligomycin and BPTES countered the nucleotide imbalances caused by NAD[+] treatment (Fig. 4g and Extended Data Fig. 7e). Together, these results showed that exogenous NAD[+] regulates the levels of nucleotides by modulating the activity of the TCA cycle and the ETC.

Considering the involvement of the TCA cycle and the ETC in mediating the effects of exogenous NAD[+], we measured basal levels of oxygen consumption rate (OCR), an indicator of mitochondrial activity, and extracellular acidification rate (ECAR), a readout of glycolytic activity. While all cell lines were sensitive to exogenous NAD[+], no clear pattern in their basal metabolic rate under untreated conditions was apparent to predict their sensitivity (Fig. 5a). Further characterization of HeLa, U2OS and BJ-T cells showed that mitochondrial activity was stimulated by NAD[+] treatment while the glycolytic activity was diminished (Fig. 5b and Extended Data Fig. 8a). Furthermore, we found that the mitochondrial membrane potential was increased in both HeLa and U2OS cells upon treatment with NAD[+] but not with NR, indicating increased ETC activity and demonstrating differential effects between treatment with NAD[+] and its precursor. Both oligomycin and BPTES were able to reduce the elevated membrane potential induced by NAD[+]. To ascertain the specificity of these findings, we included treatment with the protonophore carbonyl cyanide-*p*-trifluoromethoxyphenylhydrazone (FCCP) as a control, aiming to abolish the mitochondrial membrane potential (Fig. 5c,d and Extended Data Fig. 8b). While FCCP alone hindered DNA replication in HeLa and U2OS cells, co-treatment with FCCP and NAD[+] had an additive effect on EdU incorporation only in HeLa cells, impairing DNA synthesis further (Fig. 5e). These experiments showed that the effect of NAD[+] is linked to hyperpolarization of the mitochondrial

**Fig. 3 | Exogenous NAD[+] affects nucleotide pools. a**, Relative ATP level and cell number for each cell line treated for 72 h (BJ and BJ-T) or 48 h (HeLa and U2OS) with NAMPTi or NAD[+]. Data are smoothed conditional means and CI95. Biological replicates, *n* = 18 (non-treated conditions) and *n* = 6 (all other conditions) except for BJ-T (non-treated, *n* = 12) and U2OS (NAMPTi treatments ATP levels: non-treated, *n* = 15; all others, *n* = 5). **b**, Mitochondrial NAD(H) quantification in HeLa cells treated for 24 h without (NT) or with NAD[+] or 10 nM NAMPTi. Data are mean + s.d., *n* = 3 biological replicates, two-tailed Student's *t*-test. **c**, Metabolomics analysis in HeLa cells treated for 24 h without (NT) or with NAD[+]. Asterisks for metabolites indicate significant differences to NT (adjusted *P* value <0.05, FDR-adjusted *P* values from two-tailed Student's *t*-test). *n* = 4 biological replicates of individual samples collected from four different experiments. **d**, Nucleotide levels from metabolomics analysis and replication fork speed relative to non-treated (NT) cells. HeLa cells were treated for 24 h with NAD[+]. Fork speed values from Fig. 2b. **e**, Nucleotide levels from metabolomics analysis and replication fork speed relative to non-treated HeLa cells (HeLa NT). HeLa and U2OS cells were treated with 2 mM NAD[+] as indicated. Fork speed values from

Fig. 2d and Extended Data Fig. 2d. **f**, Gene expression levels from RNA sequencing analysis in cells treated for 24 h without (NT) or with 2 mM NAD[+]. *y* axis shows normalized counts, *n* = 3 independent biological samples per condition sequenced once with multiplexing. **g**, Metabolomics analysis from cells treated without (NT) or with 2 mM NAD[+]. *y* axis shows relative metabolite quantification, *n* = 4 biological replicates of individual samples collected from four different experiments, bars represent means. **h**, QIBC analysis of EdU-positive cells treated for 24 h without (NT) or with AMP. Cells quantified: HeLa NT (left), 2,264; HeLa 100 μM, 3,059; HeLa NT (right), 1,948; HeLa 2 mM, 2,207; U2OS NT (left), 3,441; U2OS 100 μM, 2,842; U2OS NT (right), 1,607; U2OS 2 mM, 1,937, from two biological replicates. Two-tailed Welch's *t*-test. In **c**–**e**, means of *n* = 4 biological replicates of individual samples collected from four different experiments. For boxplots in **d**, **e**, **f** and **h**, the centre line indicates the median; box limits indicate the 25th and 75th percentiles; minima and maxima of whiskers extend 1.5 times the inter-quartile range from 25th and 75th percentiles, respectively. Numerical data are available in source data files.

membrane and, in HeLa cells, the presence of an additional mechanism for the regulation of pyrimidine biosynthesis.

## Exogenous NAD⁺ impairs pyrimidine biosynthesis

AMP can inhibit the UMPS enzymatic step[34], which occurs downstream of the mitochondrial conversion of dihydroorotate to orotate by DHODH in pyrimidine biosynthesis. The conversion of AMP to the

less toxic hypoxanthine has been proposed to protect against the cellular toxicity of AMP and could potentially confer resistance in U2OS cells, which express lower levels of enzymes involved in generating purine monophosphates compared with HeLa cells (Extended Data Fig. 5b). Since NAD⁺ contains an AMP moiety and could serve as a potential source of AMP, we explored the role of AMP in the regulation of DNA synthesis. We performed a series of experiments involving

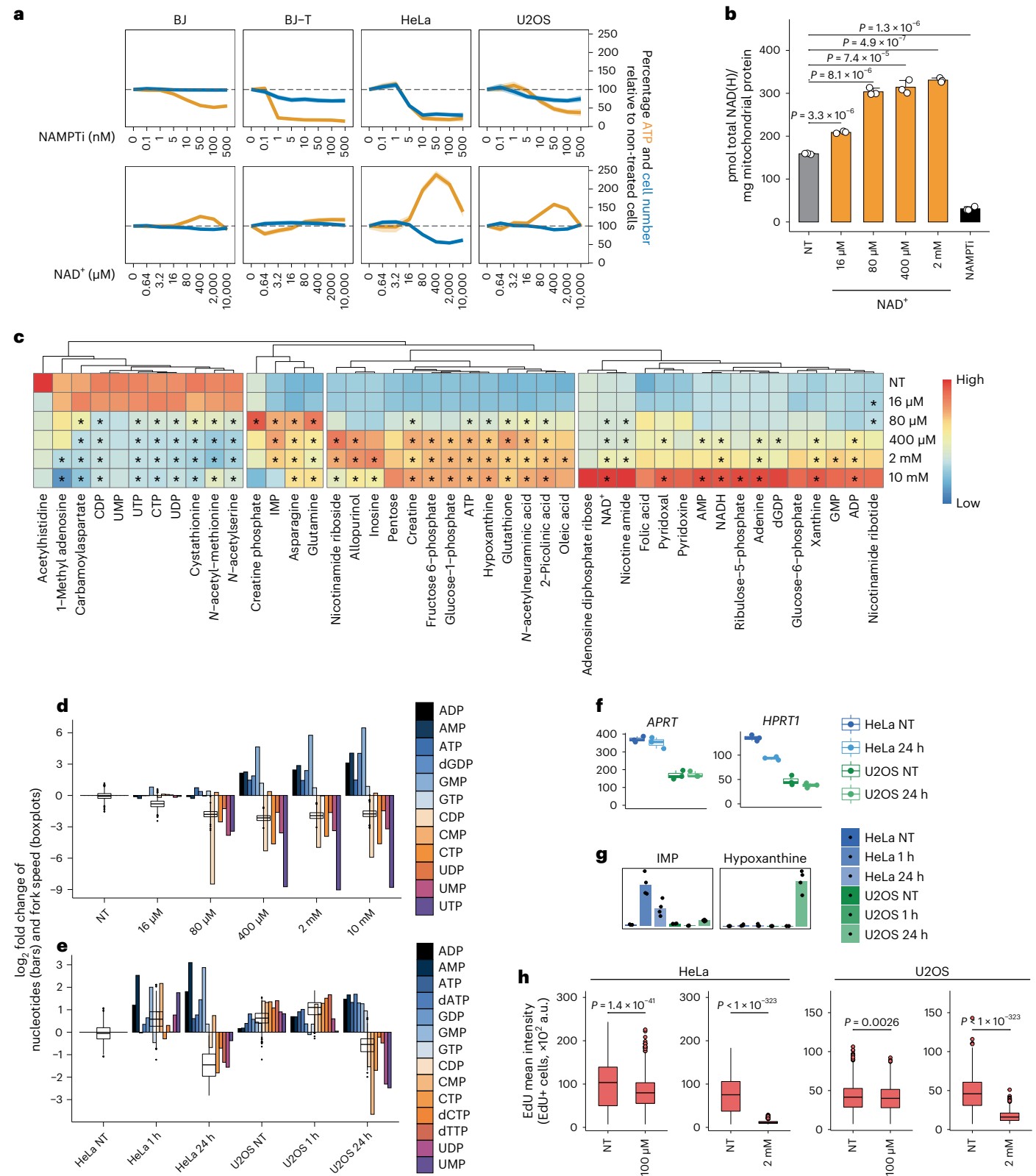

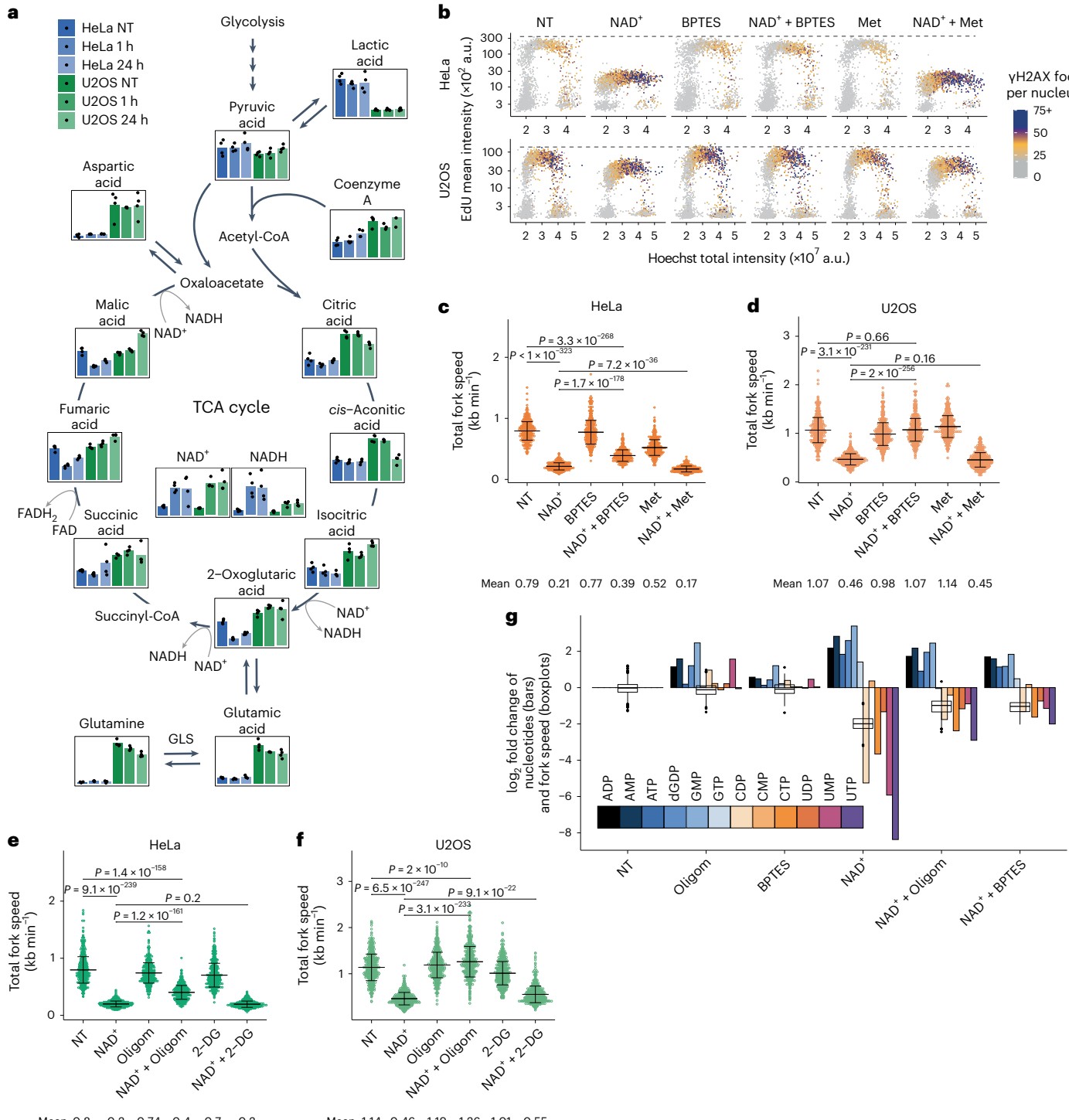

**Fig. 4 | Effects by exogenous NAD⁺ require mitochondrial metabolic proficiency. a**, Metabolomics analysis in cells treated without (NT) or with 2 mM NAD⁺. y axis shows relative metabolite quantification, bars represent means. $n = 4$ biological replicates of individual samples collected from four different experiments. GLS, glutaminase. **b**, QIBC analysis of γH2AX foci, EdU intensity and DNA content in cells treated for 24 h without (NT) or with 2 mM NAD⁺, 10 μM BPTES and/or 1 mM metformin (Met). Representative experiment from three (HeLa Met) and four (all other conditions) independent experiments. **c,d**, Replication fork speed in HeLa (**c**) and U2OS (**d**) cells treated as in **b**. Scored forks in **c**: NT, 524; NAD⁺, 554; BPTES, 516; NAD⁺ + BPTES, 520; Met, 522; NAD⁺ + Met, 511. Scored forks in **d**: NT, 529; NAD⁺, 575; BPTES, 559; NAD⁺ + BPTES, 518; Met, 535; NAD⁺ + Met, 651. **e,f**, Replication fork speed in HeLa (**e**) and U2OS (**f**) cells treated for 24 h without (NT) or with 2 mM NAD⁺, 5 μM oligomycin

(Oligom) and/or 1 mM 2-DG. Scored forks in **e**: NT, 521; NAD⁺, 535; Oligom, 536; NAD⁺ + Oligom, 537; 2-DG, 526; NAD⁺ + 2-DG, 541. Scored forks in **f**: NT, 553; NAD⁺, 554; Oligom, 520; NAD⁺ + Oligom, 515; 2-DG, 548; NAD⁺ + 2-DG, 571. **g**, Nucleotide levels from metabolomics analysis ($n = 4$ biological replicates of individual samples collected from four different experiments, bars represent means) and replication fork speed relative to non-treated (NT) cells from **c** and **e**. HeLa cells were treated for 24 h with 5 μM oligomycin (Oligom), 10 μM BPTES and/or 2 mM NAD⁺. In **c**–**f**, data are mean ± s.d., means indicated, forks were scored from two biological replicates, two-tailed Welch's *t*-test. For boxplots in **g**, the centre line indicates the median; box limits indicate the 25th and 75th percentiles; minima and maxima of whiskers extend 1.5 times the inter-quartile range from 25th and 75th percentiles, respectively. Numerical data are available in source data files.

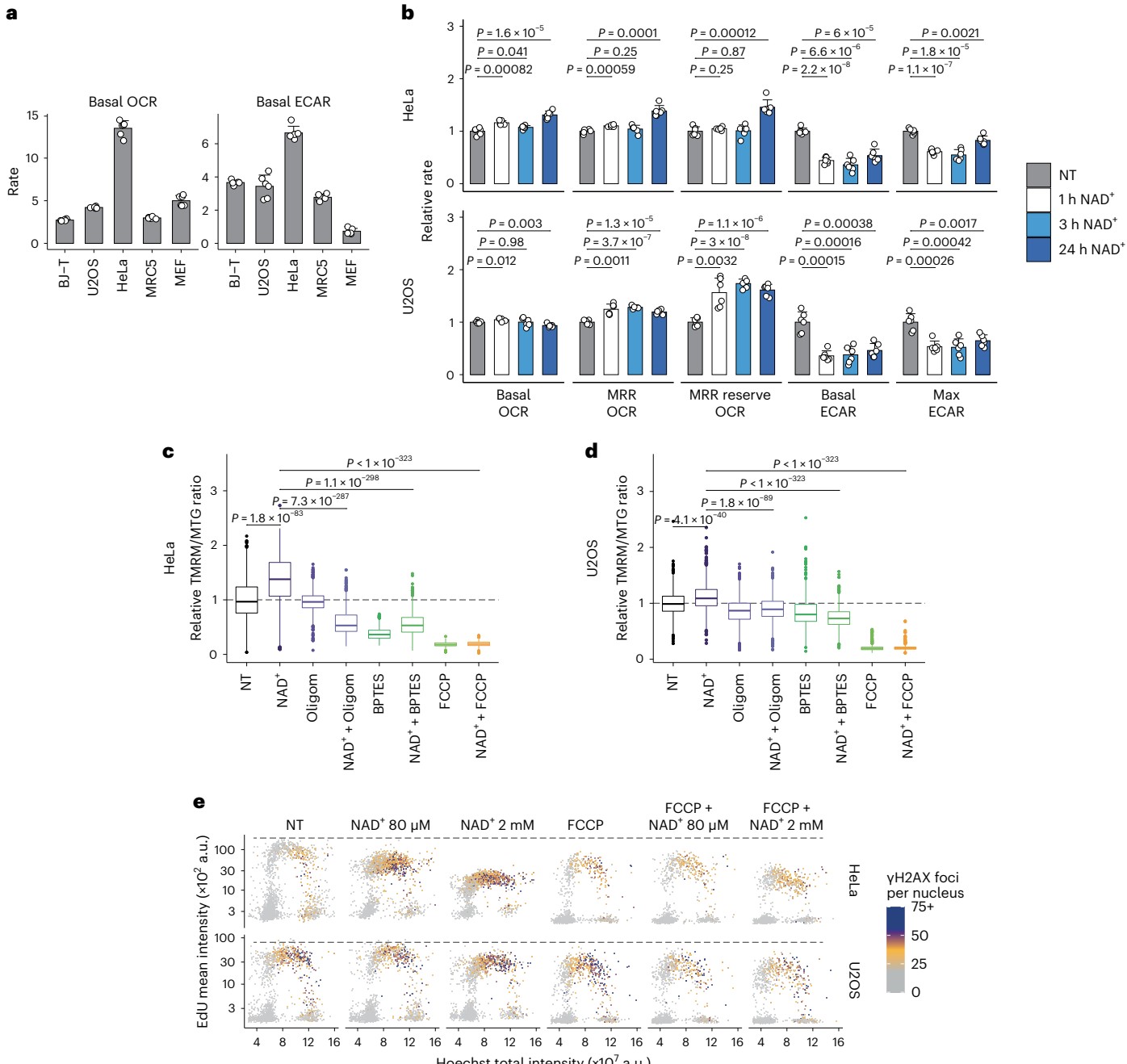

**Fig. 5 | NAD⁺ treatment affects mitochondrial respiration. a**, Basal OCR and ECAR for the indicated cell lines. Data are mean + s.d. Independent experiments, $n = 5$ (BJ-T) and $n = 6$ (all other cell lines). **b**, Metabolic parameters from mitochondrial stress analysis relative to non-treated (NT) cells. HeLa and U2OS cells were treated with 2 mM NAD⁺ as indicated. Data are mean + s.d., $n = 6$ independent experiments, two-tailed Welch's $t$-test. MRR, maximum respiratory rate. **c,d**, QIBC analysis of mitochondrial membrane potential in HeLa (**c**) and U2OS (**d**) cells using TMRM and MTG cell-permeant dyes. The ratios of TMRM and MTG intensities are normalized to non-treated (NT) cells. Cells were treated for 24 h with 2 mM NAD⁺, 5 μM oligomycin (Oligom), 10 μM BPTES and/or 10 μM FCCP. Cells quantified in **c**: NT, 1,570; NAD⁺, 923; Oligom, 2,107; NAD⁺ + Oligom,

1,518; BPTES, 1,274; NAD⁺ + BPTES, 1,258; FCCP, 1,261; NAD⁺ + FCCP, 1,330. Cells quantified in **d**: NT, 1,177; NAD⁺, 2,025; Oligom, 821; NAD⁺ + Oligom, 808; BPTES, 635; NAD⁺ + BPTES, 1,029; FCCP, 634; NAD⁺ + FCCP, 462. Data are from a representative experiment performed twice with similar results. Two-tailed Welch's $t$-test. **e**, QIBC analysis of γH2AX foci, EdU intensity and DNA content in HeLa and U2OS cells treated for 24 h without (NT) or with NAD⁺ as indicated and/or 10 μM FCCP. Representative experiment from two independent experiments. For boxplots in **c** and **d**, the centre line indicates the median; box limits indicate the 25th and 75th percentiles; minima and maxima of whiskers extend 1.5 times the inter-quartile range from 25th and 75th percentiles, respectively. Numerical data are available in source data files.

co-treatment of cells with AMP and either oligomycin or BPTES. Our results showed a pronounced impairment of DNA synthesis induced by AMP in HeLa cells compared with U2OS cells, a phenomenon that remained unalleviated by either oligomycin or BPTES (Fig. 6a), suggesting that the effect of exogenous NAD⁺ on mitochondrial activity

is not mediated through its conversion to AMP. Uridine can also serve as a source for pyrimidine synthesis independently of DHODH activity and, indeed, supplementation with uridine was able to fully rescue DNA synthesis in cells treated with NAD⁺ (Fig. 6b). If NAD⁺ treatment results in the indirect impairment of DHODH activity, we predicted

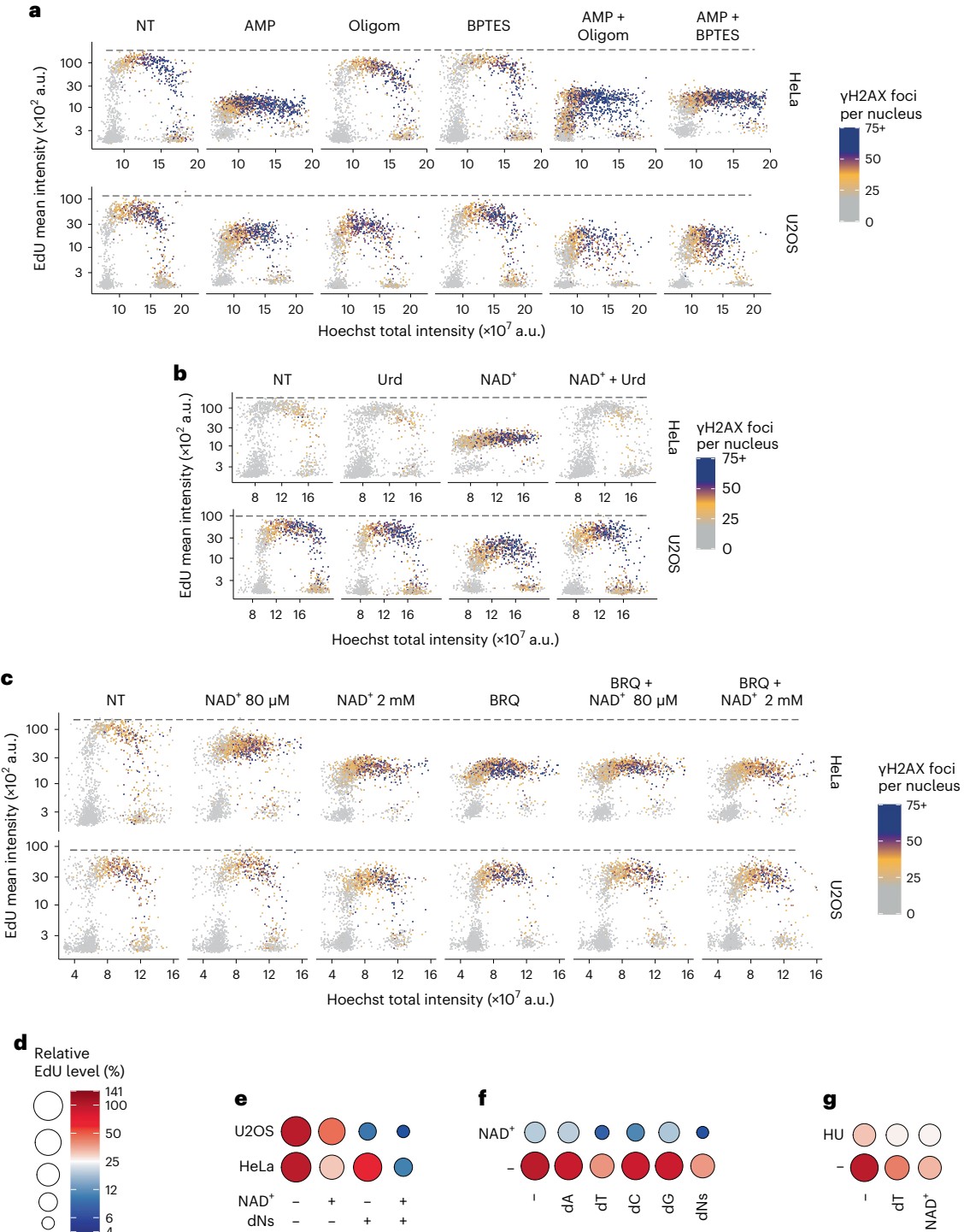

**Fig. 6 | Exogenous NAD⁺ impairs pyrimidine biosynthesis. a**, QIBC analysis of γH2AX foci, EdU intensity and DNA content in HeLa and U2OS cells treated for 24 h without (NT) or with 2 mM AMP, 5 µM oligomycin (Oligom) and/or 10 µM BPTES. Representative experiment from two independent experiments. **b**, Analysis of γH2AX foci, EdU intensity and DNA content with QIBC in HeLa and U2OS cells treated for 24 h without (NT) or with 100 µM uridine (Urd) and/or 2 mM NAD⁺. Representative experiment from three (HeLa conditions NT, NAD⁺ and NAD⁺ + Urd) and two (all other conditions) independent experiments **c**, QIBC analysis of γH2AX foci, EdU intensity and DNA content in HeLa and U2OS

cells treated for 24 h without (NT) or with NAD⁺ as indicated and/or 2 µM BRQ. Representative experiment from two independent experiments. **d–g**, Flow cytometry analysis of EdU incorporation in S-phase cells relative to non-treated (NT) cells. The code for all panels is shown in **d**. U2OS and HeLa cells (**e**) or HeLa cells (**f** and **g**) were treated as indicated for 24 h with 2 mM NAD⁺, 10 µM of one deoxynucleoside (dA, dT, dC and dG) or all combined (dNs), and/or 1 mM HU. Flow cytometry experiments were performed twice with similar results. Numerical data are available in source data files.

that there should be no additional effects upon co-treating cells with NAD⁺ and brequinar (BRQ), a strong inhibitor of the DHODH[35]. Both at the level of global DNA synthesis and DDR, BRQ phenocopied the

effect of exogenous NAD⁺ (Fig. 6c). Moreover, we did not observe any further effect on DNA synthesis in cells treated with both BRQ and NAD⁺.

It has been shown that an excess of thymidine increases intracellular TTP, an allosteric inhibitor of the RNR in the conversion of CDP to dCDP, and as a consequence, deoxycytidine triphosphate becomes depleted[36]. Our analysis of metabolites showed that exogenous NAD$^+$ resulted in depletion of deoxycytidine triphosphate and deoxythymidine triphosphate. Next, we tested whether exogenous deoxynucleosides could rescue the effect of NAD$^+$. Equimolar excess of deoxynucleosides was unable to rescue genomic replication upon NAD$^+$ treatment, on the contrary, this exacerbated the inhibition of DNA synthesis rather than ameliorating it (Fig. 6d,e). Incubation with individual deoxynucleosides identified thymidine as the inhibitor of DNA synthesis (Fig. 6f). HU combined with NAD$^+$ had additive effects on the inhibition of DNA synthesis, indicating that high levels of NAD$^+$ do not impair RNR's activity (Fig. 6g). Together, these results showed that exogenous NAD$^+$ affects the ETC and, indirectly, impairs DHODH activity and pyrimidine biosynthesis.

### Cytostatic effects of exogenous NAD$^+$ and pyrimidine analogues

U2OS cells depend heavily on glutamine for survival[37]. The accumulation of AMP in HeLa cells after NAD$^+$ treatment may affect the synthesis of pyrimidines independently of the activity of ETC. We, therefore, predicted that exogenous NAD$^+$ would only impair DNA synthesis in U2OS cells under growth conditions that sustain mitochondrial OXPHOS, that is in the presence of glutamine, while HeLa cells would be sensitive to NAD$^+$ treatment independently of growth conditions. To test this, we measured DNA synthesis in cells cultured in minimal growth medium supplemented differentially with glucose, glutamine and NAD$^+$. DNA synthesis in U2OS cells was only reduced by NAD$^+$ in the presence of glutamine, while HeLa cells were sensitive to NAD$^+$ treatment across all tested conditions (Fig. 7a,b). Furthermore, exogenous pyruvate or aspartate, required for the biosynthesis of both purines and pyrimidines, did not rescue the effects of NAD$^+$ treatment (Extended Data Fig. 8c). Together, these results indicate that exogenous NAD$^+$ regulates pyrimidine biosynthesis both via mitochondrial activity and AMP-mediated mechanisms in a cell-type dependent manner.

Without the mitochondrial NAD$^+$ transporter SLC25A51, OXPHOS is impaired and cell proliferation is compromised[19,38,39]. Thus, the depletion of this transporter should directly influence the effects of exogenous NAD$^+$ (Extended Data Fig. 9a). While SLC25A51 depletion alone severely impaired DNA synthesis in U2OS cells, HeLa cells were only mildly affected (Fig. 7c). SLC25A51 depletion suppressed NAD$^+$-induced effects in U2OS cells, while SLC25A51-depleted HeLa cells remained sensitive to NAD$^+$. Furthermore, SLC25A51 depletion severely compromised the growth of U2OS cells while only slightly affecting HeLa cells (Fig. 7d). Importantly, in the absence of SLC25A51, NAD$^+$ was unable to elevate the mitochondrial levels of NAD(H) in either cell line (Extended Data Fig. 9b–d). Together, these results show that the effects of exogenous NAD$^+$ on cell proliferation and genomic DNA synthesis depend on its mitochondrial transporter, while in some cell lines such as HeLa, the synthesis of pyrimidines can also be modulated by AMP.

Given our observation that U2OS cells maintain a larger pool of nucleotides than HeLa cells, we treated cells with BRQ to inhibit the de novo synthesis of pyrimidines. Whereas the proliferation of HeLa cells was impaired by BRQ, U2OS cells remained unaffected, suggesting a greater dependency on de novo pyrimidine biosynthesis in HeLa cells (Extended Data Fig. 9e). Since the clinically approved PARP inhibitor olaparib competes with NAD$^+$ for the catalytic site of PARP1, we investigated whether co-treatment with NAD$^+$ could influence the effect of olaparib on cell viability. The growth of non-transformed human fibroblast MRC5 cells was highly sensitive to treatment with olaparib but not with NAD$^+$, while on the contrary, HeLa cells were more resistant to olaparib but sensitive to NAD$^+$. The combination of olaparib and NAD$^+$ resulted in a small but significant increase in cell death in HeLa cells (Fig. 7e and Extended Data Fig. 9f–j).

Given that NAD$^+$-treated cells had compromised pyrimidine biosynthesis, we combined a low dose of NAD$^+$ (80 μM) with pyrimidine analogues used for cancer therapy. Co-treatment of cells with NAD$^+$ and 5-fluorouracil (5-FU), a uracil analogue that inhibits thymidine synthesis, or gemcitabine, a deoxycytidine analogue that inhibits DNA synthesis, additively impaired cell growth and increased cell death in HeLa cells. Similar to U2OS cells, MRC5 cells were sensitive to gemcitabine but not to 5-FU or NAD$^+$ (Fig. 7f and Extended Data Fig. 9k–n). The observation of additive effects upon co-treatment with NAD$^+$ and either of the two pyrimidine analogues in HeLa cells, alongside pronounced sensitivity of MRC5 cells to gemcitabine but not 5-FU, presents a foundation for the clinical potential of therapeutic interventions involving NAD$^+$ and 5-FU for treatment of cancers that heavily rely on de novo synthesis of pyrimidines.

## Discussion

We found that exogenous NAD$^+$ levels play a pivotal role in orchestrating the interplay between basal metabolism and DNA replication. The accepted belief that intact NAD$^+$ molecules are incapable of entering human cells[13] has been challenged by both historical[40] and recent evidence, which highlights the ability of extracellular NAD$^+$ to utilize non-specific transporters for cytoplasmic entry[15–17,41]. Since treatment with extracellular NAD$^+$ precursors did not recapitulate the effects of NAD$^+$, our data suggest that at least a fraction of the exogenous intact NAD$^+$ may enter the cells and reach mitochondrial crestae[18,42], the latter probably via the mitochondrial transporter SLC25A51. We demonstrate that a slight elevation in exogenous NAD$^+$ levels increases the flux of the TCA cycle and the ETC with substantial implications for the cell's metabolism and genome duplication (Extended Data Fig. 10a). Intriguingly, precursors of NAD$^+$ increased the mitochondrial levels of NAD$^+$, but at a slower rate than exogenous NAD$^+$, and were unable to impact genomic DNA synthesis. A possible explanation for this phenomenon is the inability of precursors to induce immediate changes in mitochondria. Such temporal delay may provide cells with the opportunity to adapt and respond differently to high levels of mitochondrial NAD$^+$. Indeed, after days of NR supplementation, more mitochondrial cristae were detected in skeletal muscle cells[26], most likely to buffer the effects of high NAD$^+$ levels. In this scenario, the precursors of NAD$^+$ may be a safe option to increase intracellular NAD$^+$; however, at the cellular level, their long-term administration may induce DNA damage.

Following NAD$^+$ exposure, a rapid and substantial elevation in all nucleotide levels was observed, coinciding with an accelerated rate of fork progression. However, this phenotype cannot solely be ascribed to an excess of deoxynucleotides, as supplementation with exogenous deoxynucleotides fails to accelerate replication forks[43,44]. On the contrary, high concentrations of deoxynucleosides, in particular thymidine, can inhibit DNA synthesis. Hence, the initial increase in the speed of DNA synthesis can probably be attributed to an overall metabolic acceleration. Based on our results, we propose a model where accelerated metabolism results in higher activity of the TCA cycle and the ETC, leading to pyrimidine depletion, RS and DDR (Fig. 7g and Extended Data Fig. 10b–d). Our results additionally revealed that NAD$^+$ treatment can induce mitochondrial fatigue, especially in cells that heavily rely on de novo pyrimidine synthesis. This observation underscores the potential of utilizing the dependence on pyrimidine synthesis as a predictive parameter for sensitivity to NAD$^+$, with promising implications for oncology. Moreover, high levels of NAD$^+$ may clear cells with defective mitochondria, which could be a protective mechanism to maintain organismal fitness.

NAD$^+$ levels are lower in ageing humans, and exercise in elderly people restores their NAD$^+$ levels similar to those found in younger individuals. The abundance of NAD$^+$ was positively associated with mitochondrial function and muscle strength after exercising[45]. However, highly structured exercise may be too demanding for some elderly individuals; therefore, there is currently a great focus on finding a way

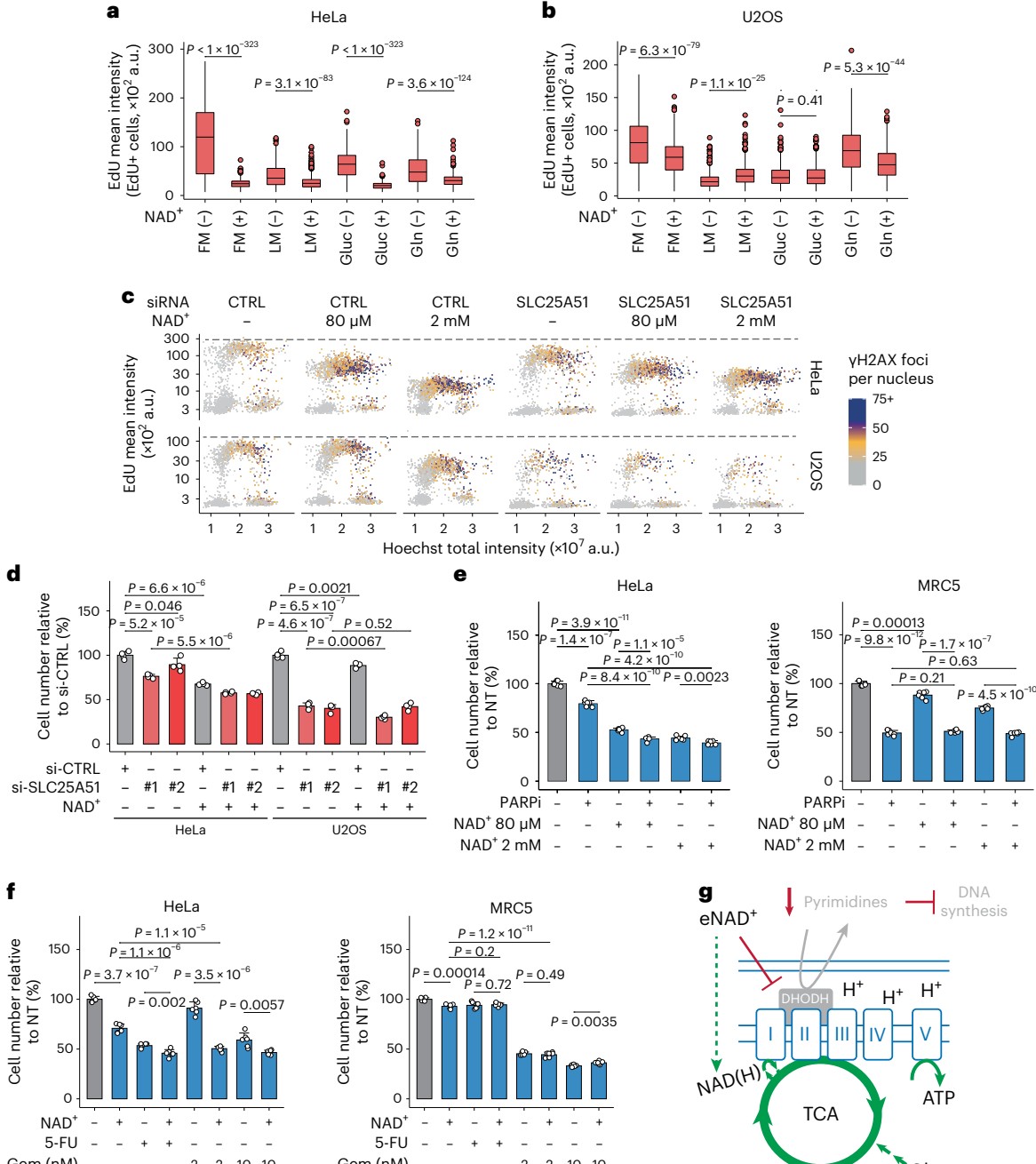

**Fig. 7 | Cytostatic effects by NAD⁺ treatment and chemotherapeutic agents.**
**a,b**, QIBC analysis of EdU-positive HeLa (**a**) and U2OS (**b**) cells treated for 24 h as indicated with 2 mM NAD⁺ in complete medium (FM) or minimal medium (LM) supplemented with glucose (Gluc) or glutamine (Gln). Cells quantified in **a**: FM, 2,055; FM + NAD⁺, 4,577; LM, 1,397; LM + NAD⁺, 1,500; Gluc, 2,803; Gluc + NAD⁺, 4,144; Gln, 1,367; Gln + NAD⁺, 2,086, cells quantified in **b**: FM, 1,678; FM + NAD⁺, 1,572; LM, 599; LM + NAD⁺, 649; Gluc, 808; Gluc + NAD⁺, 495; Gln, 737; Gln + NAD⁺, 1,479, from $n = 2$ (Gln) or 3 (all other conditions) biological replicates. Two-tailed Welch's $t$-test. **c**, QIBC analysis of γH2AX foci, EdU intensity and DNA content in cells treated with control siRNA (CTRL) or si-SLC25A51 #1 and for 24 h with NAD⁺ as indicated. Representative experiment from two independent experiments. **d**, Relative cell number to cells treated only with control siRNA (si-CTRL) for each cell line. Cells were transfected with one of two si-SLC25A51 (#1 and #2) for 48 h

and treated for 24 h with 2 mM NAD⁺. Mean + s.d., $n = 4$ biological replicates, two-tailed Student's $t$-test. **e**, Relative cell number to non-treated (NT) cells. Cells were treated for 72 h with 1 μM PARPi and/or NAD⁺ as indicated. **f**, Relative cell number to non-treated (NT) cells. Cells were treated for 72 h with 80 μM NAD⁺, 1 μM 5-FU and/or gemcitabine (Gem) as indicated. **g**, Proposed model. Exogenous NAD⁺ initially boosts cellular metabolism and DNA replication, but after prolonged treatment, pyrimidine biosynthesis is impaired and pyrimidines are depleted. In **e** and **f**, mean + s.d., $n = 6$ biological replicates, two-tailed Welch's $t$-test. For boxplots in **a** and **b**, centre line indicates the median; box limits indicate the 25th and 75th percentiles; minima and maxima of whiskers extend 1.5 times the inter-quartile range from 25th and 75th percentiles, respectively. Numerical data are available in source data files.

to maintain the 'youth-associated' levels of NAD⁺ in elderly people. Some precursors of NAD⁺ are used as dietary supplements, readily obtainable without the need for a prescription. Their benefits are still debatable[46] but promise to relieve some age-related changes. However, the molecular mechanism(s) of how these supplements work are not yet fully understood.

Catabolic products of NAD+ include AMP and adenosine. Adenosine plays a role in homeostatic sleep regulation[47] and is a well-known neuromodulator with a wide range of effects. The increased levels of adenosine inhibit excessive neural excitation often seen during morphine/cocaine withdrawal[48]; therefore, it is not surprising that full-NAD+-based treatments seem to be highly effective in addiction therapies[49]. Finally, our results pointed to NAD+ as an indirect natural regulator of the activity of DHODH. This enzyme has been a target for cancer therapy; however, due to high toxicity, the DHODH inhibitor BRQ is not used in the clinic[35]. We propose that for clinical use in cancer therapy, a combined treatment with NAD+ and 5-FU could be beneficial for patients with highly proliferative tumours, especially those that rely on de novo nucleotide synthesis. Together, the results presented here open new perspectives for assessing NAD+ supplementation and/ or exercise in cancer therapy and biohacking strategies to enhance the healthspan and longevity of the globally ageing human population.

## Online content

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

## Methods

### Statistics and reproducibility

Statistical analyses were performed using R software (v4.0.4). The statistical tests used are indicated in the figure legends. No statistical methods were used to pre-determine the sample size. The experiments were not randomized, and the investigators were not blinded to allocation during experiments and outcome assessment.

### Cell culture

Human U2OS, HeLa, BJ, BJ-5ta (BJ-T), T98G, U87 MG and MRC5 cell lines and mouse embryonic fibroblasts (MEF) were grown in Dulbecco's modified Eagle medium (Gibco 31966021) supplemented with heat-inactivated 10% foetal bovine serum (FBS; Gibco) and 1% penicillin–streptomycin (Gibco). All cell lines were purchased from the American Type Culture Collection (U2OS, HTB-96; HeLa, CCL-2; BJ, CRL-2522; BJ-5ta, CRL-4001; T98G, CRL-1690; U87 MG, HTB-14; MRC5, CCL-171; MEF, CRL-2991) and were frequently tested for mycoplasma contamination. Where indicated, cells were grown and treated in a minimal medium (Gibco A1443001) containing heat-inactivated 10% FBS and 1% penicillin–streptomycin without or with glucose (4.5 g l$^{-1}$, Sigma) or GlutaMAX (4 mM, Gibco).

### Drugs and compounds

Before treatment, fresh medium was added to cells for all conditions. The drugs and compounds used were: 2-DG (Sigma D8375), 5-FU (Sigma F6627), adenosine monophosphate (Sigma A1752), aspartate (Sigma A7219), BRQ (Sigma SML0113), BPTES (Sigma SML0601), CD38 inhibitor (Sigma 538763), thymidine (dT, Sigma T1895), 2′-deoxyadenosine (dA, Sigma D8668), 2′-deoxycytidine (dC, Sigma D3897), 2′-deoxyguanosine (dG, Sigma 854999), gemcitabine (Sigma G6423), metformin (Sigma PHR1084), NAD (Abcam ab120403), NMN (Selleckchem S5259), NR (Biochempartner BCP20094), GMX1778 (NAMPTi, Selleckchem S8117), oligomycin A (Sigma 75351), olaparib (PARPi, Selleckchem S1060), EX 527 (SIRT1i, Selleckchem S1541), uridine (Sigma U3003), sodium pyruvate (Sigma S8636), carbonyl cyanide 4-(trifluoromethoxy)phenylhydrazone (FCCP, Sigma C2920) and HU (Sigma H8627).

### Metabolomics analysis

Cells were washed twice in cold phosphate-buffered saline (PBS) on ice before quenching and metabolite extraction in ice-cold 80% methanol (MS grade), $n = 4$ biological replicates per condition per cell line. Sample analysis was carried out by MS-Omics. The analysis was performed using a Thermo Scientific Vanquish LC coupled to a Thermo Q Exactive HF MS. An electrospray ionization interface was used as ionization source. Analysis was performed in negative and positive ionization mode. For semi-polar metabolites, the UPLC was performed using a slightly modified version of the protocol described by ref. 50. For polar metabolites, the UPLC was performed using a slightly modified version of the protocol described by ref. 51. Reconstituted samples were diluted 1:3 with 5-μl injections or 1:4 with 2-μl injections, consistently for each experiment. Peak areas were extracted using Compound Discoverer 3.1 (Thermo Scientific). Identification of compounds was performed at four levels; level 1: identification by retention times (compared against in-house authentic standards), accurate mass (with an accepted deviation of 3 ppm) and MS/MS spectra; level 2a: identification by retention times (compared against in-house authentic standards), accurate mass (with an accepted deviation of 3 ppm); level 2b: identification by accurate mass (with an accepted deviation of 3 ppm) and MS/MS spectra; level 3: identification by accurate mass alone (with an accepted deviation of 3 ppm). Only metabolites with high-confidence identification (level 1 + 2a) were used for downstream analysis, and the obtained areas were normalized to cell count. The non-specific metabolites hexose1/2/3 were not included in visualizations. Heatmaps were made using the pheatmap (v1.0.12) package in R with hierarchical clustering of metabolites based on Euclidean distance. In heatmap visualizations, only metabolites are shown with an absolute log$_2$ fold change >1.25 for at least one treatment condition relative to untreated cells. Adjusted $P$ values were obtained from false discovery rate (FDR) correction of $P$ values from two-tailed Student's $t$-tests. In panels of metabolic pathway(s) with metabolites, the $y$ axis on metabolite plots shows relative abundances from the relative quantification.

### RNA isolation

Total RNA was extracted from cells with the RNeasy Mini Kit (QIAGEN) according to the manufacturer's instructions, including QIAshredder spin columns (QIAGEN) for homogenization. RNA concentrations were measured with NanoDrop, and all RNA samples were estimated as pure (260/280 nm ratio ≥2).

### Library preparation and nanopore sequencing

Total RNA ($n = 3$ biological replicates per cell line per condition) was used as input for the library preparation SQK-PCB109 kit (Oxford Nanopore Technologies, ONT) according to manufacturer's instructions. Two libraries were prepared: one with HeLa and U2OS samples, and one with BJ and BJ-T samples. In the barcoding-PCR step, 14 cycles with 4 min extension time were performed. Barcoded complementary DNA samples were pooled into a library with a final mass of 100 ng. Each library was sequenced on one FLO-MIN106 flow cell (ONT) for 48 h using MinKNOW software (v20.06.4).

### Long-read RNAseq analysis

Base-calling was performed with Guppy (v4.0.15, ONT) using the high-accuracy model, –qscore_filtering and –min_qscore 7 settings. Reads that passed the quality cutoff were demultiplexed and trimmed for adapters using Porechop (v0.2.4), and then filtered on the basis of quality score using NanoFilt (v2.6.0) with the -q 7 setting. nanoQC (v0.9.4) and NanoPlot (v1.29.0) were used for quality inspections[52]. In total, 617,000–1,601,000 reads were obtained per sample. To first assemble the transcriptome for the samples, reads were aligned to the hg38 genome with Minimap2 (ref. 53) (v2.17) using the -ax splice option. Samtools[54] (v1.9) was used for alignment sorting, Binary Alignment/Map conversion and genome indexing. Wherever relevant, the GENCODE v35 comprehensive annotation was provided. Alignments were then passed to StringTie2 (ref. 55) (v2.1.4) for transcript assembly with the -L -G options, and the subsequent merging of transcripts for all samples into a non-redundant transcriptome. GffCompare (v0.12.1) was used to compare and annotate the assembled transcriptome using the reference annotation. Reads were then realigned to the assembled transcriptome converted to FASTA format with GffRead[56] (v0.12.3) using Minimap2 with -ax map-ont, -p 0.99 and -N 100 settings. Transcriptome alignments were quantified with Salmon[57] (v1.4.0) using settings for long reads (–noErrorModel and –noLengthCorrection). For transcripts assigned to annotated genes by GffCompare (class codes: =, c, k, m, n, j, e, o), the original transcript counts were imported into R (v4.0.4) and summarized to gene level using the tximport package[58] (v1.18.0). The differential expression analysis was performed using the DESeq2 package[59] (v1.30.1) in R, and the gene-set enrichment analysis was done with STRING[60] (v11-0b). For panels with boxplots of individual gene expression levels, counts normalized with DESeq2 were plotted.

### Immunofluorescence and QIBC

Cells grown on coverslips were incubated with 10 μM EdU for 30 min at 37 °C before fixation in 4% cold formaldehyde for 10 min at room temperature (RT). Cells were then permeabilized in 1% Triton X-100 for 15 min. After washing with PBS and blocking in PBS+ (PBS with 1% bovine serum albumin and 0.1% Tween 20), coverslips were incubated as follows: Click-iT EdU Imaging Kits (Invitrogen) were used for EdU detection according to the manufacturer's instructions, then the primary antibody for 1 h, and goat anti-mouse Alexa Fluor 488 (Invitrogen A11029, 1:1,000) together with Hoechst 33342 (Invitrogen) for 45 min.

In between and after the last incubation, coverslips were washed with PBS and PBS+. Finally, coverslips were washed with ddH$_2$O, air-dried and mounted using Fluoromount-G (Invitrogen). The primary antibody used was p-H2A.X S139 (γH2AX, Abcam ab22551, 1:500). Single-cell level QIBC image acquisition was performed using a ScanR high-content screening station (Olympus) equipped with an IX83 motorized microscope (Olympus), a UPLSAPO 2 40× air immersion objective and an ORCA-Flash4.0 V3 camera. The analysis was performed using ScanR analysis software (v3.2.0) and R. To detect EdU-positive cells, the same EdU intensity threshold was applied in all analyses. In each cell cycle plot, the same number of cells is visualized per condition per cell line. The QIBC data pre-processing is outlined in Extended Data Fig. 1c.

## Flow cytometry cell cycle analysis

Cells were incubated with 10 µM EdU for 30 min at 37 °C, trypsinized, washed with cold PBS, fixed in ice-cold 70% ethanol and stored at −20 °C overnight. Cells were then washed with PBS and PBS+, and EdU was detected using the Click-iT EdU Alexa Fluor 647 Imaging Kit (Invitrogen) according to the manufacturer's instructions. After washing with PBS+, cells were incubated with Hoechst 33342 in PBS for 30 min, washed with PBS+ and resuspended in PBS. Cells were analysed on a FACSVerse flow cytometer (Becton Dickinson), and data analysis was performed using FlowJo (v10) software (for gating example see Extended Data Fig. 1b).

## siRNA transfection

Knockdown with siRNA was performed using the Lipofectamine RNAiMAX Transfection Reagent (Invitrogen) according to the manufacturer's instructions. Control siRNA (luciferase GL3 siRNA) was obtained from Eurofins Genomics. siGENOME siRNAs targeting *SLC25A51* (#1: D-007358–01, #2: D-007358–02) were obtained from Dharmacon RNAi Technologies.

## Immunoblotting

Whole-cell lysates were prepared in the Laemmli sample buffer (50 mM Tris pH 6.8, 2% sodium dodecyl sulfate, 100 mM dithiothreitol, 10% glycerol and 0.1% bromophenol blue), separated by sodium dodecyl sulfate–polyacrylamide gel electrophoresis and transferred onto nitrocellulose membranes using the iBlot 2 Dry Blotting System (Thermo Fisher Scientific). Membranes were then blocked in 5% milk in PBS with 0.1% Tween 20 and incubated with primary antibodies, followed by horseradish peroxidase-conjugated secondary antibodies (Vector Laboratories, anti-rabbit PI-1000, anti-mouse PI-2000, 1:10,000). Proteins were visualized with ECL detecting reagents (Cytiva and Lumigen). The primary antibodies used were: p-CHK1 S317 (Cell Signaling 2344, 1:250), CHK1 (Santa Cruz sc-8408, 1:100), p-CHK2 T68 (Abcam ab32148, 1:200), CHK2 (Abcam ab109413, 1:20,000), p-H2A.X S139 (γH2AX, Abcam ab22551, 1:500), β-Actin (Sigma A1978, 1:5,000), p-AMPKα T172 (Cell Signaling 2535, 1:500), AMPKα (Cell Signaling 2603, 1:500), p-mTOR S2481 (Cell Signaling 2974, 1:250), mTOR (Cell Signaling 2972, 1:250), α-Tubulin (GeneTex GTX628802, 1:5,000), SLC25A51/MCART1 (Abcam ab237054, 1:500), Vinculin (Sigma V9131, 1:50,000-100,000), VDAC (Cell Signaling 4661, 1:500), CIV (Cell Signaling 11967, 1:500), OPA1 (Abcam ab157457, 1:1,000) and ATPB (Abcam ab14730, 1:50).

## Total NAD(H) quantification

Total NAD(H) was extracted and measured using the colourimetric NAD/NADH Assay Kit (Abcam) according to the manufacturer's instructions with the following modification: the volume of NADH/NAD Extraction Buffer was adjusted to the cell count for each sample (400 µl per 10$^6$ cells). Assay measurements were obtained using a SpectraMax iD3 plate reader (Molecular Devices).

## Mitochondrial isolation and NAD(H) quantification

Cells were washed twice with cold PBS on ice before collection and mitochondrial isolation performed using Mitochondria Isolation Kit for Cultured Cells (Abcam) according to manufacturer's instructions. Pelleted mitochondria were then resuspended in lysis buffer for protein quantification and immunoblot analysis, as described above, or in NADH/NAD Extraction Buffer for NAD(H) quantification. NAD(H) quantification was performed using the colourimetric NAD/NADH Assay Kit (Abcam) according to the manufacturer's instructions starting from the extraction step, and measurements were obtained using a SpectraMax iD3 plate reader (Molecular Devices).

## Cell viability assay

To obtain the total and dead cell count, cells were stained with Hoechst 33342 and propidium iodide (Invitrogen) for 10 min at 37 °C, and analysed using a Celigo imaging cytometer (Nexcelom). The viable cell count for each condition was calculated subsequently as (total cell number − dead cell number).

## Calcium assay

Cells cultured and treated in medium supplemented with 1% FBS were incubated for the last 30 min of the treatment time with Fluo-8 dye-loading solution from the Fluo-8 Calcium Flux Assay Kit (Abcam, ab112129) to assess intracellular calcium levels. Fluorescence intensity was then measured on a SpectraMax iD3 plate reader (Molecular Devices) according to manufacturer's instructions. Subsequently, cell viability assay was performed as described above to obtain cell numbers for normalization.

## ATP measurement

The quantification of ATP levels was performed using the ATPlite Luminescence Assay System (PerkinElmer) according to the manufacturer's instructions with white opaque microplates (PerkinElmer). Measurements were obtained on a SpectraMax iD3 plate reader (Molecular Devices). Subsequently, cell viability assay was performed as described above to obtain cell numbers for normalization.

## Mitochondrial membrane potential analysis

Live cells were stained with 20 nM tetramethylrhodamine methyl ester (TMRM, Thermo Fisher Scientific), 50 nM MitoTracker Green (MTG, Invitrogen) and Hoechst 33342 for 30 min in normal culturing conditions. Live-cell imaging was performed using QIBC in complete FluroBrite Dulbecco's modified Eagle medium (Gibco) under normal culturing conditions. For the flow cytometry analysis, cells were stained with TMRM as above, resuspended in PBS and analysed immediately on a FACSVerse flow cytometer.

## DNA fibres

Cells were incubated with 25 µM CldU (Sigma-Aldrich) for 20 min, gently washed with the pre-warmed medium, and incubated with 250 µM IdU (Sigma-Aldrich) for subsequent 20 min. Pulse-labelled cells were collected and DNA fibre spreads were prepared as previously described[3]. For each experimental condition, DNA fibres were stretched on five microscope slides, and two to three slides were processed for staining. CldU was detected first using a rat anti-BrdU antibody (Serotec, OBT0030, 1:500), followed by IdU detection using a mouse anti-BrdU antibody (Becton Dickinson, 347580, 1:500). Secondary antibodies used were anti-rat DyLight 550 (Thermo Fisher Scientific, SA5-10019, 1:400) and donkey anti-mouse Alexa Fluor 488 (Invitrogen A21202, 1:400). Images of well-spread fibres were acquired on a LSM800 confocal microscope (Carl Zeiss) with a Plan-Apochromat 63x/1.4 numerical aperture oil immersion objective (Carl Zeiss). The acquisition was performed semi-automatically using tile arrays and LSM software autofocus. LSM ZEN (lite v2.6 blue edition) software was used to manually analyse double-labelled replication forks. Between 100 and 250 replication forks were scored for each slide, and fork measurements for all slides for the same experimental condition were pooled together. At least one additional

independent experiment was performed, and, if the experiments did not differ statistically, the total number of DNA fibres from all experiments is presented.

### BrUTP incorporation assay

The in situ transcription analysis was performed according to ref. 61. Cells were seeded on coverslips and reached 75% confluency on the day of the experiment. Coverslips were washed with PBS and incubated for 2 min at RT in the permeabilization buffer (20 mM Tris–HCl pH 7.4, 5 mM $MgCl_2$, 0.5 mM egtazic acid, 25% glycerol, 0.1% Triton X-100, 1 mM phenylmethylsulfonyl fluoride and 20 U ml$^{-1}$ ribonuclease inhibitor). Coverslips were then incubated for 8 min at 37 °C in the transcription buffer (50 mM Tris–HCl pH 7.4, 10 mM $MgCl_2$, 0.5 mM egtazic acid, 25% glycerol, 1 mM phenylmethylsulfonyl fluoride, 100 mM KCl, 20 U ml$^{-1}$ ribonuclease inhibitor, 200 μM BrUTP, 200 μM CTP, 200 μM GTP and 1 mM ATP). Coverslips were then washed with cold PBS and fixed in 4% formaldehyde for 15 min at RT. An anti-BrdU antibody (Becton Dickinson, 347580, pure B44 100T, 1:500) was used for the detection of BrUTP incorporation into nascent transcripts together with an anti-Fibrillarin antibody (Abcam ab5821, 1:500). Images were acquired on a LSM800 confocal microscope with a Plan-Apochromat 63x/1.4 numerical aperture oil immersion objective.

### Mitochondrial flux analysis

Real-time OCRs and ECARs were measured using the Seahorse XFe96 extracellular flux analyzer (Seahorse Bioscience). Cells ($2 \times 10^4$ per well) were seeded in XFe96 cell culture microplates (Seahorse Bioscience) and incubated overnight. OCR and ECAR were measured at 37 °C in the Seahorse assay medium (10 mM glucose and 10 mM pyruvate, pH 7.4). The sequential addition of different compounds that affect mitochondrial respiration was used to investigate different mitochondrial respiratory states. First, 'basal OCR' was measured to assess cellular respiration during normal physiological conditions. Then, oligomycin (1 μM), an inhibitor of the mitochondrial $F_0/F_1$ ATP synthase (complex V), was added to acquire information about the mitochondrial 'proton leak' (OCR happening mainly due to the leakage of protons across the mitochondrial inner membrane), and 'oligomycin-sensitive OCR' (the OCR fraction directly interconnected with mitochondrial ATP production). To obtain the maximum respiratory rate (MRR), that is, the maximum capacity of the electron transport system, the uncoupler FCCP (1 μM) was added. The difference between the MRR and basal OCR indicates the reserve capacity of the electron transport system, which is the potential OCR capacity that can be engaged during energy-demanding conditions. Lastly, a combination of antimycin-A (2.5 μM) and rotenone (2.5 μM) was added to inhibit complex III and I, respectively, to acquire 'non-mitochondrial OCR' (NM OCR) used for correction. Specific OCR parameters were calculated as follows: Basal OCR = OCR before compound addition − NM OCR; Proton leak = OCR after oligomycin − NM OCR; MRR = OCR after FCCP − NM OCR; oligomycin-sensitive OCR = OCR before compound addition − OCR after oligomycin; MRR reserve = OCR after FCCP − OCR before compound addition. Basal ECAR was measured to give an estimation of basal glycolytic activity under physiological conditions. Then, 'max ECAR' (the maximum glycolytic capacity) was measured after addition of oligomycin, which prevents ATP production by mitochondria and causes cells to compensate through increased glycolytic ATP production, resulting in increased ECAR. Finally, 'reserve ECAR' was calculated as the difference between max ECAR and basal ECAR (reserve ECAR = max ECAR − basal ECAR). This gives an estimation of spare glycolytic capacity that cells can utilize upon increased energy demand. After OCR and ECAR measurements, cells were stained with Hoechst 33342 and propidium iodide in PBS, and subjected to the cell viability assay as described above. Viable cell numbers were then used for the correction of OCR measurements.

### Reporting summary

Further information on research design is available in the Nature Portfolio Reporting Summary linked to this article.

### Data availability

RNA-seq data reported in this study have been deposited in the European Nucleotide Archive with accession PRJEB64552. Differential expression analyses are provided in source data files. Metabolomics data are provided as supplementary tables. Other numerical source data are provided in source data files. All other data supporting the findings of this study are available from the corresponding author on request. Source data are provided with this paper.

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

### Acknowledgements

We thank C. Dinant and the BioImaging Core Facility at the Danish Cancer Institute. M. Barisic and A. Aranda-Anzaldo for comments and discussion on the early versions of the paper. J.B. laboratory is supported by grants from The Danish Council for Independent Research (number DFF-1026-00241B), The Novo Nordisk Foundation (grant number 0060590), The Swedish Research Council VR-MH 2014-46602-117891-30 and The Swedish Cancer Foundation/Cancerfonden (number 170176). A.M.-M. laboratory is supported by grants, KBVU R302-A17590 and The Danish National Research Foundation (project CARD, DNRF 125).

### Author contributions

S.H.N.M. and A.M.-M. conceptualized and designed the study. S.H.N.M., J.M.M.-M., A.A.R., A.H., G.P., G.M., M.L., L.G.J. and A.M.-M.

performed experiments. P.G. designed selected experiments, provided critical input and edited the paper. J.B. and A.M.-M. supervised the study. S.H.N.M., J.M.M.-M., A.M.-M. and J.B. periodically discussed and interpreted the data. J.B. and A.M.-M. provided the funding for this project. S.H.N.M., J.M.M.-M., A.M.-M. and J.B. wrote the paper. All authors approved the final paper.

## Competing interests

The authors declare no competing interests.

## Additional information

**Extended data** is available for this paper at https://doi.org/10.1038/s41556-023-01280-z.

**Correspondence and requests for materials** should be addressed to Jiri Bartek or Apolinar Maya-Mendoza.

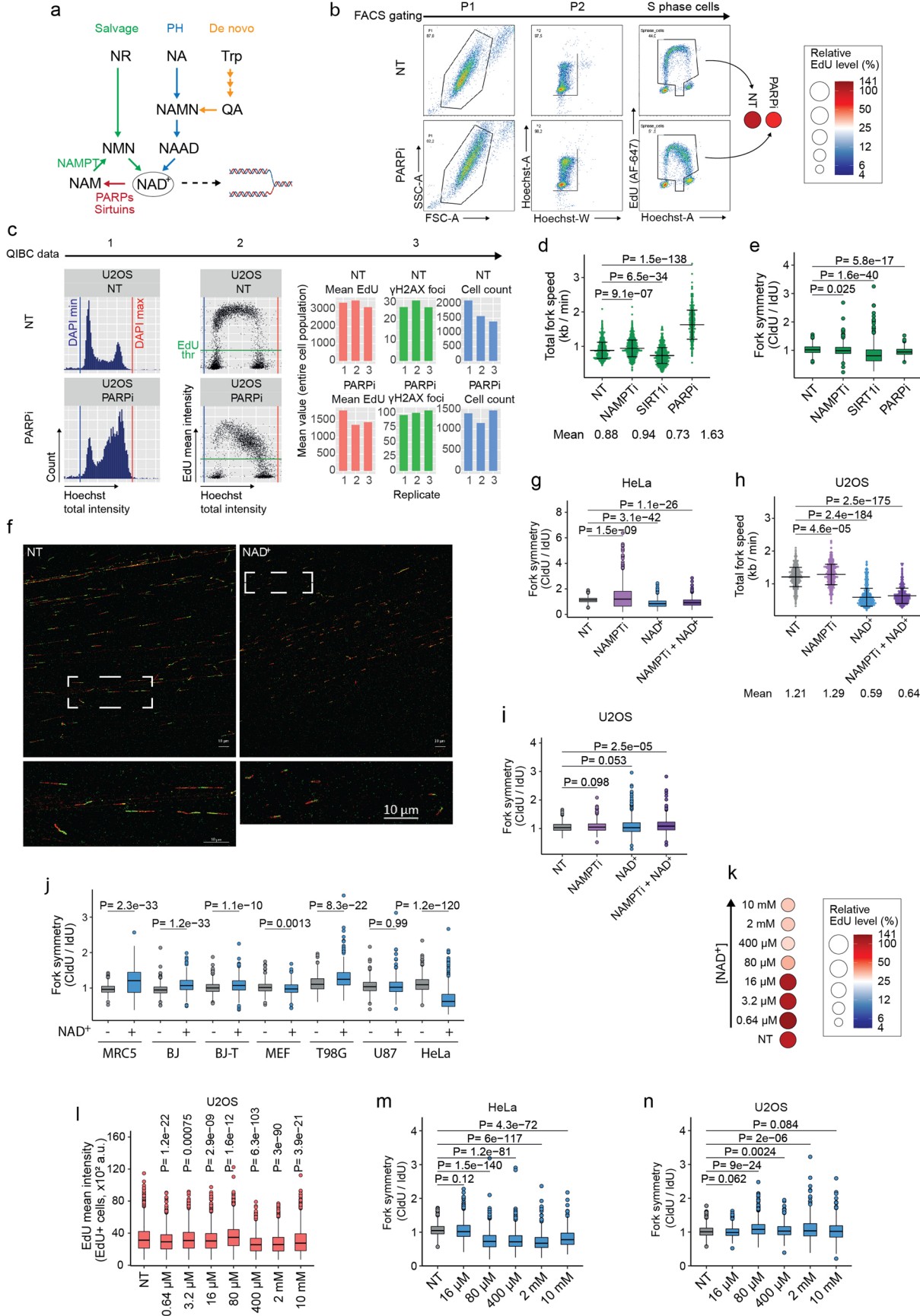

**Extended Data Fig. 1 | See next page for caption.**

**Extended Data Fig. 1 | Exogenous NAD⁺ inhibits DNA synthesis. a**, NAD⁺ synthesis pathways. NR, nicotinamide riboside; NMN, nicotinamide mononucleotide; NAM, nicotinamide; PH, Preiss-Handler; NA, nicotinic acid; NAMN, nicotinic acid mononucleotide; NAAD, nicotinic acid dinucleotide; Trp, tryptophan; QA, quinolinic acid. **b**, Flow cytometry gating to exclude duplets and dead cells (P1 + P2). EdU mean intensities for S-phase populations were normalized to non-treated (NT) cells. Example for U2OS treated for 24 h with 10 μM PARPi. **c**, QIBC thresholding to remove duplets and dead cells (1), and obtain EdU-positive cell populations (2), before analysis of γH2AX foci, EdU incorporation and/or DNA content (3). **d-e**, Replication fork speed (**d**, mean ± SD) and symmetry (**e**) in U2OS cells treated for 24 h with 10 nM NAMPTi, 2 μM SIRT1i or 10 μM PARPi. Scored forks: non-treated (NT) = 689; NAMPTi = 792; SIRT1i = 882; PARPi = 426. **f**, Representative images of stained DNA fibres from CldU- and IdU-pulsed HeLa cells treated for 24 h without (NT) or with 2 mM NAD⁺. Representative images for 7 experiments performed with similar results. **g**, Symmetry of forks from Fig. 1e. **h-i**, Replication fork speed

(**h**, mean ± SD) and symmetry (**i**) in U2OS cells treated as in Fig 1e. Scored forks: non-treated (NT) = 515; NAMPTi = 525; NAD⁺ = 576; NAMPTi+NAD⁺ = 549. **j**, Symmetry of forks from Fig. 1f. **k**, Flow cytometry analysis of EdU incorporation in S-phase cells relative to non-treated (NT) cells. HeLa cells were treated for 24 h with NAD⁺ as indicated. The experiment was performed twice with similar results. **l**, QIBC analysis of EdU incorporation in EdU-positive U2OS cells treated for 24 h with NAD⁺ as indicated. Cells quantified: non-treated (NT) = 4,812; 0.64 μM = 3,139; 3.2 μM = 4,908; 16 μM = 4,858; 80 μM = 4,538; 400 μM = 3,412; 2 mM = 4,466; 10 mM = 4,349, from 2 (0.64 μM, 400 μM) and 3 (all other conditions) biological replicates. Two-tailed Welch's t-test compared to NT. **m**, Symmetry of forks from Fig. 2b. **n**, Symmetry of forks for Fig. 2c. Replication forks were scored for two biological replicates, two-tailed Welch's t-test. For boxplots in panels **e, g, i, j, l, m, n**, the center line indicates the median; box limits indicate the 25th and 75th percentiles; minima and maxima of whiskers extend 1.5 times the inter-quartile range from 25th and 75th percentiles, respectively. Numerical data are available in Source Data Files.

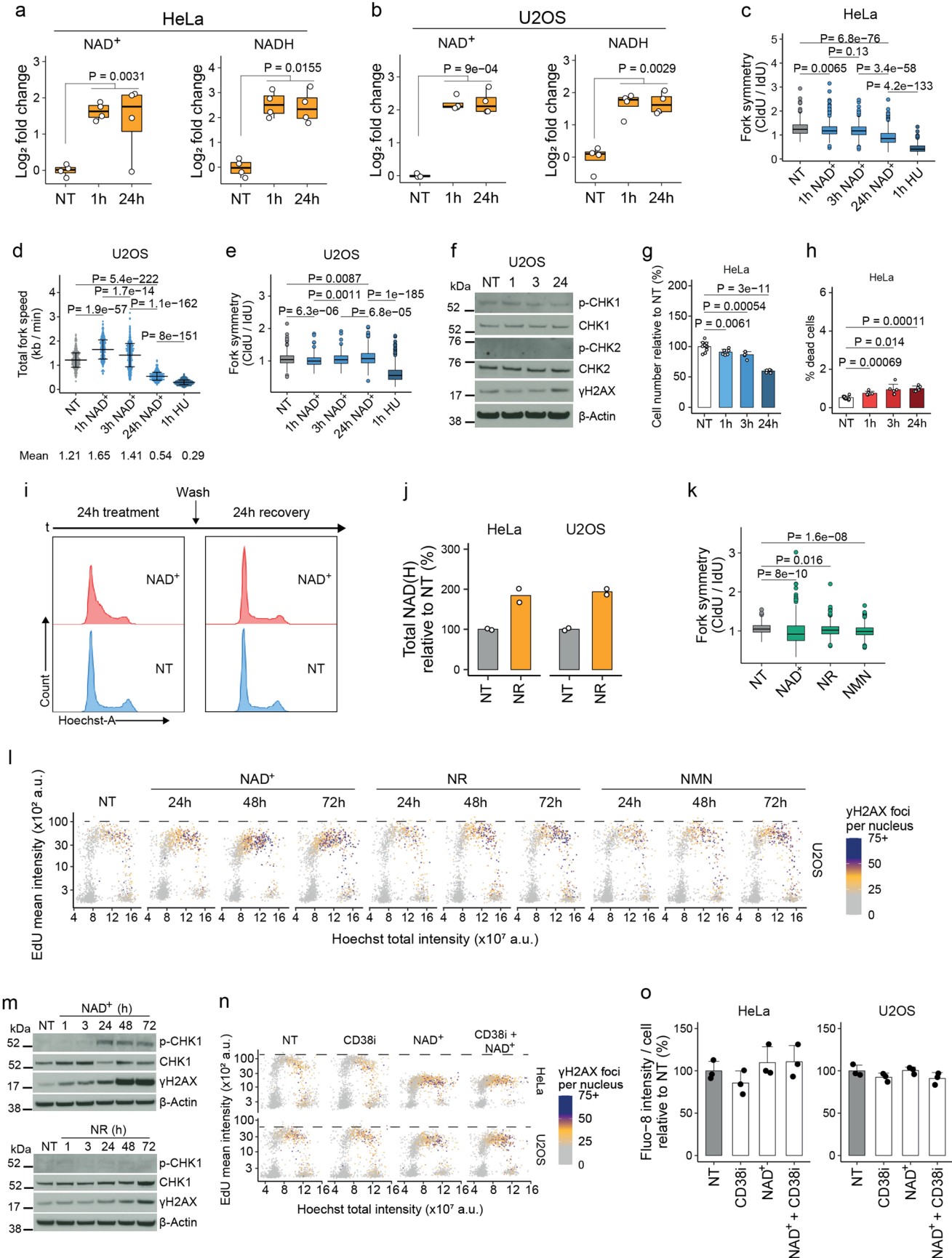

Extended Data Fig. 2 | See next page for caption.

**Extended Data Fig. 2 | Exogenous NAD⁺ causes replication stress. a**, **b**, NAD⁺ and NADH levels from metabolomics analysis relative to non-treated (NT) cells. HeLa (**a**) and U2OS (**b**) cells were treated with 2 mM NAD⁺ as indicated. n = 4 biological replicates, Welch's one-way ANOVA. **c**, Replication fork symmetry in HeLa cells treated with 2 mM NAD⁺ or hydroxyurea (HU) as indicated. Symmetry of forks from Fig. 2d. **d**, **e**, Replication fork speed (**d**, mean ± SD) and symmetry (**e**) in U2OS cells treated as in **c**. Scored forks: non-treated (NT) = 521; 1 h NAD⁺ = 364; 3 h NAD⁺ = 493; 24 h NAD⁺ = 562; 1 h HU = 593. **f**, Immunoblot from U2OS cells treated without (NT) or with 2 mM NAD⁺ for the indicated hours. **g**, **h**, Relative cell number (**g**) and cell death (**h**) of HeLa cells treated without (NT) or with 2 mM NAD⁺ as indicated. Mean+SD, n = 12 (NT) and n = 6 (all other conditions) biological replicates, two-tailed Welch's t-test. **i**, Flow cytometry analysis of DNA content in HeLa cells treated for 24 h with 2 mM NAD⁺ followed by 24 h in non-treated (NT) conditions. t, time. **j**, NAD(H) quantification in cells treated for 24 h without (NT) or with 2 mM NR. Bars represent means, n = 2 biological replicates.

**k**, Symmetry of forks from Fig. 2f. **l**, QIBC analysis of γH2AX foci, EdU intensity and DNA content in U2OS cells treated without (NT) or with 2 mM of NAD⁺, NR or NMN as indicated. Representative from 3 independent experiments. **m**, Immunoblots from HeLa cells treated without (NT) or with 2 mM NAD⁺ or NR as indicated. **n**, QIBC analysis of γH2AX foci, EdU intensity and DNA content in cells treated for 24 h without (NT) or with 1 μM CD38i and/or 2 mM NAD⁺. **o**, Relative intracellular calcium changes measured with the Fluo-8 indicator in cells treated as in **n**. Mean+SD, n = 3 biological replicates. Replication forks were scored for two biological replicates, two-tailed Welch's t-test. Immunoblots and flow cytometry experiments were performed twice with similar results. For boxplots in panels **a**, **b**, **c**, **e**, **k**, the center line indicates the median; box limits indicate the 25th and 75th percentiles; minima and maxima of whiskers extend 1.5 times the inter-quartile range from 25th and 75th percentiles, respectively. Numerical data and uncropped immunoblots are available in Source Data Files.

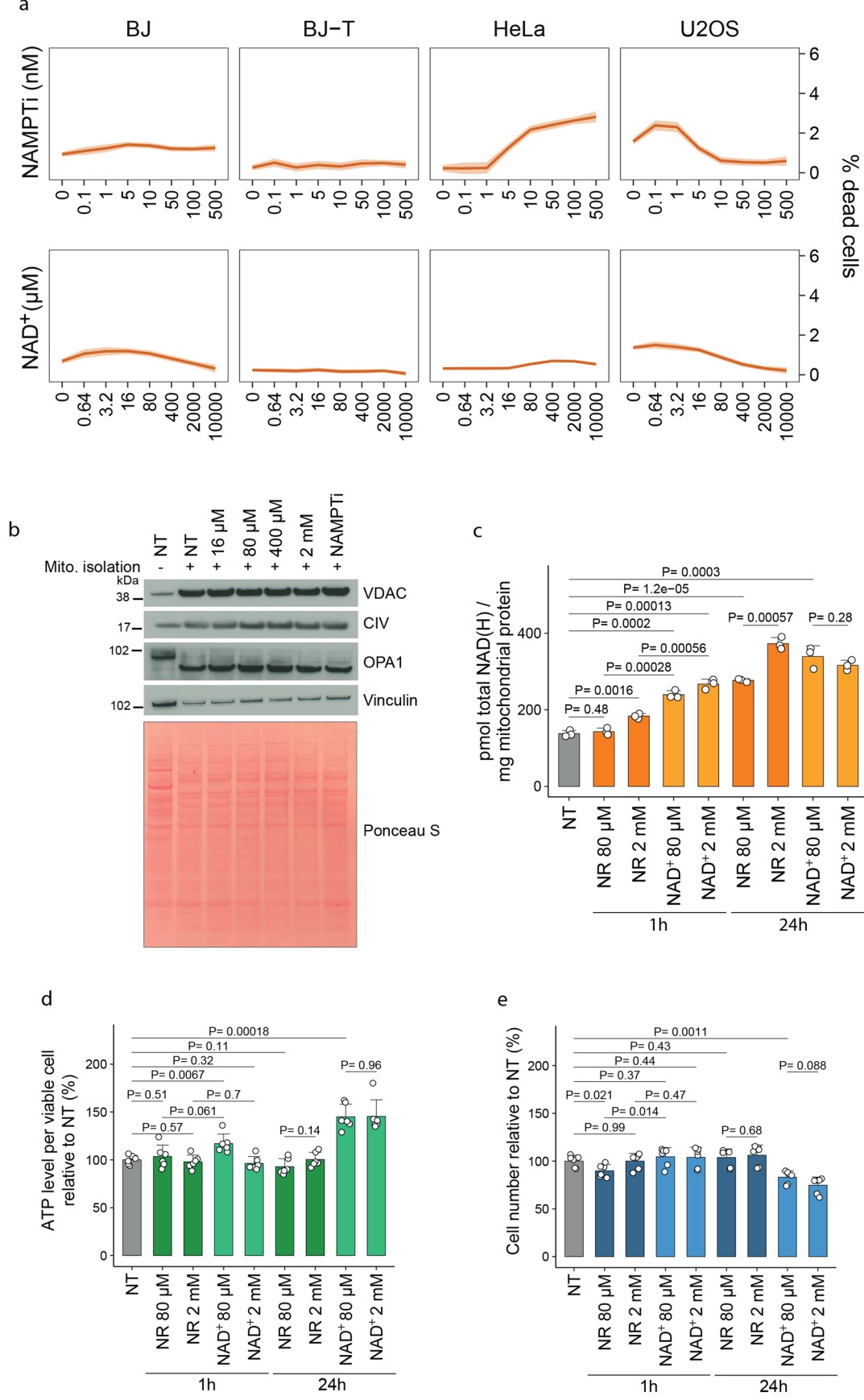

**Extended Data Fig. 3 | See next page for caption.**

**Extended Data Fig. 3 | Cytostatic effects and ATP levels after NAD+ treatment.**
**a**, Percentage of dead cells for the indicated cell lines treated with NAMPTi or NAD+ as indicated for 72 h (BJ and BJ-T) or 48 h (HeLa and U2OS). Data are smoothed conditional means and CI95. Biological replicates, n = 18 (non-treated conditions) and n = 6 (all other conditions) except for BJ-T (non-treated = 12). **b**, Immunoblot confirmation of mitochondrial (mito.) isolation from HeLa cells treated for 24 h without (NT) or with NAD+ as indicated or 10 nM NAMPTi.

The experiment was performed twice with similar results. **c**, Mitochondrial NAD(H) quantification in HeLa cells treated without (NT) or as indicated with NAD+ or NR. Data are mean+SD, n = 3 biological replicates, two-tailed Student's t-test. **d-e**, Relative ATP levels (**d**) and cell number (**e**) of HeLa cells treated without (NT) or with NAD+ or NR as indicated. Data are mean+SD, n = 6 biological replicates, two-tailed Welch's t-test. Numerical data and uncropped immunoblots are available in Source Data Files.

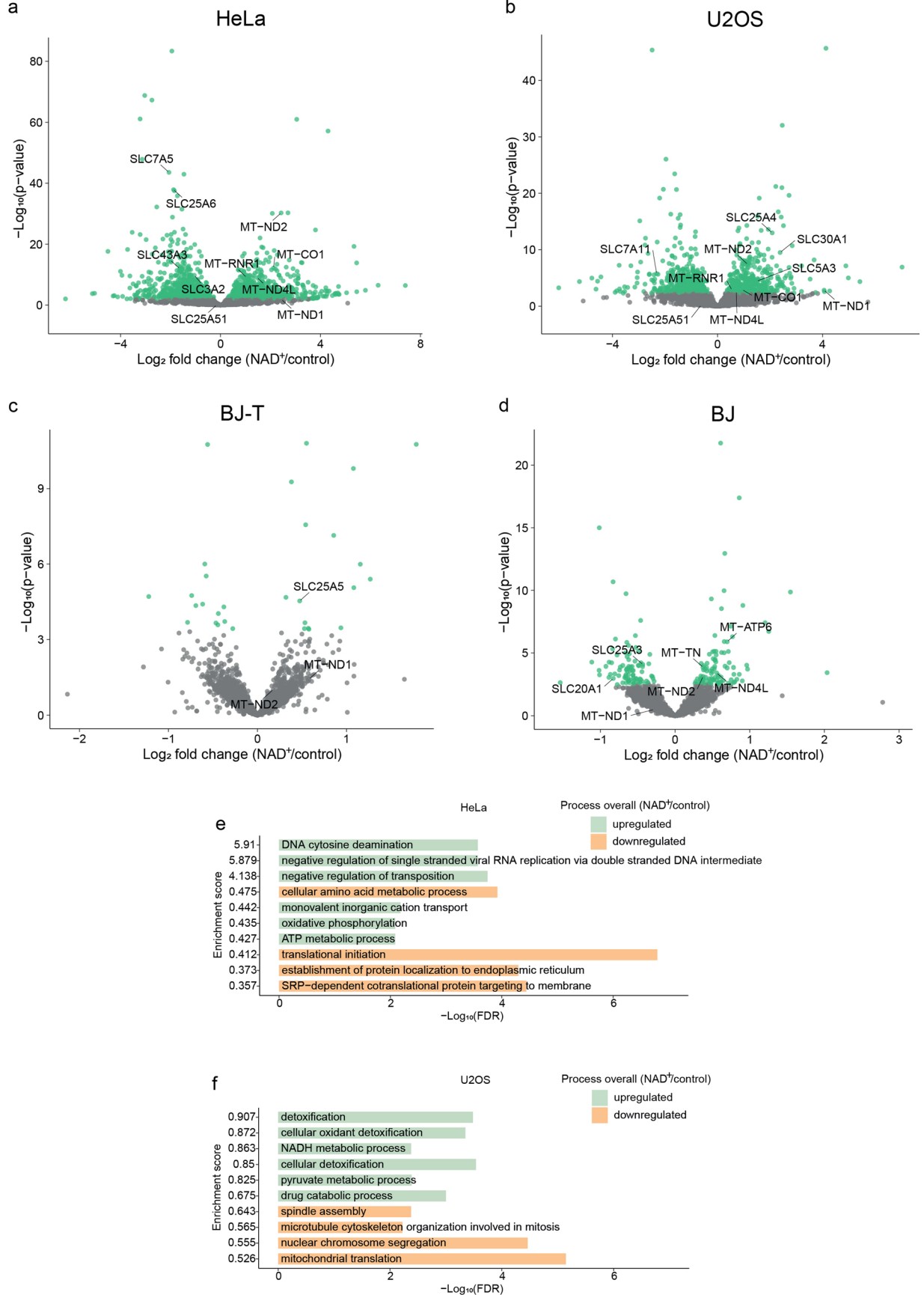

**Extended Data Fig. 4 | See next page for caption.**

**Extended Data Fig. 4 | RNA-seq analysis of NAD$^+$-treated cells. a-d**, Volcano plots of differentially expressed genes after treatment for 24 h with 2 mM NAD$^+$ in HeLa (**a**), U2OS (**b**), BJ-T (**c**), and BJ (**d**) cells. n = 3 independent biological samples per condition sequenced once with multiplexing. Green color indicates significant changes (FDR-adjusted p-value < 0.05), non-significant changes are in grey. SLC-, solute carrier genes; MT-, mitochondrial genes. **e-f**, Top 10 most enriched biological processes (FDR < 0.01) from STRING (Functional Enrichment Analysis) affected by 2 mM NAD$^+$ for 24 h treatment from RNA-seq analysis of HeLa (**e**) and U2OS (**f**) cells. Numerical data are available in Source Data Files.

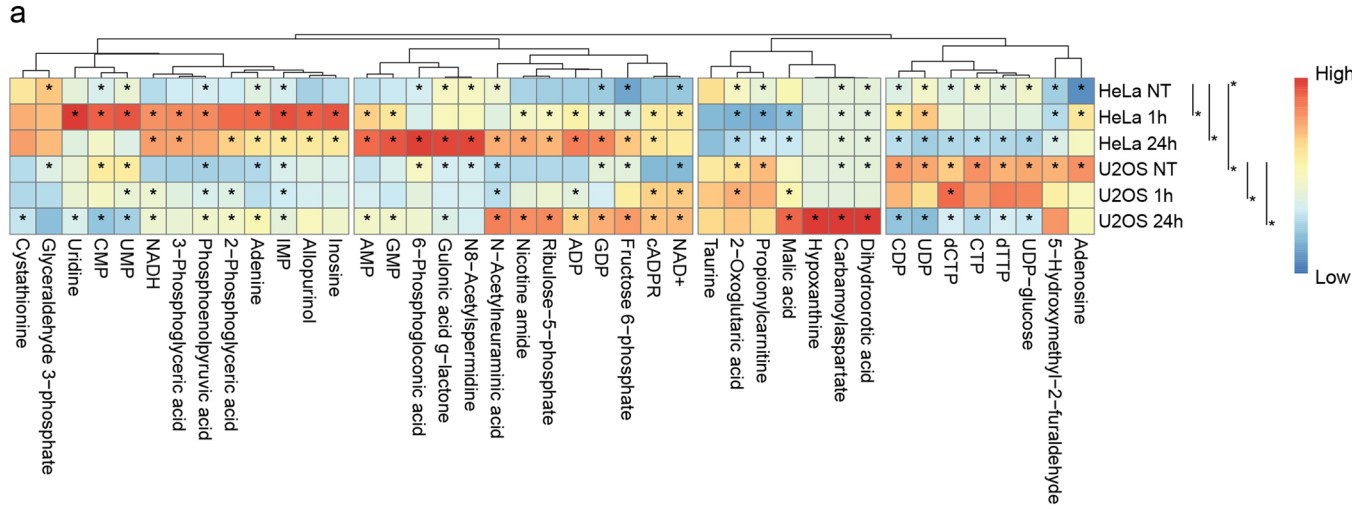

**Extended Data Fig. 5 | See next page for caption.**

**Extended Data Fig. 5 | Metabolomic analysis of cells treated with NAD⁺.**
**a**, Heatmap of metabolite levels (n = 4 biological replicates of individual samples collected from 4 different experiments) from HeLa and U2OS cells treated without (NT) or with 2 mM NAD⁺ as indicated. Asterisks for metabolites indicate significant differences between the indicated conditions (adjusted p-value < 0.05, FDR-adjusted p-values from two-tailed Student's t-test). Asterisks for metabolites for HeLa NT and U2OS NT indicate significant differences between the two untreated cell lines. **b**, Purine synthesis and metabolic pathways with metabolite levels (left) and gene expression levels (right) from metabolomics and RNA-seq analysis, respectively. HeLa and U2OS cells were treated without (NT) or with 2 mM NAD⁺ for the indicated time. In metabolite plots, the y-axis shows relative metabolite quantification, bars represent means, n = 4 biological replicates of individual samples collected from 4 different experiments. In gene expression plots, the y-axis shows normalized counts, n = 3 independent biological samples per condition sequenced once with multiplexing. Dashed lines indicate multiple synthesis steps. For boxplots in panel **b**, the center line indicates the median; box limits indicate the 25th and 75th percentiles; minima and maxima of whiskers extend 1.5 times the inter-quartile range from 25th and 75th percentiles, respectively. Numerical data are available in Source Data Files.

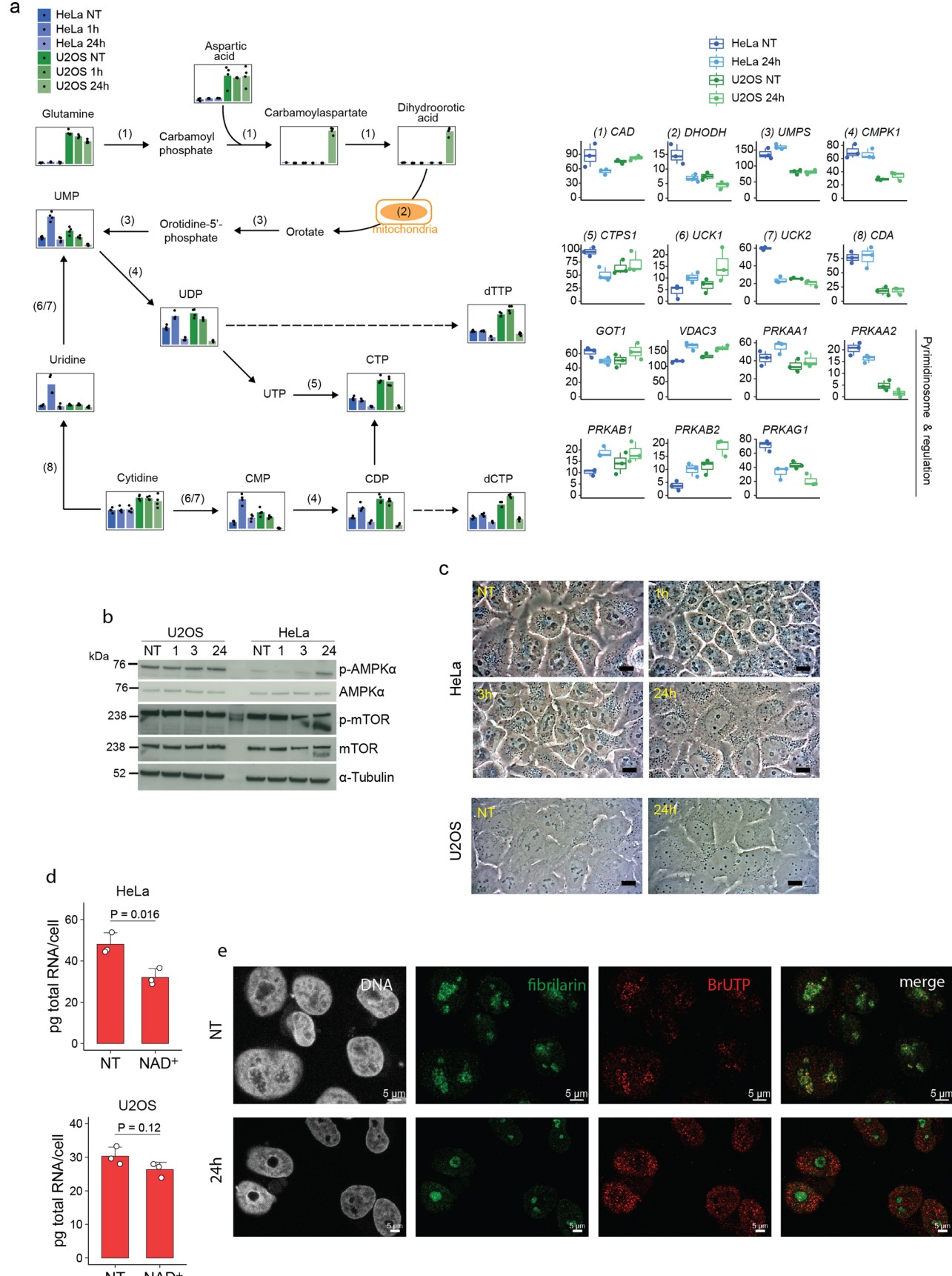

**Extended Data Fig. 6 | See next page for caption.**

**Extended Data Fig. 6 | Exogenous NAD⁺ depletes pyrimidine nucleotides.**
**a**, Pyrimidine synthesis pathways with metabolite levels (left) and gene
expression levels (right) from metabolomics and RNA-seq analysis, respectively.
HeLa and U2OS cells were treated without (NT) or with 2 mM NAD⁺ for the
indicated time. In metabolite plots, the y-axis shows relative metabolite
quantification, bars represent means, n = 4 biological replicates of individual
samples collected from 4 different experiments. In gene expression plots, the
y-axis shows normalized counts, n = 3 independent biological samples per
condition sequenced once with multiplexing. Dashed lines indicate multiple
synthesis steps. Expression of the pyrimidinosome enzymes (CAD, DHODH,
UMPS, GOT1 and VDAC3) and subunits of the modulator AMPK (PRKA- genes)
are included. **b**, Immunoblot from U2OS and HeLa cells treated without (NT) or
with 2 mM NAD⁺ for the indicated hours. The experiment was performed twice

with similar results. **c**, Representative bright-field microscopy images of HeLa
and U2OS cells treated without (NT) or with 2 mM NAD⁺ for the indicated time.
Scale bars are 10 μm. Representative images for two experiments performed with
similar results. **d**, Quantification of total RNA relative to cell count in HeLa and
U2OS cells treated for 24 h without (NT) or with 2 mM NAD⁺. Data are mean+SD,
n = 3 biological replicates, two-tailed Student's t-test. **e**, Representative
images of fibrillarin and BrUTP incorporation in HeLa cells treated without
(NT) or with 2 mM NAD⁺ for the indicated time. Representative images for two
experiments performed with similar results. Scale bars are 5 μm. For boxplots
in panel **a**, the center line indicates the median; box limits indicate the 25th and
75th percentiles; minima and maxima of whiskers extend 1.5 times the inter-
quartile range from 25th and 75th percentiles, respectively. Numerical data and
uncropped immunoblots are available in Source Data Files.

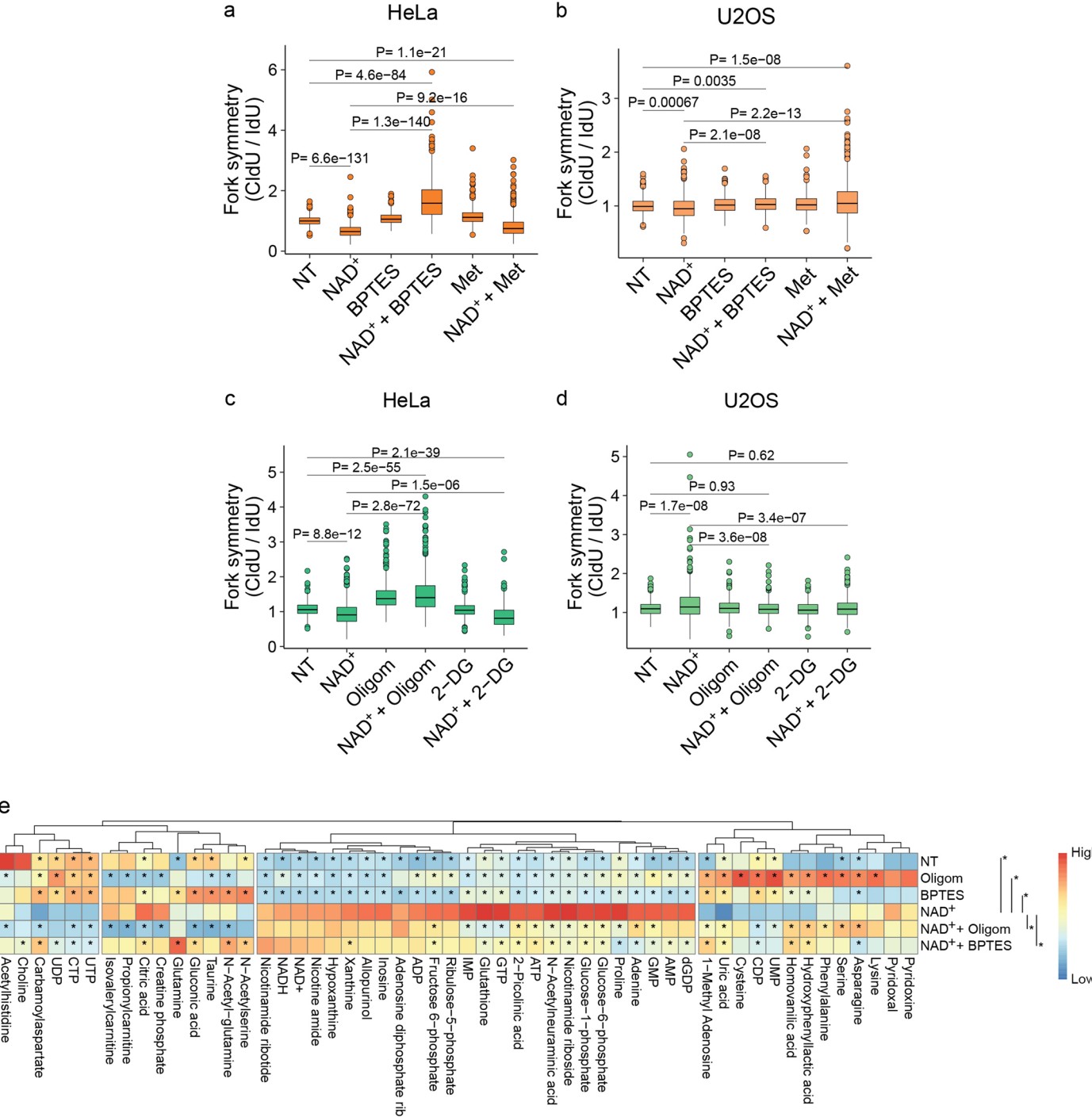

**Extended Data Fig. 7 | Metabolomics analysis of cells co-treated with NAD⁺, oligomycin and BPTES. a**, Symmetry of forks from Fig. 4c. **b**, Symmetry of forks from Fig. 4d. **c**, Symmetry of forks from Fig. 4e. **d**, Symmetry of forks from Fig. 4f. **e**, Heatmap of metabolite levels (n = 4 biological replicates of individual samples collected from 4 different experiments) from HeLa cells treated for 24 h without (NT) or with 5 µM oligomycin (Oligom), 10 µM BPTES, and/or 2 mM NAD⁺. Asterisks for metabolites indicate significant differences to NAD⁺-treated cells (adjusted p-value < 0.05, FDR-adjusted p-values from two-tailed Student's t-test). Replication forks were scored for two biological replicates, two-tailed Welch's t-test. For boxplots in panels **a,b,c,d**, the center line indicates the median; box limits indicate the 25th and 75th percentiles; minima and maxima of whiskers extend 1.5 times the inter-quartile range from 25th and 75th percentiles, respectively. Numerical data are available in Source Data Files.

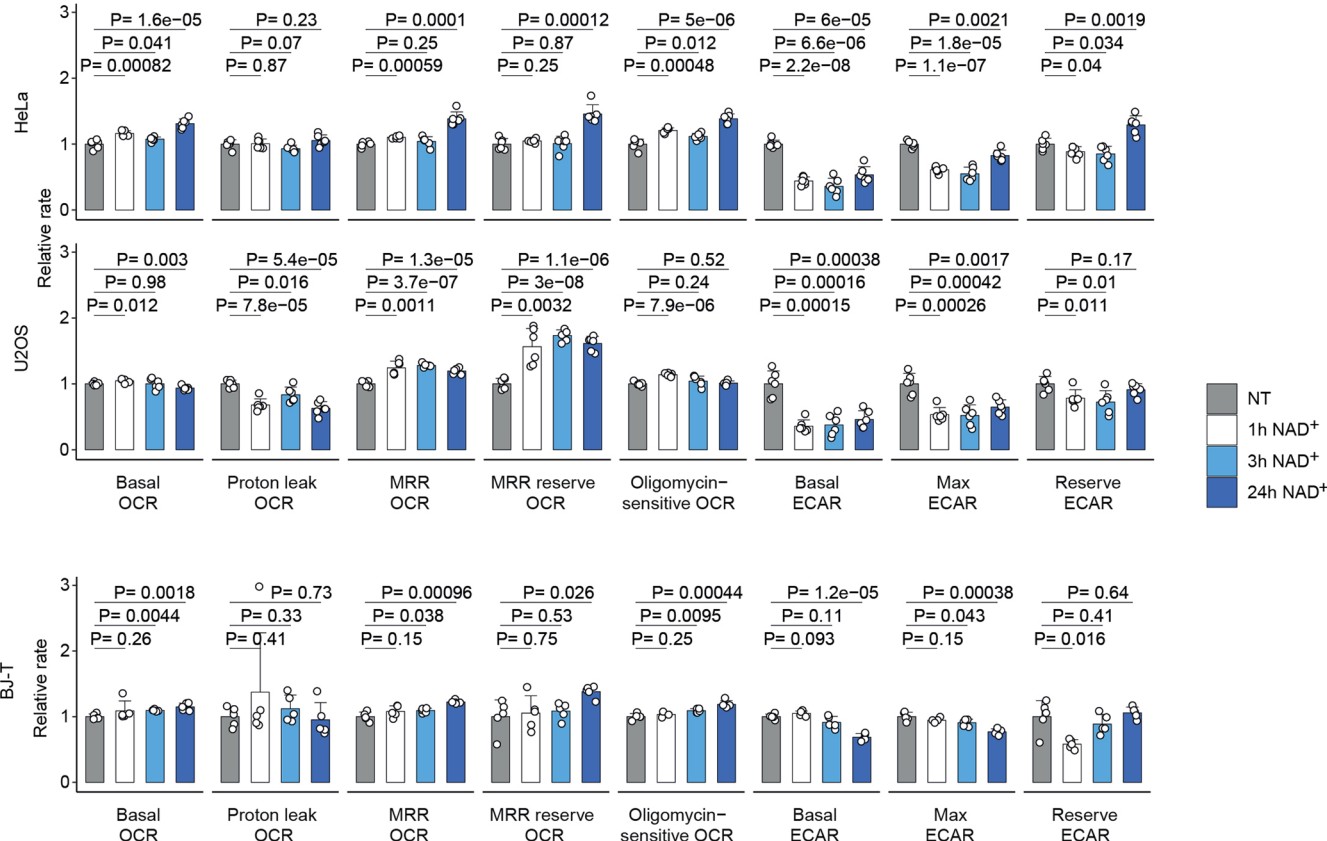

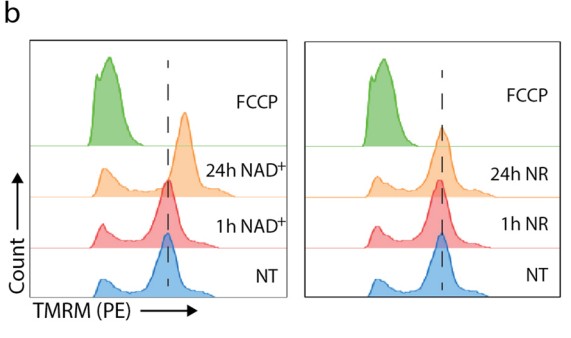

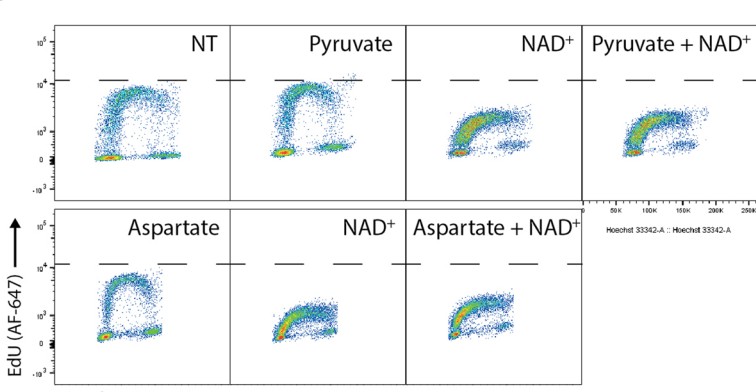

**Extended Data Fig. 8 | Mitochondrial activity of NAD⁺-treated cells. a**, All metabolic parameters from mitochondrial stress analysis relative to non-treated (NT) cells. The indicated cell lines were treated with 2 mM NAD⁺ for the indicated time. Data are mean+SD, n = 5 (BJ-T) and n = 6 (HeLa and U2OS) independent experiments, two-tailed Welch's t-test. OCR, oxygen consumption rate; MRR, maximum respiratory rate; ECAR, extracellular acidification rate. **b**, Flow cytometry analysis of mitochondrial membrane potential using the TMRM cell-permeant dye. HeLa cells were treated without (NT) or with 2 mM of NAD⁺ (left panel) or NR (right panel) for the indicated time, or with 10 µM FCCP for 30 min. The same data for NT and FCCP conditions are presented in both panels. **c**, Flow cytometry analysis of EdU incorporation and cell cycle in HeLa cells treated for 24 h without (NT) or with 5 mM pyruvate, 2 mM NAD⁺, and/or 20 mM aspartate. Flow cytometry experiments were performed twice with similar results. Numerical data are available in Source Data Files.

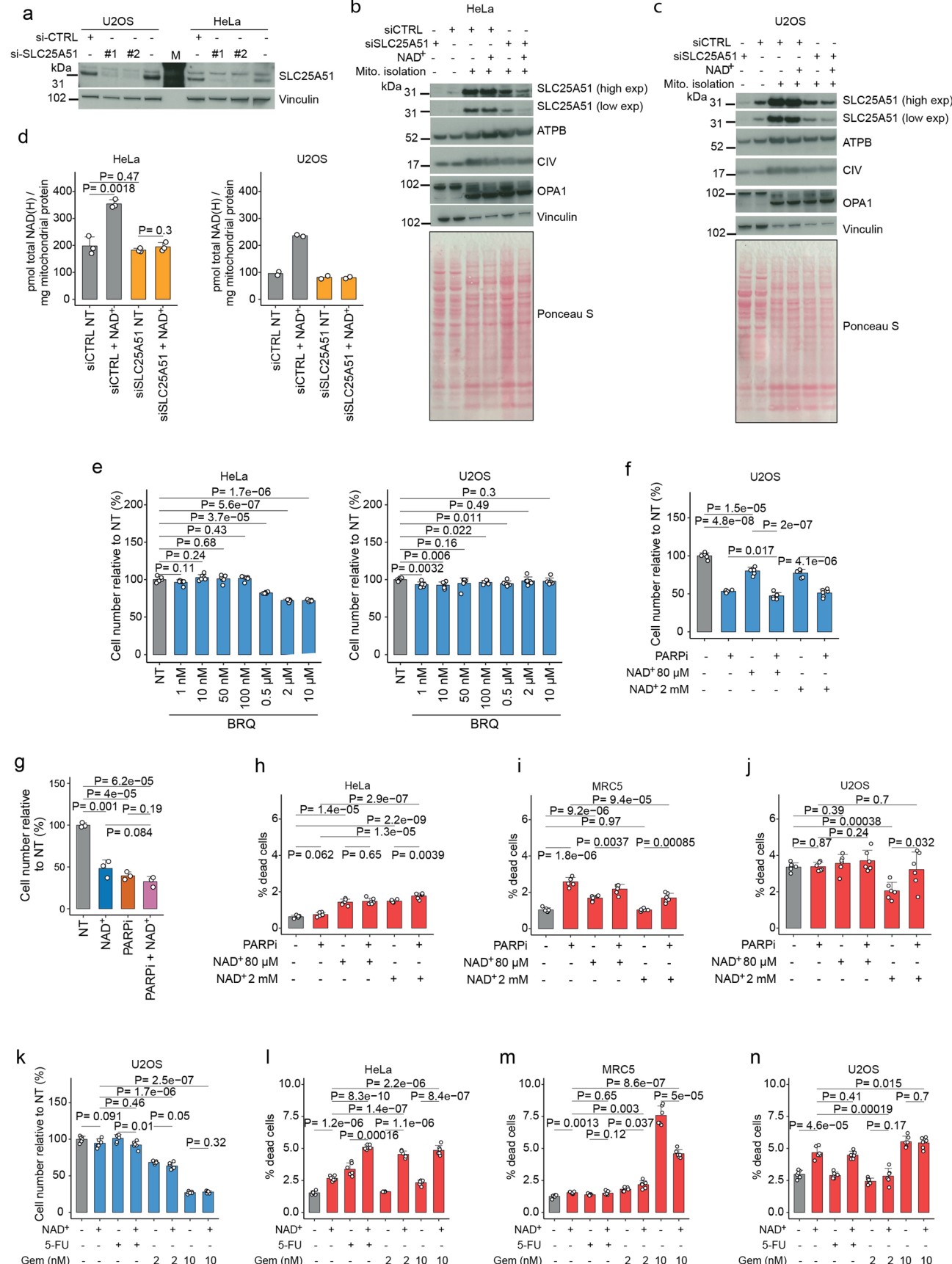

**Extended Data Fig. 9 | See next page for caption.**

**Extended Data Fig. 9 | Cell viability in combined treatments. a**, Immunoblot detection of SLC25A51 depletion in U2OS and HeLa cells. Cells were treated with control siRNA (si-CTRL) or one of two si-SLC25A51 (#1 and #2). M denotes lane with molecular weight marker. **b**, **c**, Immunoblot confirmation of mitochondrial (mito.) isolation from HeLa (**b**) and U2OS (**c**) cells treated for 24 h with control siRNA (si-CTRL) or SLC25A51-targeting siRNA and with 2 mM NAD$^+$ as indicated. **d**, Mitochondrial NAD(H) quantification in cells treated as in **b**, **c**, NT = non-treated. Data are mean+SD, n = 3 (HeLa) and n = 2 (U2OS) biological replicates, two-tailed Student's t-test (HeLa). **e**, Relative cell number of HeLa and U2OS cells treated for 24 h without (NT) or with brequinar (BRQ) as indicated. Data are mean+SD, n = 6 biological replicates, two-tailed Welch's t-test. **f**, Relative cell number of U2OS cells treated for 72 h without (NT) or with 1 μM PARPi and/ or NAD$^+$ as indicated. Data are mean+SD, n = 6 biological replicates, two-tailed Welch's t-test. **g**, Relative cell number of HeLa cells treated for 72 h without (NT) or with 2 mM NAD$^+$ and/or 10 μM PARPi. Data are mean+SD, n = 3 biological replicates, two-tailed Student's t-test. **h-j**, Percentage dead cells for HeLa (**h**), MRC5 (**i**) and U2OS (**j**) cells treated as in **f**. Data are mean+SD, n = 6 biological replicates, two-tailed Welch's t-test. **k**, Relative cell number of U2OS cells treated for 72 h without (NT) or with 80 μM NAD$^+$, 1 μM 5-fluorouracil (5-FU) and/or gemcitabine (Gem) as indicated. Data are mean+SD, n = 6 biological replicates, two-tailed Welch's t-test. **l-n**, Percentage dead cells for HeLa (**l**), MRC5 (**m**) and U2OS (**n**) cells treated as in **k**. Data are mean+SD, n = 6 biological replicates, two-tailed Welch's t-test. Immunoblots were performed twice with similar results. Numerical data and uncropped immunoblots are available in Source Data Files.

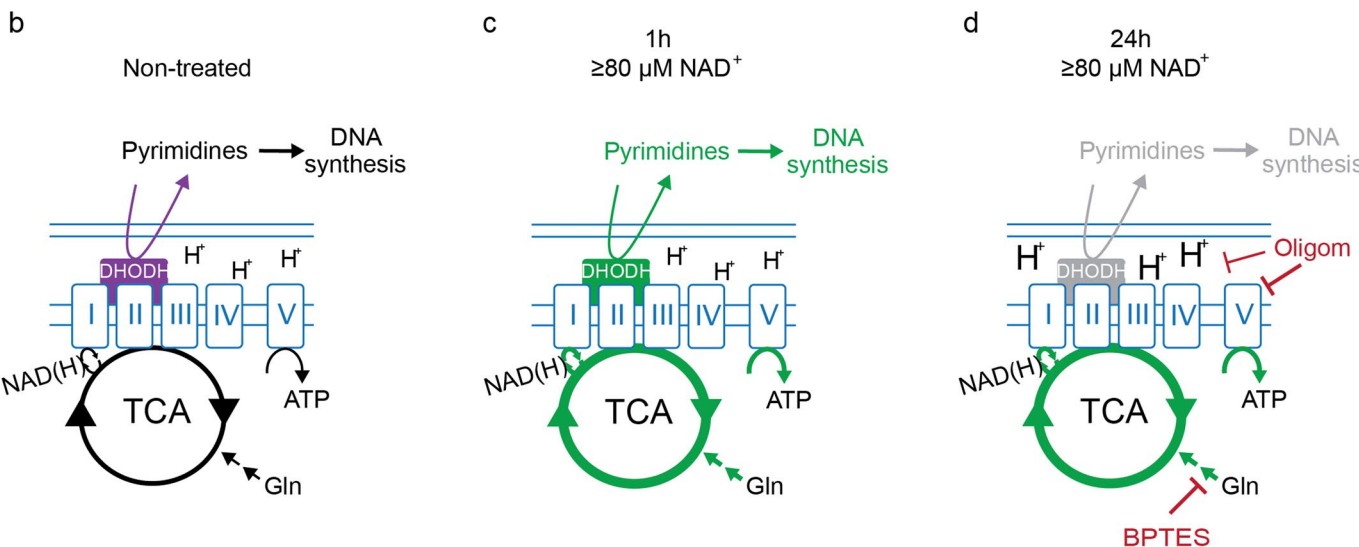

**Extended Data Fig. 10 | Proposed model of how NAD⁺ regulates pyrimidines.**
**a**, Pathways that maintain mitochondrial NAD(H) levels and metabolic activity.
Oaa, oxaloacetate; α-KG, α-ketoglutarate; Succ, succinate; Fum, fumarate;
Mal, malate; I-V, electron transport chain complexes. Illustration made with
BioRender. **b-d**, Temporal effects of NAD⁺ treatment. During non-treated
conditions, mitochondrial metabolic activity sustains ATP production and

pyrimidine synthesis for DNA replication (**b**). Short-term treatment with
NAD⁺ boosts cellular metabolism, nucleotide synthesis and DNA replication
(**c**). Extended NAD⁺ treatment results in increased mitochondrial membrane
potential, impaired pyrimidine biosynthesis, and depletion of pyrimidines for
DNA replication (**d**). Gln, glutamine; Oligom, oligomycin.

Apolinar Maya-Mendoza

# Reporting Summary

## Statistics

For all statistical analyses, confirm that the following items are present in the figure legend, table legend, main text, or Methods section.

| n/a | Confirmed | |
|---|---|---|
| ☐ | ☒ | The exact sample size (*n*) for each experimental group/condition, given as a discrete number and unit of measurement |
| ☐ | ☒ | A statement on whether measurements were taken from distinct samples or whether the same sample was measured repeatedly |
| ☐ | ☒ | The statistical test(s) used AND whether they are one- or two-sided *Only common tests should be described solely by name; describe more complex techniques in the Methods section.* |
| ☒ | ☐ | A description of all covariates tested |
| ☐ | ☒ | A description of any assumptions or corrections, such as tests of normality and adjustment for multiple comparisons |
| ☐ | ☒ | A full description of the statistical parameters including central tendency (e.g. means) or other basic estimates (e.g. regression coefficient) AND variation (e.g. standard deviation) or associated estimates of uncertainty (e.g. confidence intervals) |
| ☐ | ☒ | For null hypothesis testing, the test statistic (e.g. *F*, *t*, *r*) with confidence intervals, effect sizes, degrees of freedom and *P* value noted *Give P values as exact values whenever suitable.* |
| ☒ | ☐ | For Bayesian analysis, information on the choice of priors and Markov chain Monte Carlo settings |
| ☒ | ☐ | For hierarchical and complex designs, identification of the appropriate level for tests and full reporting of outcomes |
| ☒ | ☐ | Estimates of effect sizes (e.g. Cohen's *d*, Pearson's *r*), indicating how they were calculated |

*Our web collection on statistics for biologists contains articles on many of the points above.*

## Software and code

Policy information about availability of computer code

| Data collection | Zeiss ZEN (v3.2 blue edition) software, for LSM microscope image acquisition<br>Olympus scanR 3.3 software, for QIBC image acquisition<br>BD FACSuite V1.0.6 software, for flow cytometry data acquisition<br>MinKNOW (v20.06.4), for RNAseq data acquisition |
|---|---|
| Data analysis | ZEN lite (v2.6 blue edition) software, analysis of LSM-acquired micrographs<br>FlowJo_V10, flow cytometry analysis<br>R (v4.0.4), numerical and statistical analysis and final figure assembly<br>Guppy (v4.0.15, ONT), basecalling<br>Porechop (v0.2.4), demultiplexing and trimming for adapters<br>NanoFilt (v2.6.0), filtering<br>nanoQC (v0.9.4), quality inspections<br>NanoPlot (v1.29.0), quality inspections<br>Minimap2 (v2.17), genome alignment<br>Samtools (v1.9), alignment sorting, BAM conversion, and genome indexing<br>StringTie2 (v2.1.4), transcript assembly<br>GffCompare (v0.12.1), comparison and annotation of the assembled transcriptome<br>GffRead (v0.12.3), realignment to the assembled transcriptome<br>Salmon (v1.4.0), transcriptome alignments quantification<br>DESeq2 (v1.30.1), RNA-seq differential expression analysis<br>Tximport (v1.18.0), transcript count import and gene level summarization |

Pheatmap (v1.0.12), heatmap visualizations
Compound Discoverer (3.1), metabolomics compound identification
STRING (v11-0b, webtool), functional enrichment analysis
ScanR analysis software (v3.2.0), analysis of data from ScanR QIBC acquisition

For manuscripts utilizing custom algorithms or software that are central to the research but not yet described in published literature, software must be made available to editors and reviewers. We strongly encourage code deposition in a community repository (e.g. GitHub). See the Nature Portfolio guidelines for submitting code & software for further information.

## Data

Policy information about availability of data

All manuscripts must include a data availability statement. This statement should provide the following information, where applicable:
- Accession codes, unique identifiers, or web links for publicly available datasets
- A description of any restrictions on data availability
- For clinical datasets or third party data, please ensure that the statement adheres to our policy

Data availability
RNA–seq data reported in this study have been deposited in the European Nucleotide Archive (ENA) with accession PRJEB64552. Differential expression analyses are provided in Source Data Files. Metabolomics data are provided as Supplementary Tables. Other numerical source data have been provided in Source Data Files. All other data supporting the findings of this study are available from the corresponding author on request.

## Human research participants

Policy information about studies involving human research participants and Sex and Gender in Research.

| | |
|---|---|
| Reporting on sex and gender | This study does not involve human research participants. |
| Population characteristics | N/A |
| Recruitment | N/A |
| Ethics oversight | N/A |

Note that full information on the approval of the study protocol must also be provided in the manuscript.

# Field-specific reporting

Please select the one below that is the best fit for your research. If you are not sure, read the appropriate sections before making your selection.

☒ Life sciences ☐ Behavioural & social sciences ☐ Ecological, evolutionary & environmental sciences

For a reference copy of the document with all sections, see nature.com/documents/nr-reporting-summary-flat.pdf

# Life sciences study design

All studies must disclose on these points even when the disclosure is negative.

| | |
|---|---|
| Sample size | No sample-size calculation were performed. The sample size was determined to be sufficient, taking into account the substantial and consistent differences observed between the groups. For DNA fiber experiments, if equivalent experiments were not different statistically (p value of t-test <0.001), the total number of DNA fibers from all experiments is shown. Data reported from high-content microscopy experiments are from 2. 3 or 4 biological replicates as are indicated in the figure legends. |
| Data exclusions | No exclusion was applied. |
| Replication | Reported results were tested and confirmed in at least two independent experiments (i.e. minimum two biological replicates). |
| Randomization | Our experimental designs did not permit randomization to be applied. |
| Blinding | Wherever possible, the investigators were blinded for data acquisition. Samples were labeled only by sequential numbers during their preparation before and, in the majority of cases, during data collection, image acquisition and metabolite analysis. |

# Reporting for specific materials, systems and methods

We require information from authors about some types of materials, experimental systems and methods used in many studies. Here, indicate whether each material, system or method listed is relevant to your study. If you are not sure if a list item applies to your research, read the appropriate section before selecting a response.

## Materials & experimental systems

| n/a | Involved in the study |
|-----|----------------------|
| ☐ | ☒ Antibodies |
| ☐ | ☒ Eukaryotic cell lines |
| ☒ | ☐ Palaeontology and archaeology |
| ☒ | ☐ Animals and other organisms |
| ☒ | ☐ Clinical data |
| ☒ | ☐ Dual use research of concern |

## Methods

| n/a | Involved in the study |
|-----|----------------------|
| ☒ | ☐ ChIP-seq |
| ☐ | ☒ Flow cytometry |
| ☒ | ☐ MRI-based neuroimaging |

## Antibodies

| Antibodies used | Rat anti-BrdU antibody, Serotec, OBT0030 (diluted 1:500)<br>Mouse anti-BrdU antibody, Becton Dickinson, 347580 (diluted 1:500)<br>DyLight 550, Thermo Fisher Scientific, SA5-10019 (diluted 1:400)<br>Alexa Fluor 488, Invitrogen, A21202 (DNA fibres, diluted 1:400)<br>Alexa Fluor 488, Invitrogen, A11029 (QIBC, diluted 1:1,000)<br>Fibrillarin, Abcam, ab5821 (diluted 1:500)<br>p-CHK1 S317, Cell Signaling, 2344 (diluted 1:250)<br>CHK1, Santa Cruz, sc-8408 (diluted 1:100)<br>p-CHK2 T68, Abcam, ab32148 (diluted 1:200)<br>CHK2, Abcam, ab109413 (diluted 1:20,000)<br>p-H2A.X S139 (γH2AX), Abcam, ab22551 (diluted 1:500)<br>β-Actin, Sigma, A1978 (diluted 1:5,000)<br>p-AMPKα T172, Cell Signaling, 2535 (diluted 1:500)<br>AMPKα, Cell Signaling, 2603 (diluted 1:500)<br>p-mTOR S2481, Cell Signaling, 2974 (diluted 1:250)<br>mTOR, Cell Signaling, 2972 (diluted 1:250)<br>α-Tubulin, GeneTex, GTX628802 (diluted 1:5,000)<br>SLC25A51/MCART1, Abcam, ab237054/CUSABIO CSB-PA875649LA01HU (diluted 1:500)<br>Vinculin, Sigma, V9131 (diluted 1:50,000-100,000)<br>VDAC, Cell Signaling, 4661 (diluted 1:500)<br>COX IV/CIV, Cell Signaling, 11967 (diluted 1:500)<br>OPA1, Abcam, ab157457 (diluted 1:1,000)<br>ATPB, Abcam, ab14730 (diluted 1:50)<br>Horseradish peroxidase-conjugated anti-rabbit secondary antibody (PI-1000, Vector Laboratories, diluted 1:10,000)<br>Horseradish peroxidase-conjugated anti-mouse secondary antibody (PI-2000, Vector Laboratories, diluted 1:10,000) |
|---|---|
| Validation | Rat anti-BrdU antibody, validated by the company and cited in 284 research publications<br>Mouse anti-BrdU antibody, monoclonal antibody originally described in Science 1982; 218:474, validated by the company and cited in 1036 research publications<br>DyLight 550, validated by the company<br>Alexa Fluor 488 (Invitrogen, A21202), validated by the company<br>Alexa Fluor 488 (Invitrogen, A11029), validated by the company<br>Fibrillarin, validated by the company and cited in 158 research publications<br>p-CHK1 S317, validated by the company and cited in 243 research publications<br>CHK1, validated by the company and cited in 789 research publications<br>p-CHK2 T68, validated by the company and cited in 19 research publications<br>CHK2, knockout validated by the company and cited in 17 research publications<br>p-H2A.X S139 (γH2AX), validated by the company and cited in 131 research publications<br>β-Actin, validated by the company and cited in nearly 4000 research publications<br>p-AMPKα T172, validated by the company and cited in 2954 research publications<br>AMPKα, validated by the company and cited in 428 research publications<br>p-mTOR S2481, validated by the company and cited in 376 research publications<br>mTOR, validated by the company and cited in 1605 research publications<br>α-Tubulin, validated by the company and cited in 122 research publications<br>SLC25A51/MCART1, validated by the company and cited in Sci. Adv. 2020; 6 : eabe5310.<br>Vinculin, validated by the company (enhanced-validation strategy) and cited in 1937 research publications<br>VDAC, validated by the company and cited in 298 research publications<br>COX IV, validated by the company and cited in 124 research publications<br>OPA1, validated by the company and cited in 52 research publications<br>ATPB, validated by the company and cited in 229 research publications<br>Horseradish peroxidase-conjugated anti-rabbit secondary antibody, validated by the company and cited in 659 research publications<br>Horseradish peroxidase-conjugated anti-mouse secondary antibody, validated by the company and cited in 428 research publications |

# Eukaryotic cell lines

Policy information about cell lines and Sex and Gender in Research

| | |
|---|---|
| Cell line source(s) | U2OS osteosarcoma (ATCC HTB-96, female), HeLa cervical carcinoma (ATCC CCL-2, female), BJ normal skin fibroblasts (ATCC CRL-2522, male), MRC5 normal lung fibroblasts (ATCC CCL-171), MEFs (ATCC CRL-2991), BJ 5ta hTERT-immortalized skin fibroblasts (ATCC CRL-4001, male), T98G glioblastoma multiforme (ATCC CRL-1690, male) and U87 MG likely glioblastoma multiforme (ATCC HTB-14, male) from ATCC. |
| Authentication | All cell lines were purchased from ATCC and the catalog numbers have been added in Methods. From the purchased date, cells have not been authenticated . |
| Mycoplasma contamination | All cell lines used in this study were regularly tested by PCR for mycoplasma contamination and were negative. |
| Commonly misidentified lines (See ICLAC register) | No commonly misidentified cell lines were used in this study. |

# Flow Cytometry

## Plots

Confirm that:

☒ The axis labels state the marker and fluorochrome used (e.g. CD4-FITC).

☒ The axis scales are clearly visible. Include numbers along axes only for bottom left plot of group (a 'group' is an analysis of identical markers).

☒ All plots are contour plots with outliers or pseudocolor plots.

☒ A numerical value for number of cells or percentage (with statistics) is provided.

## Methodology

| | |
|---|---|
| Sample preparation | Cells were incubated with EdU for 30 min at 37 degrees C, trypsinized, washed with PBS and fixed with 70% ice-cold ethanol. Samples were stored at -20 degrees C before EdU was detected using the Click-iT EdU Alexa Fluor 647 Imaging Kit according to manufacturer's instructions. After washing with PBS(+), cells were stained with Hoechst 33342, washed again with PBS(+), resuspended in PBS and analysed on the flow cytometer. |
| Instrument | FACSVerse (Becton Dickinson) |
| Software | Data were collected using BD FACSuite V1.0.6 software and analysed using FlowJo_V10 software. |
| Cell population abundance | Full cell populations were analysed without post-sort fractions. |
| Gating strategy | All flow cytometry experiments were gated and analysed similarly. The same gates were used for the control and experimental conditions. Cell debris were gated out (FSC/SSC values below 50K and above 200K were considered cell debris). |

☒ Tick this box to confirm that a figure exemplifying the gating strategy is provided in the Supplementary Information.

