## [Peer Review File · Nature Cell Biology]

Peer Review Information

Journal: Nature Cell Biology

Manuscript Title: NAD⁺ regulates nucleotide metabolism and genomic DNA replication

Corresponding author name(s): Jiri Bartek, Apolinar Maya-Mendoza

Reviewer Comments & Decisions:

Decision Letter, initial version:
--

Message: Dear Professor Bartek,

Thank you for submitting your manuscript "NAD⁺ regulates nucleotide levels and genomic DNA replication", to Nature Cell Biology. It has now been seen by 4 referees, who are experts in nucleotide metabolism (Referee #1); DNA replication stress, PARP (Referee #2); NAD⁺, aging (Referee #3); and NAD⁺ homeostasis (Referee #4), and whose comments are pasted below. Thank you very much for your patience with the peer review process. In light of their advice, we regret that we cannot offer to publish the study in Nature Cell Biology.

As you will see, although the reviewers found this work interesting, they raised serious concerns that question the conceptual advance that these findings represent over previous work (Rev#1, Rev#3), and the strength of the data and of the novel conclusions that can be drawn at this stage. The reviewers shared doubts about the data interpretation, models, and approaches (all reviewers), the use of high exogenous doses of NAD⁺ (Rev#1, Rev#2), and the depth of mechanistic understanding throughout the studies (Rev#2, Rev#3).

We have discussed these points within the editorial team in detail and unfortunately, we find that these concerns are too significant and preclude further consideration at the journal.

We are very sorry that we could not be more positive on this occasion, but we thank you for the opportunity to consider this work.

With kind regards,
Melina

Melina Casadio, PhD
Senior Editor, Nature Cell Biology
ORCID ID: <https://orcid.org/0000-0003-2389-2243>

Reviewers' comments:

Reviewer #1 (Remarks to the Author):

In this manuscript, Munk et al., set out to define how NAD levels impact metabolism and DNA damage response (DDR). The authors show that supplementation of cells with high levels of exogenous NAD results in an increase in ATP levels and DNA synthesis short term, while sustained high levels of NAD reduce cell proliferation, possibly due to unbalanced nucleotide pools. While the authors have performed a comprehensive analysis of the metabolic outputs and DDR in response to exogenous NAD supplementation, the latter represents a major flaw in this study given that NAD is not cell-permeable, and to date there are no known NAD transporters at the plasma membrane. Therefore, it becomes difficult to interpret the results not knowing how NAD regulates NAD or ATP levels, or nucleotide imbalance. Also, the manuscript does not present a major advance in the field due to published work illustrating links between nucleotide imbalance or NAD levels and replication stress (PMIDs: 35927450; 34806016

Major comments:

- 1) The concentrations of NAD used throughout the study are supraphysiological. Moreover, there are no known NAD transporters at the plasma membrane, raising the question of whether NAD signals through the purinergic receptors, similar to other purine nucleotides. Does exogenous NAD enter the cells after being converted to NMN or NR?
- 2) Fig. 1: NR or NAD⁺ treatment increases cellular NAD(H) level, but only NAD⁺ treatment induces DDR response and inhibits DNA synthesis. Why is this response NAD specific? Are purinergic receptors or CD38 involved in metabolizing NAD and inducing downstream signaling pathways (Ca⁺ or C-ADPR)?
- 3) How do NAD⁺ levels increase the levels of purines, but decrease pyrimidines (Fig 2)? Since the metabolite patterns between HeLa and U2OS are different, more cellular contexts are required to understand whether these phenotypes are generalizable.
- 4) Line 253: How does NAD⁺ regulate the activity of the TCA cycle and ETC to regulate DNA replication? In extended fig 3b, the TCA cycle metabolites fumarate and malate decrease in HeLa cells upon NAD treatment. There is not sufficient evidence to conclude that NAD regulates the DNA synthesis through "the NAD⁺ mediated-high activity of TCA cycle and ETC"
- 5) Line 254: Extend fig 7.g, what is the evidence for the conclusions that high AMP level inhibits pyrimidine synthesis?
- 6) Line: 288-290: The presented data are not sufficient to conclude that the "effects of NAD⁺ depend on mitochondrial NAD⁺ transport in U2OS cells, while HeLa cells remain sensitive to NAD⁺ in the absence of SLC25A51 owing to AMP-regulated mechanisms". At the very least, mitochondrial NAD(H) levels should be measured as lack of mitochondrial NAD⁺ transporter may result in the inhibition of DNA synthesis independently of its transport functions. Moreover, the presented data is not compelling to suggest an AMP-regulated mechanism.
- 7) The biological or physiological relevance of high exogenous NAD levels is unclear, given that plasma levels of NAD are in micromolar rather than millimolar range. Does supplementation of high levels of NAD in vivo have similar effects as presented in the two cell culture models?

Reviewer #2 (Remarks to the Author):

In this manuscript, the authors investigated how NAD⁺ levels affect DNA synthesis and genomic stability. They found that exogenous NAD⁺ inhibits DNA synthesis and induces DNA damage. By metabolic profiling, they showed that exogenous NAD⁺ induces an imbalance of nucleotides, reduces overall transcription, and activates mTOR. Their RNA-seq analysis of NAD⁺ treated cells uncovered that ETC activity is enhanced by NAD⁺ in mitochondria. Blocking ETC by oligomycin rescued the effects of NAD⁺ on DNA synthesis. Inhibition of the conversion of glutamine to glutamate and the TCA cycle by BPTES also rescued the effects of NAD⁺ on DNA synthesis. These results suggest that the TCA cycle and ETC are important for the effects of NAD⁺ on DNA synthesis, which may be mediated by unbalanced nucleotides. The authors also noticed differences between HeLa and U2OS cells in their responses to exogenous NAD⁺. HeLa cells accumulate higher levels of AMP after NAD⁺ treatment, which may also contribute to the inhibition of DNA synthesis. Finally, the authors proposed that NAD⁺ can be combined with drugs that alter pyrimidine levels in cancer therapy.

The main finding of this study that high NAD⁺ can alter nucleotide balance and inhibit DNA synthesis is interesting. However, almost of the experiments supporting this conclusion are based on the use of a high concentration of exogenous NAD⁺. Whether this approach is relevant to any physiological or pathological conditions in aging, cancer, and therapy is unclear. Whether NAD⁺ induced changes in nucleotide balance are the actual cause of reduced DNA synthesis and induction of DNA damage is not tested directly. Whether and how enhanced ETC is responsible for the unbalanced nucleotides is also untested. The overall model of this study is still too preliminary, with major gaps to be filled. Although this study is focused on the effects of NAD⁺ on DNA synthesis and genomic stability, the metabolomics data from this study show that NAD⁺ affects many other pathways and processes. It seems to me that the impacts of NAD⁺ in cells are much more complex than what is proposed in the model, and I am not sure if this study can establish general principles as claimed.

1. In Fig. 1b and 1c, the use of SIRT1i and PARPi to study the effects of increased NAD⁺ is not completely justified. Both SIRT1 and PARP1 have many functions independent of their effects on NAD(H) levels. In Fig. 1b, it is clear that SIRT1i and PARPi only modestly increased NAD(H) levels. However, it is unclear whether the modest increase of NAD(H) in PARPi treated cells is relevant to the effects of PARPi on DNA synthesis and genomic stability.

2. In Fig. 1b, NAMPTi drastically reduced NAD(H) in both HeLa and U2OS cells, but it only reduced DNA synthesis in HeLa cells. This difference between HeLa and U2OS is never explained in the following experiments. The 1.5 fold higher NAD⁺ in U2OS is unlikely sufficient to explain the difference. The authors completely switched to using exogenous NAD⁺ to inhibit DNA replication, which is a completely different phenomenon.

3. The NAD⁺ rescue experiments in Fig. 1d and 1e are confusing. NAD⁺ partially rescued the effects of NAMPTi in HeLa, but it acted independently of NAMPTi in U2OS cells. Since NAMPTi reduced endogenous NAD(H) in U2OS (Fig. 1b), should the effects of exogenous NAD⁺ be at least reduced by NAMPT1i?

4. Adding 2mM exogenous NAD⁺ to cells is a very artificial manipulation. It is not clear whether this high concentration of NAD⁺ is relevant to any physiological or pathological conditions in aging, cancer, or therapy. The effects of this extremely high concentration of exogenous NAD⁺ may or may not be achievable in cells under any physiological or pathological conditions. This is a major concern on the experimental approach of the current study. The authors should confirm their findings using other strategies to increase endogenous NAD⁺.
5. The authors did not interpret Fig. 1m clearly. Even if high levels of NAD⁺ do not affect RRM activity, why would addition of dNTPs reduce DNA synthesis? Can this be explained by unbalanced nucleotide pools? Why?
6. In Fig. 2, the authors found that 24 h of NAD⁺ treatment reduced pyrimidines but increased purines, among many other metabolic changes. They speculated that NAD⁺ induced imbalance of nucleotides is the cause of reduced DNA synthesis and genomic instability. However, this idea is never directly tested. This is a major weakness of the current study.
7. In Fig. 3i and 3j, the rescuing effects of oligomycin on NAD⁺ induced fork slowing is impressive. Can the authors confirm these effects by directly disrupting ETC?
8. The effects of FCCP are confusing. Since FCCP should disrupt ATP synthesis, should it also rescue fork speed? Why would FCCP alone reduce DNA synthesis in Fig. 3k? Why are the effects of FCCP different from Oligomycin?
9. At the end of Fig. 3, the authors speculated that ETC is important for the maintenance of balanced nucleotide pools. Again, this idea is based on correlations of changes and is never tested directly. This is another major weakness of the current study.
10. In Fig. 4c, the rescuing effects of BPTES on fork speed are clear. If BPTES affects TCA and indirectly affects ETC, it should have similar effects on mitochondrial membrane potential as Oligomycin and FCCP. The authors should test this directly.
11. The point on AMP in Fig. 4 is confusing. The authors tried to explain why HeLa cells are sensitive to NAD⁺ than U2OS by showing that HeLa cells accumulate more AMP and AMP inhibits DNA synthesis. However, even in HeLa cells, BPTES fully rescued DNA synthesis in the presence of exogenous NAD⁺ (Fig. 4c), even though fork speed was only partially rescued (Fig. 4a). Does this mean that the effects of endogenous AMP in HeLa cells are not detected by overall EdU incorporation? The effects of exogenous AMP in Fig. 4d could be artificial.
12. In Fig. 5a, it seems that SLC25A51 does not have any impact on the NAD⁺ induced replication inhibition in HeLa cells. This is not consistent with the other results that ETC has a major contribution to this NAD⁺ effect. AMP should be only partially responsible for the effect.
13. The interpretations of Fig. 5e and 5f are confusing. The effects of PARPi and NAD⁺ are clearly distinct (Fig. 5f) and additive (Fig. 5e). This argues that PARPi and NAD⁺ are not epistatic. The results of these experiments actually suggest that the effects of PARPi are independent of its impact on NAD(H).

14. The authors idea that NAD⁺ precursors are safer because they affect mitochondria more slowly does not make sense. They should analyze the effects of these precursors on RS and DDR at later time points.

Reviewer #3 (Remarks to the Author):

Apolinar Maya-Mendoza's group and colleagues presented a research article entitled 'NAD⁺ regulates nucleotide levels and genomic DNA replication'. By the application of different cell lines (Human U2OS, HeLa, BJ, BJ-5ta (BJ-tert), T98G, and U87 MG cell lines) and a series of biochemical and omics approaches, the authors manipulated cellular NAD⁺ levels and investigated changes in metabolism and genomic DNA replication. The authors presented a large amount of well-organized data which could be informative to the NAD⁺ and DNA repair fields. However, my enthusiasm was dramatically dampened due to the below major concerns:

- Low novelty on much of the data. While impressive amount of data are presented in this paper, a significant amount of 'discoveries' claimed in this paper are not new. E.g., the statement 'Here we show that high levels of NAD⁺ rapidly boost Krebs's cycle, increase the ATP level, and accelerate the speed of genomic DNA synthesis' is not new as there are mountains of data in the fields from last century enabled the writings in our text book that NAD⁺ is essential for glycolysis, TCA cycle, and OXPHOS (thus more ATP) (see review summaries from the Guarente, Auwerx and Bohr groups). The importance of NAD⁺ in DNA replication was evidenced from 1960s and has been well-studied onwards (PMID: 4310516).
- Much of the data are descriptive, lacking in-depth mechanistic studies. E.g, the authors have noticed that NAD⁺ regulates nucleotide levels, but the underlying molecular mechanisms are elusive. One possibility could be a feed-forward loop between NAD⁺ and amino acids generation via autophagy (PMIDs: 18296641; PMID: 24813611); a recent paper also showed that autophagy regulates NAD⁺ (PMID: 36413951).
- When working on metabolism (especially mitochondrial metabolism) and DNA replication, it is necessary to distinguish cancer-like-immortalized cell system and the normal cellular system. The majority of the data of this paper were from cancer (cancer-like) lines (although a small portion of data were repeated in primary cells), one should be extremely cautious in data interpretation. The used cell lines are majorly cancer cells: U2OS is from a moderately differentiated sarcoma, HeLa is from cervical cancer cells, while U87 MG is original from malignant gliomas. BJ (CRL-2522) is from skin taken from normal foreskin from a neonatal male. This cell line, although from a normal individual, shows signs of 'immortalization' (and likely cancer-like feature), including telomerase negative (although could lead to accelerated ageing) and 'the capacity to proliferate to a maximum of 72 population doublings before the onset of senescence' (<https://www.atcc.org/products/crl-2522>); for normal human fibroblasts, the average passages numbers are around 30-40 when entering to senescence (see papers from Campisi lab etc). Although BJ-tert cell line is used, it is important to use multiple primary cell lines to validate the data from cancer cell lines, if the authors want to generalize their findings to the normal cells.

Collectively, the authors are applauded for a large amount of informative data generated. This paper may fit better for a cancer journal. In view of the importance of NAD⁺ in proliferation and chemotherapy-induced resilience, data from this paper may be more

interested to cancer researchers.

Reviewer #4 (Remarks to the Author):

NAD⁺ regulated nucleotide and genomic DNA replication

Manuscript #: NCB_LE50081

Summary:

Munk et. al., aimed to address if the levels of NAD(H) had an impact on DNA damage response and/or genomic DNA synthesis. These processes are critical for maintaining homeostasis and genomic integrity. Most importantly, NAD⁺ is a key redox cofactor that serves as a key substrate for PARP enzymes, which regulate genomic DNA synthesis. The authors were able to show that increasing NAD⁺ concentrations led to increased mitochondrial activity and DNA synthesis. However, prolonged NAD⁺ elevation triggered a reduction in pyrimidine biosynthesis and cell cycle arrest. This observation is critical as we continue to understand how to utilize NAD⁺ supplements for various aged-related diseases, and how increase NAD⁺ can have negative feedback on organismal homeostasis. The authors proposed a model that showcases and reconcile how NAD⁺ controls mitochondrial function, DNA replication, and cell proliferation—with the hope of explaining how NAD⁺ metabolism modulates basal metabolism and DNA synthesis.

Comments/Questions:

- Key literature is not cited and discussed in the current state of the manuscript. This is misleading to the audiences who will read and use this work as a resource. Please be sure to include relevant citations that span across the field.
- Figures throughout the paper can be labeled better and more properly to help the reader follow along with the authors. Recommend adding titles to each figure for feasibility.
- For the global metabolomics—the authors should preform FDR correction on their results to ensure no false discoveries are identified
- One limitation is that this paper only represents cell lines. It's understood that NAD⁺ metabolism drastically changes between in vitro and in vivo conditions (PMID: 29685734). It would be of great interest to test this hypothesis/model in vivo to confirm or discuss the limitations in the discussion if unable to perform this key experiment.
- The authors are relying on NAD/NADH assays kit measurements to quantify NAD⁺. This is extremely limiting for their observations, their findings, and many additional reasons. Combining liquid chromatography and mass spectrometry is key for accurately and precisely quantitating NAD(H) levels. The conclusions are too strong for using a kit and not LCMS.

In summary, the authors have revealed potential mechanisms that link NAD to modulating basal metabolism and DNA synthesis. However, prolonged NAD⁺ elevation does trigger negative downstream events. Interestingly, the authors propose how NAD-supplements could indeed be a safer option for increasing intracellular NAD⁺ without the negative impacts (triggering DDR, etc.).

**Although we cannot publish your paper, it may be appropriate for another journal in the Nature Portfolio. If you wish to explore the journals and transfer your manuscript please use our manuscript transfer portal. You will not have to re-supply manuscript metadata and files, but please note that this link can only be used once and remains active until used. For more information, please see our manuscript transfer FAQ page.

Note that any decision to opt in to In Review at the original journal is not sent to the receiving journal on transfer. You can opt in to In Review at receiving journals that support this service by choosing to modify your manuscript on transfer. In Review is available for primary research manuscript types only.

**For Nature Portfolio general information and news for authors, see <http://npg.nature.com/authors>.

Author Rebuttal to Initial comments

Responses to Reviewers' comments:

Before providing our specific point-by-point responses to the comments raised by each of the four reviewers, we wish to express our appreciation for the time and effort that you used to provide your expert opinion and guidance for our further work on this study. As you will see from the revised version of our manuscript and this rebuttal document, we have addressed every point raised, in the majority of cases by new sets of experiments inspired by the constructive criticisms and suggestions from the reviewers. Given their complementary expertise and the fact that four reviewers were consulted, it is natural that a large number of comments needed to be addressed, and despite we have also involved some new authors to help revise the manuscript, the revision has taken us some 6 months to complete. The vast majority of your comments were fully justified and very helpful to guide our new experimental efforts. As a result, we very much hope that our answers and the new data presented in this revised version of the manuscript, along with the explanation of a few misunderstandings mainly reflecting our original insufficient formulations of rationale or interpretation of some results or published literature, due to the space restrictions of the letter format, will convince the reviewers of the novelty and conclusive results, that we believe much better support and extend our original conclusions and model. Based on our own long-term experience with reviewing for NCB, Nature and other top journals, we do realize such review efforts take a lot of valuable time, so we hope the improved dataset provided in this revised manuscript will be of interest and maybe even provide some inspiration for your own research.

Among the new results and changes included in our revised manuscript in response to the reviewers' comments are the following (just to list a few important ones):

- Data on a range of NAD⁺ concentrations to show that also the much lower, more physiologically relevant concentrations recapitulate the effects presented in the original manuscript;
- Substantially deeper mechanistic insights including the role of NAD⁺ in modulation of nucleotide metabolism and activity of Dihydroorotate dehydrogenase, the key enzyme that links mitochondrial bioenergetics and pyrimidine synthesis;
- An extended spectrum of cell lines to further support the concept;
- Better insights into how the mechanism we describe here affects the fundamental process of DNA replication;
- Extended full metabolomics analyses to justify the focus on nucleotides;
- Explanations of why the published literature does not undermine the novelty of our findings, and what is the conceptual advance that our study provides to the field;

The major changes in the text addressing referees' comments have been highlighted in green.

Reviewer #1 (Remarks to the Author): nucleotide metabolism

In this manuscript, Munk et al., set out to define how NAD levels impact metabolism and DNA damage response (DDR). The authors show that supplementation of cells with high levels of exogenous NAD results in an increase in ATP levels and DNA synthesis short term, while sustained high levels of NAD reduce cell proliferation, possibly due to unbalanced nucleotide pools. While the authors have performed a comprehensive analysis of the metabolic outputs and DDR in response to exogenous NAD supplementation, the latter represents a major flaw in this study given that NAD is not cell-permeable, and to date there are no known NAD transporters at the plasma membrane. Therefore, it becomes difficult to interpret the results not knowing how NAD regulates NAD or ATP levels, or nucleotide imbalance. Also, the manuscript does not present a major advance in the field due to published work illustrating links between nucleotide imbalance or NAD levels and replication stress (PMIDs: 35927450; 34806016).

We appreciate this referee's comments and we address all his/her concerns below, at the same time explaining the changes we have made in the revised version of our manuscript.

*Referee #1 is correct, NAD⁺ is not cell membrane-permeable, however, while the specific membrane transporter(s) have remained largely enigmatic, mammalian cells can transport exogenous NAD⁺ into the cytoplasm (Billington et al., 2008; Bruzzone et al., 2001; Bruzzone et al., 2006; Buonvicino et al., 2021). NAD⁺ can be imported through the purinergic receptor P2X7 (Alano et al., 2004) or the connexin 43 channels (Bruzzone et al., 2001). We included these important references in our manuscript. Given that the access of NAD⁺ to intracellular space is known, we aimed to study its impact on important cellular processes, rather than identifying the precise mechanism of its cell entry, which is another topic, not investigated in our present study. Experimentally, we tested whether exogenous NAD⁺ or some of its precursors could influence mitochondrial functions and we found that only intact NAD⁺ could rapidly alter mitochondrial activity.

Our results are consistent with the notion that the mitochondrial NAD⁺ pool can be established through the direct import of NAD⁺ (Davila et al., 2018), using the mitochondrial-specific NAD⁺ transporter SLC25A51 (Kory et al., 2020; Luongo et al., 2020). We interpreted our results as the most likely explanation for how NAD⁺ regulates mitochondrial functions that impact nucleotide biosynthesis. In the revised version we incorporated multiple new results to better justify our conclusions.

The mitochondrial pool of NAD(H), of course, can be synthesized from precursors which were shown before and reported by Ziegler's laboratory some years ago (Nikiforov et al., 2011). Of relevance, over a decade later, Zeigler co-authored a related paper showing the existence of the NAD⁺ transporter in the mitochondria (Luongo et al., 2020). Both models are correct and co-exist, and our new data support this concept. We integrated into our model both of these possibilities (see Extended Data Fig. 10a in the new version of our manuscript).

*We do not share the view that our dataset does not present a major advance in the field. Indeed, our present work is likely to considerably advance the current knowledge of how cells regulate genomic DNA synthesis and its interplay with mitochondrial functions. Furthermore, since treatment with NAD⁺ and its precursors are used in the literature interchangeably to manipulate intracellular NAD⁺ levels in cultured cells, our new findings, that NAD⁺ treatment impairs the synthesis of pyrimidine nucleotides with the concomitant accumulation of purine nucleotides, will lead to re-interpretation of published studies in which exogenous NAD⁺ was used as a mean to increase cellular NAD⁺: This important shift in viewing those previous studies is inevitable given that the authors of those papers were unaware of the previously uncharacterized effect of NAD⁺ treatment on nucleotide levels (discovered and reported in our present manuscript) when drawing their conclusions about NAD⁺ effects.

* At the end of his/her comments, Referee #1 included two very good recent publications, which are however not closely overlapping with the cellular events studied in our present research and our manuscript. Vander Heiden laboratory PMID: 35927450 (Diehl et al., 2022) reported that nucleotide imbalance decouples cell growth from proliferation: it is indeed a good manuscript that strengthens the relevance of our new, complementary results, and we cite this work in the revised manuscript. Nucleotide imbalances are known as a source of replication stress (Bester et al., 2011), and the Diehl et al. study shows that specifically the imbalances of nucleosides are sensed by the ATR-dependent signalling pathway, and this observation is probably the main aspect of the novelty of their manuscript. In the Diehl et al manuscript, there is no experimental evidence of how nucleotide imbalances can be naturally achieved, nor any molecular mechanistic insights into the cross-regulation between the nucleotide pools. Furthermore, Diehl et al. showed that supra-physiological levels of guanine reduced the levels of adenylate nucleotides and inhibited cell proliferation. After the addition of guanine, GTP was accumulated. In our case, the addition of NAD⁺ did not induce the accumulation of GTP, indicating

that the new mechanism of regulation of the nucleotide metabolism that we report here is different from that reported by Diehl et al.

Furthermore, our manuscript addresses an entirely different set of important questions, such as How do the levels of NAD⁺ impact RNA/DNA synthesis. While the Diehl et al paper studied the effects of treatment with nucleotide precursors, we demonstrate that extended NAD⁺ treatment directly modulates the mitochondrial metabolic activity with downstream consequences for DHODH functionality, causing the impairment of dihydroorotate to orotate conversion in *de novo* pyrimidine synthesis and, ultimately, depletion of pyrimidine nucleotides and accumulation of purines. This mechanism is not described in the aforementioned study or any other published work, and thereby represents our new, original findings that we now present in this NCB manuscript. We hope that these newly generated mechanistic results will also help to highlight the novelty of our revised manuscript. We also describe potential metabolic vulnerabilities in cancer cells that could be exploited in therapy, again an important issue not addressed by the papers mentioned by the reviewer.

The article PMID: 34806016 published in NAR cancer (Li et al., 2021) is even less related to our present manuscript, but it is more related to our work that we published in Nature some 5 years ago (Maya-Mendoza et al., 2018) where we reported the impact of PARylation on speed of replication fork progression and the different scenarios under which replication stress occurs, overall extending the concept of the major role of replication stress in genomic instability and tumorigenesis, a research direction that we pioneered almost 20 years ago in our two Nature papers by Bartkova et al. (Bartkova et al., 2005; Bartkova et al., 2006) and which has since then grown into an entire new field of biomedical research pursued by many laboratories. The Li et al. study is related to PARG inhibition and the accumulation of Poly-ADP-ribosylation (PARylation) in glioma cells. The authors combine PARG-inhibitors with dihydronicotinamide riboside (NRH), not surprisingly, cells treated with this combination accumulate PARylation that, as we discovered and reported in 2018 (Maya-Mendoza et al) impairs replication fork progression. Again, this work by Li et al. is unrelated to our present manuscript and we hope that the revised version of our manuscript and this rebuttal help to clarify this point.

Major comments:

1) The concentrations of NAD used throughout the study are supraphysiological. Moreover, there are no known NAD transporters at the plasma membrane, raising the question of whether NAD signals through the purinergic receptors, similar to other purine nucleotides. Does exogenous NAD enter the cells after being converted to NMN or NR?

We thank the referee for pointing out this previous weakness in our story. We agree that 2 mM is a supraphysiological concentration. We initially supplemented cells with 2 mM of NAD⁺ for two reasons; first, NAD⁺ at the millimolar range has been reported to boost intracellular NAD⁺ levels (Bruzzzone et al., 2001; Buonvicino et al., 2021; Maynard et al., 2022; Ying et al., 2003). Second, when we tested the effect of different concentrations of NAD⁺ on cell viability, these results were included in Fig 2a and Extended Data Fig. 3a in the first submission, 2 mM of NAD⁺ caused a notable peak of ATP increase with the arrest of cell proliferation.

Inspired by this reviewer's comment, we now present extensive evaluations of the effects of NAD⁺ treatment at a wide range of concentrations, at 16 μ M – 10 mM, on cell physiology. Interestingly, the effects of NAD⁺ treatment seem to be induced already at around 50 μ M, which is a physiologically relevant concentration, taking into account that the concentration of NAD⁺ in mitochondria ranges between 50-250 μ M (Cambronne et al., 2016) and that the Km for NAD⁺ transport by SLC25A51 is approximately 200 μ M (Luongo et al., 2020).

For an easier evaluation of this newly added large dataset, we include here below the results of the metabolomics analysis for different concentrations of exogenous NAD^+ . Consistent with our previous findings the pyrimidine depletion is achieved in experiments with the physiologically relevant 50-100 μM concentrations of NAD^+ .

NAD^+ treatment induces nucleotide imbalances. Shown in manuscript as Fig. 2c. Heatmap of metabolite levels from metabolomics analysis in HeLa cells treated with the indicated NAD^+ concentrations for 24h. Asterisks indicate significant changes ($p\text{-adj.} < 0.05$) to the non-treated (NT) condition.

NAD^+ degradation products, such as nicotinamide riboside and AMP, did not accumulate in the cell at these concentrations. However, since we observed differential effects by treatment with NAD^+ and its precursors on genomic DNA synthesis, this is consistent with the notion that such effects of NAD^+ are direct and not downstream of NAD^+ degradation into its precursors. Furthermore, since the effects of AMP treatment did not phenocopy those of NAD^+ treatment, this also indicates that extracellular degradation of NAD^+ to NMN and AMP does not activate purinergic receptors responsible for the observed effects of NAD^+ treatment on mitochondria.

2) Fig. 1: NR or NAD^+ treatment increases cellular NAD(H) level, but only NAD^+ treatment induces DDR response and inhibits DNA synthesis. Why is this response NAD specific? Are purinergic receptors or CD38 involved in metabolizing NAD and inducing downstream signaling pathways (Ca+ or C-ADPR)?

Indeed, one of the main findings presented in our manuscript is the differential response of the cell to treatment with NAD^+ or its precursors. The conversion to NAD^+ from precursors is determined by the level of expression of the relevant enzymes and the availability of the substrates. NAD^+ supplementation directly alters mitochondrial activity. Certainly, the treatment of cells with NR did not increase the mitochondrial membrane potential (Extended Data Figure 8b) in contrast to NAD^+ treatment, and only treatment with NAD^+ was able to induce an increase in ATP levels. The increase in the concentration of intracellular NAD^+ from the precursors such as NR or NMN seems not to alter mitochondrial functions. The excess of NAD^+ originating from precursors could be buffered by the increase of mitochondrial mass (Canto et al., 2012) or possibly by some additional regulatory mechanism(s) that are currently unknown.

Here, we have tested experimentally the potential involvement of CD38 and intracellular calcium signaling. Effects of NAD^+ treatment were unaltered by co-treatment with 1 μM CD38 inhibitor for 24h, indicating an involvement of CD38-independent independent mechanism(s). We appreciate this

comment from referee#1 and include these results in the revised version of our manuscript (see below and Extended Data Fig. 2n, o).

Effects of NAD⁺ treatment are independent of CD38 activity. Shown in the manuscript as Extended Data Fig. 2n-o. **Left panel:** QIBC analysis of γ H2AX foci during cell cycle in cells treated for 24h without (NT) or with 1 μ M CD38i and/or 2 mM NAD⁺. **Right panel:** Relative intracellular calcium changes measured with the Fluo-8 indicator in cells after treatment as in the left panel, mean+SD, n=3.

3) How do NAD⁺ levels increase the levels of purines, but decrease pyrimidines (Fig 2)?. Since the metabolite patterns between HeLa and U2OS are different, more cellular contexts are required to understand whether these phenotypes are generalizable.

We appreciate that the reviewer finds these results new and surprising, and we fully agree that the examination of more cellular models is helpful in this context. We have now tested a panel of different cell types and all of them exhibit replication sensitivity to NAD⁺ treatment (Fig. 1f). A natural cause for the variation in sensitivity between cell types could be due to the pre-existent levels of nucleotide pools and the potential for anaplerosis of TCA cycle intermediates. Indeed, we observed a higher potential for anaplerosis in metabolomics analysis of U2OS cells compared to HeLa cells (Fig. 3a), and the higher pyrimidine levels maintained in U2OS cells compared to HeLa cells (Fig. 2e) could further provide them with the observed increased resistance to DHODH inhibitor-induced pyrimidine synthesis impairment (Fig. 4g).

The accumulation of purines after the incubation of cells treated with NAD⁺ is a later event that follows pyrimidine depletion. There can be two possible scenarios for the accumulation of purines; 1) NAD⁺ treatment only depletes pyrimidines, thus purines will accumulate since they are not used for DNA synthesis and 2) NAD⁺ treatment depletes pyrimidines and, e.g. through its hydrolysis to NMN and AMP, elevates AMP levels, which can also be converted to GMP, and can thereby also function as a source to increase the pool of purine nucleotides. As suggested by this reviewer, In this new version of our manuscript, we have included additional cell types (Fig. 1f) and we have improved the analysis and presentation of the new and old metabolomic data (Figs. 2, 3).

4) Line 253: How does NAD⁺ regulate the activity of the TCA cycle and ETC to regulate DNA replication? In extended fig 3b, the TCA cycle metabolites fumarate and malate decrease in HeLa cells upon NAD treatment. There is not sufficient evidence to conclude that NAD regulates the DNA synthesis through “the NAD⁺ mediated-high activity of TCA cycle and ETC”

Our apology for not expressing our idea clearly in the previous version of the manuscript. We also provide more experimental evidence to support this conclusion. We stated that NAD⁺ treatment increased the activity of the TCA and ETC based on the following results:

*Treatment with NAD⁺ depleted most of the intermediates of the TCA in HeLa cells, presumably because the low endogenous levels of glutamine and aspartate in these cells were unable to sustain anaplerosis.

*After 1h of exposure to NAD⁺, the levels of UMP and uridine increased, and there was an increase in the mitochondrial activity in both U2OS and HeLa cell lines.

*The overactivation of mitochondrial function(s), specifically the TCA cycle and ETC, lead to the arrest of pyrimidine biosynthesis. Without pyrimidines, DNA synthesis cannot be sustained and exhibits higher vulnerability to such condition than the general transcription.

*Importantly, impairing TCA cycle anaplerosis and ETC proficiency using the inhibitors BPTES and oligomycin, respectively, rescued the replication defects caused by NAD⁺ treatment, directly demonstrating that TCA cycle and ETC activity are necessary for NAD⁺ induced i) hyperpolarization of the mitochondrial membrane potential, and ii) depletion of pyrimidines that are essential for genomic DNA replication and transcription.

In this revised version, we included new results showing that the DHODH inhibitor brequinar (BRQ), which caused inhibition of the *de novo* pyrimidine synthesis, phenocopies the effect of treatment with NAD⁺, demonstrating the NAD⁺ treatment impairs DHODH activity and that proficiency of the mitochondrial membrane potential is required for NAD⁺ effects on genomic DNA replication as observed by co-treatment with FCCP. Some of these results are shown below (and see Fig. 4 in the manuscript).

Effects of NAD⁺ treatment phenocopy DHODH inhibition and depend on mitochondrial membrane potential proficiency. Shown in the manuscript as Fig. 4d,g. **Upper panel:** Analysis of γH2AX foci during cell cycle with QIBC in cells treated for 24h without (NT) or with NAD⁺ as indicated and/or 2

μM brequinar (BRQ). **Lower panel:** Analysis of γH2AX foci during cell cycle with QIBC in cells treated for 24h without (NT) or with NAD^+ as indicated and/or 10 μM FCCP.

5) Line 254: Extend fig 7.g, what is the evidence for the conclusions that high AMP level inhibits pyrimidine synthesis?

It has been shown and is well-established that AMP could inhibit the activity of UMP5, resulting in the depletion of UTP (Weisman et al., 1988). Our results showed that the sensitivity to AMP is also cell-type dependent. The results were previously shown in Extended Data Fig. 7. To clarify this issue, in this new version of the manuscript, we show some of the results regarding sensitivity to AMP as a part of Fig. 2 (here shown after treatment with 100 μM and 2 mM AMP).

AMP treatment impairs DNA synthesis. Shown in the manuscript as Fig. 2h. QIBC quantification of EdU incorporation in EdU-positive cells. HeLa and U2OS cells were treated without (NT) or with the indicated AMP concentrations for 24h. $n > 1,600$ cells per condition.

To further validate the aforementioned published effects of AMP treatment for the reviewer, we co-treated cells with 2 mM of AMP and uridine for 24h, which would rescue the AMP-induced pyrimidine deficiency, and evaluated the effect thereof by cell cycle profiling, and the rescue effect of uridine on eliminating the S-phase accumulation was indeed observed (shown below and Fig. 4f).

Uridine supplementation rescues AMP-impaired S phase progression. HeLa and U2OS cells were treated for 24h without or with 2 mM AMP and/or 100 μM uridine as indicated. Hoechst was then used to stain the DNA of the cells for cell cycle analysis by QIBC.

In this manuscript version, we included the manipulation of AMP levels (at 2 mM) in combination with the inhibitors oligomycin or BPTES, two inhibitors we found to rescue the effects of NAD⁺ treatment on DNA replication. These and previous results showed that the regulation of DNA synthesis, via regulation of pyrimidine synthesis, by AMP occurs downstream of the reaction catalyzed by DHODH.

Effects of AMP treatment are independent of mitochondrial metabolic proficiency. Shown in the manuscript as Fig. 4e. QIBC analysis of γ H2AX foci during cell cycle in HeLa and U2OS cells treated without (NT) or with 2 mM AMP, 5 μ M oligomycin (Oligom), and/or 10 μ M BPTES for 24h.

Moreover, we agreed with the referee's concern and modify the sentence by saying that our experimental evidence shows that a high level of AMP reduced the incorporation of EdU incorporation (Page 10, line 213).

6) Line: 288-290: The presented data are not sufficient to conclude that the “effects of NAD⁺ depend on mitochondrial NAD⁺ transport in U2OS cells, while HeLa cells remain sensitive to NAD⁺ in the absence of SLC25A51 owing to AMP-regulated mechanisms”.

We thank the reviewer for this comment. We performed a new set of experiments to better support this conclusion.

First, our metabolome analysis, using different concentrations of NAD⁺ to treat the cells, showed that AMP starts to accumulate when NAD⁺ treatment at concentrations above 80 μ M is used.

AMP levels in response to treatment with increasing NAD⁺ concentrations. Relative quantification of AMP from metabolomics analysis in HeLa cells treated with the indicated NAD⁺ concentrations for 24h, n=4, mean \pm SD shown.

Therefore, we hypothesized that the effect of AMP on inhibiting the incorporation of EdU should only be seen at NAD^+ concentrations higher than $80 \mu\text{M}$ in HeLa cells, while U2OS cells should exhibit reduced sensitivity. Indeed, this was the case (see results below). These results are presented as part of the Fig. 5c.

Furthermore, if the potential AMP-regulated mechanism is independent of the mitochondrial regulation of pyrimidine anabolism, the ablation of the mitochondrial transporter SLC25A51, and hence ablation of elevation of mitochondrial NAD^+ following NAD^+ treatment (see results below), should not interfere with its regulation downstream of the mitochondrial step in pyrimidine synthesis. We confirmed this prediction, with HeLa cells remaining sensitive to NAD^+ treatment and U2OS cells losing their sensitivity after si-SLC25A51 treatment and found that the knockdown of SLC25A51 was also instrumental for maintaining genomic DNA replication.

Effect of SLC25A51 depletion on NAD^+ treatment effects and mitochondrial NAD(H) levels. Shown in the manuscript as Fig. 5c and Extended Data Fig. 9d. **Upper panel:** QIBC analysis of γH2AX foci during cell cycle. Cells transfected with control siRNA (CTRL) or si-SLC25A51 were treated for 24h with NAD^+ as indicated. **Lower panel:** Quantification of total mitochondrial NAD(H) in cells treated as in upper panel but with only 2 mM of NAD^+ , NT = non-treated. $n=3$ (HeLa) and $n=2$ (U2OS), mean+SD, Student's t-test (HeLa).

7) The biological or physiological relevance of high exogenous NAD levels is unclear, given that plasma levels of NAD are in micromolar rather than millimolar range. Does supplementation of high levels of NAD in vivo have similar effects as presented in the two cell culture models?

Our new results showed that even a slight increase in exogenous NAD⁺ changes cell physiology. At the lowest concentration tested, 16 μM NAD⁺, we already observed metabolic changes and altered dynamics of genomic DNA synthesis.

Replication sensitivity to low concentrations of NAD⁺ treatment. Shown in the manuscript as Fig. 1h-i. Replication fork speed in HeLa and U2OS cells treated without (NT) or with the indicated NAD⁺ concentrations for 24h. Mean±SD, n>500 fibers per condition per cell line.

Regarding the relevance in vivo, it seems that normal cells are more resistant to any variation in NAD⁺ levels (Fig. 5) while highly proliferative tumors that rely on the *de novo* pyrimidine synthesis can be targeted in a combined therapeutical setting.

The next phase of our research will involve clinical trials, most likely without the otherwise recommended intermediate step of testing in animal models first, because NAD⁺ is used as anti-aging therapy in humans and offered by clinics around the world. Worryingly, the concentration of NAD⁺ used in these therapies is at the millimolar level and it is injected intravenously. The typical treatment commonly uses 500 – 1000mg of NAD daily over 4 – 10 days.

<https://resetiv.com/products/nad>

<https://nadclinic.com/injections/#>

Moreover, millimolar concentrations of NAD⁺ are also used currently to treat diverse types of addictions (Braidly et al., 2020).

<https://www.springfieldwellnesscenter.com/about-nad/>

Further careful research on the effects of NAD⁺ at the cellular level is imperative to optimize the various treatments in humans, and our results contribute to the understanding of the biological impact of this fascinating molecule.

Reviewer #2 (Remarks to the Author): DNA replication stress, PARP

In this manuscript, the authors investigated how NAD⁺ levels affect DNA synthesis and genomic stability. They found that exogenous NAD⁺ inhibits DNA synthesis and induces DNA damage. By metabolic profiling, they showed that exogenous NAD⁺ induces an imbalance of nucleotides, reduces overall transcription, and activates mTOR. Their RNA-seq analysis of NAD⁺ treated cells uncovered that ETC activity is enhanced by NAD⁺ in mitochondria. Blocking ETC by oligomycin rescued the effects of NAD⁺ on DNA synthesis. Inhibition of the conversion of glutamine to glutamate and the TCA cycle by BPTES also rescued the effects of NAD⁺ on DNA synthesis. These results suggest that the TCA cycle and ETC are important for the effects of NAD⁺ on DNA synthesis, which may be mediated by unbalanced nucleotides. The authors also noticed differences between HeLa and U2OS cells in their responses to exogenous NAD⁺. HeLa cells accumulate higher levels of AMP after NAD⁺ treatment, which may also contribute to the inhibition of DNA synthesis. Finally, the authors proposed that NAD⁺ can be combined with drugs that alter pyrimidine levels in cancer therapy.

The main finding of this study that high NAD⁺ can alter nucleotide balance and inhibit DNA synthesis is interesting. However, almost of the experiments supporting this conclusion are based on the use of a high concentration of exogenous NAD⁺. Whether this approach is relevant to any physiological or pathological conditions in aging, cancer, and therapy is unclear. Whether NAD⁺ induced changes in nucleotide balance are the actual cause of reduced DNA synthesis and induction of DNA damage is not tested directly. Whether and how enhanced ETC is responsible for the unbalanced nucleotides is also untested. The overall model of this study is still too preliminary, with major gaps to be filled. Although this study is focused on the effects of NAD⁺ on DNA synthesis and genomic stability, the metabolomics data from this study show that NAD⁺ affects many other pathways and processes. It seems to me that the impacts of NAD⁺ in cells are much more complex than what is proposed in the model, and I am not sure if this study can establish general principles as claimed.

We appreciate the comments from referee#2, they have helped us in the last 6 months to improve our manuscript. The referee raised several valid concerns, most of which reflected our poor introduction and description of our results. In this new version of our manuscript, we provide numerous new experiments and amend the text to validate and support our proposed model. Aligned with the reviewer's comments, we now demonstrate that the previously observed effects of NAD⁺ are also induced by much lower, physiologically relevant concentrations of exogenous NAD⁺. The NAD⁺ treatment-induced nucleotide imbalances, namely depletion of pyrimidines with concomitant accumulation of purines, that we also demonstrate are balanced and alleviated by impairing the activity of the TCA cycle and ETC as well as co-treating cells with uridine to fuel pyrimidine synthesis independent of mitochondrial DHODH activity. We very much hope that referee#2 will find our revised manuscript sufficiently improved.

1. In Fig. 1b and 1c, the use of SIRT1i and PARPi to study the effects of increased NAD⁺ is not completely justified. Both SIRT1 and PARP1 have many functions independent of their effects on NAD(H) levels. In Fig. 1b, it is clear that SIRT1i and PARPi only modestly increased NAD(H) levels. However, it is unclear whether the modest increase of NAD(H) in PARPi treated cells is relevant to the effects of PARP1i on DNA synthesis and genomic stability.

We appreciate this comment from referee#2. Our apology for not being clear to introduce the rationale behind these experiments.

Referee #2 is correct in stating that SIRT and PARP inhibitors alter diverse processes in the cell, as one of the downstream effects of their inhibition is the increase of intracellular NAD(H). The levels of NAD(H) drop with advancing age in humans, therefore, the use of PARP inhibitors might alleviate some of the aging features (Canto et al., 2015; Pirinen et al., 2014). This may be true, however, given the role of PARPs in DNA replication and genome integrity maintenance, such beneficial anti-aging effects seem unlikely.

Our results showed that the increase in intracellular NAD(H) after the inhibition of PARP1 or SIRT1 is independent of their roles in DNA synthesis or repair. Most likely due to the reduction of PARylated proteins or altered chromatin acetylation. Indeed, the levels of intracellular NAD(H) after the inhibition of SIRT1 increased only mildly with no statistical significance, after PARP1-inhibition there was a significant increase of ~50 % (Fig. 1b), despite their shared increase of intracellular NAD(H) after the inhibition of PARP1 or SIRT1, their respective effects on DNA synthesis were entirely different as we showed by EdU incorporation (Fig. 1d in the new version of our manuscript) and DNA fibre assay (Extended Data Figure 1d,e). Interestingly, the inhibition of NAMPT depleted 90% of the total NAD(H), with a significant negative impact in HeLa cells but not in U2OS cells.

Next, in our experiments, we tried to rescue the effect of low NAD(H) by supplementing cells with NAD⁺. These experiments revealed the unexpected results that we then further characterized and mechanistically elucidated in the subsequent parts of our manuscript.

b

Manipulation of intracellular NAD(H) levels by inhibitor treatment. Shown in the manuscript as Fig. 1b. NAD(H) quantification in cells treated without (NT) or with 10 nM NAMPTi, 2 μM SIRT1i or 10 μM PARPi for 24h. Mean+SD, n=3, Student's t-test.

d

Differential DNA synthesis defects induced by NAD⁺ treatment and PARP or NAMPT inhibition. Shown in the manuscript as Fig. 1d. QIBC analysis of γH2AX foci during cell cycle in HeLa cells treated for 24h without (NT) or with 10 nM NAMPTi, 10 μM PARPi and/or 2 mM NAD⁺.

Replication forks in U2OS cells are not affected by NAMPT inhibition. Shown in the manuscript as Extended Data Fig. 1d-e. Replication fork speed (**left panel**, mean±SD) and symmetry (**right panel**) in U2OS cells treated for 24h without (NT) or with 10 nM NAMPTi, 2 μM SIRT1i or 10 μM PARPi. n>425 forks per condition.

We corrected and extended the text to explain better the rationale behind all those experiments (Page. 4, lines 80-82).

2. In Fig. 1b, NAMPTi drastically reduced NAD(H) in both HeLa and U2OS cells, but it only reduced DNA synthesis in HeLa cells. This difference between HeLa and U2OS is never explained in the following experiments. The 1.5 fold higher NAD⁺ in U2OS is unlikely sufficient to explain the difference. The authors completed switched to using exogenous NAD⁺ to inhibit DNA replication, which is a completely different phenomenon.

Again, our apology because we did not discuss enough the differences in sensitivity to NAMPT inhibitors between HeLa and U2OS. In this new version of our manuscript, we show more experimental evidence to explain the differences in sensitivity to NAMPT inhibitors between HeLa and U2OS cells (Fig. 3).

Our results showed that in general, cells tolerate low intracellular levels of NAD(H). They may shift to use some other redox molecules to support their basic metabolism. Our analysis of metabolites showed that HeLa cells under untreated conditions have a low level of certain TCA intermediates, specifically, upstream of the complex II-mediated oxidation of succinate (*succinic acid* below). Fumarate (*fumaric acid* below) is depleted after NAD⁺ treatment in these cells (Fig. 3a and below). Because the level of glutamine and glutamate is very low in HeLa cells, the anaplerosis of the TCA via α-ketoglutarate (also known as 2-oxoglutaric acid) is also limited.

Changes in the TCA cycle intermediates succinate and fumarate after NAD^+ treatment. Crop of Fig. 3a in the manuscript. Metabolite levels from metabolomics analysis in HeLa and U2OS cells treated without (NT) or with 2 mM NAD^+ for the indicated hours. The y-axis displays relative metabolite quantification. Bars represent means, $n=4$.

Together, these metabolic features suggested that HeLa cells but not U2OS cells could have impaired complex II activity, and therefore, be more sensitive to complex I inhibition as well as the depletion of $NAD(H)$ induced by NAMPT inhibitor. Indeed, the replication forks of HeLa cells were more sensitive to metformin, an inhibitor of complex I.

Differential replication sensitivity to metformin in HeLa and U2OS cells. Shown in the manuscript as Fig. 3c-d. Replication fork speed in HeLa and U2OS cells treated without (NT) or with 2 mM NAD^+ , 10 μM BPTES, and/or 1 mM metformin (Met) for 24h. Mean \pm SD, $n>500$ fibers per condition.

3. The NAD^+ rescue experiments in Fig. 1d and 1e are confusing. NAD^+ partially rescued the effects of NAMPTi in HeLa, but it acted independently of NAMPTi in U2OS cells. Since NAMPTi reduced endogenous $NAD(H)$ in U2OS (Fig. 1b), should the effects of exogenous NAD^+ be at least reduced by NAMPTi?

This is a very interesting point raised by referee#2. We thought the same way as referee #2, however, our data showed that the effect of NAD^+ in U2OS cells is dominant over the inhibition of NAMPT.

We explain this by the fact that U2OS cells tolerate a low level of $NAD(H)$, also supported by the well-tolerated inhibition of complex I by metformin, and due to their high potential for anaplerosis via the

glutamine-glutamate- α Ketoglutarate axis, they can feed the ETC by complex II. When we added NAD^+ to U2OS cells, because they rely on ETC flow to regulate the activity of the DHODH, they accumulate carbamoyl aspartate and dihydroorotate (Extended Data Figure 6a).

The partial alleviation of the effect of NAMPT inhibitors by NAD^+ in HeLa cells is likely due to the differential regulation of nucleotide anabolism.

We clarify this phenomenon on Pag. 11.

Changes in intermediates of de novo pyrimidine synthesis after NAD^+ treatment. Crop of Extended Data Fig. 6a in the manuscript. Metabolite levels from metabolomics analysis of HeLa and U2OS cells treated without (NT) or with 2 mM NAD^+ for the indicated hours. In metabolite plots, the y-axis displays relative metabolite quantification. Bars represent means, n=4.

4. Adding 2mM exogenous NAD^+ to cells is a very artificial manipulation. It is not clear whether this high concentration of NAD^+ is relevant to any physiological or pathological conditions in aging, cancer, or therapy. The effects of this extremely high concentration of exogenous NAD^+ may or may not be achievable in cells under any physiological or pathological conditions. This is a major concern on the experimental approach of the current study. The authors should confirm their findings using other strategies to increase endogenous NAD^+ .

We acknowledge this comment and, as suggested by the reviewer, we have performed an extensive set of new experiments to validate our model of how NAD^+ treatment, even at lower, physically relevant concentrations, regulates nucleotide biosynthesis, genomic DNA synthesis, and cell proliferation.

We agree with this concern, we did not emphasize enough some of our results regarding different treatment concentrations of NAD^+ in the previous version of our manuscript where we tested several concentrations of NAD^+ to test its effect on cell viability and energy production (Fig. 2a). We have now revised and updated our manuscript to provide evidence that NAD^+ treatment blocks pyrimidine anabolism, inhibiting genomic DNA synthesis at concentrations as low as 80 μM , and that this effect did not change further at concentrations beyond 400 μM (Fig. 1g).

DNA synthesis sensitivity to increasing concentrations of NAD⁺ treatment. Shown in the manuscript as Fig. 1g. EdU incorporation in EdU-positive cells quantified by QIBC. HeLa cells were treated without (NT) or with the indicated NAD⁺ concentrations for 24h, $n > 6,000$ cells per condition. P-values were calculated from comparison to the NT condition.

To allow for a stronger, better-validated interpretation of these results, we also performed the full metabolomic analysis of this experimental condition (shown now in Fig. 2c).

NAD⁺ treatment induces nucleotide imbalances. Shown in the manuscript as Fig. 2c. Heatmap of metabolite levels from metabolomics analysis in HeLa cells treated with the indicated NAD⁺ concentrations for 24h. Asterisks indicate significant changes ($p\text{-adj.} < 0.05$) to the non-treated (NT) condition.

In the previous version of our manuscript, we mainly used 2 mM of NAD⁺ because it is a concentration that has been used in recent studies to boost intracellular NAD(H) (Luongo et al., 2020; Maynard et al., 2022; Szczepanowska et al., 2020), and because our results showing that cells reach the maximum ATP production with a decrease in cell viability, a metabolic link that has never been explored in the literature.

Effects of increasing NAMPT inhibitor or NAD⁺ concentrations on cell proliferation and ATP levels. Shown in the manuscript as Fig. 2a. Relative ATP level and cell number to non-treated cells for each cell line. The indicated cell lines were treated with the indicated concentrations of NAMPTi or NAD⁺ for 48h (HeLa and U2OS) or 72h (BJ and BJ-T). Smoothed conditional means and CI95 are shown, $n > 4$.

We are confident that this revised version of our manuscript is substantially improved by the added data from the new set of experiments performed in response to the constructive criticisms and suggestions from the reviewers.

We also confirmed our results using known precursors of NAD⁺ that boost intracellular NAD⁺ levels. An interesting new result shows that only treatment with NAD⁺ directly was able to rapidly change mitochondrial activity, impact DNA replication and arrest the cell cycle.

Treatment with NAD⁺ but not its precursors impair DNA replication. Shown in the manuscript as Fig. 1l-m. **Left panel:** Replication fork speed in HeLa cells treated without (NT) or with 2 mM NAD⁺, NR or NMN for 24h. Means±SD are shown, $n > 400$ fibers per condition. **Right panel:** QIBC analysis of γH2AX foci during cell cycle in HeLa cells treated without (NT) or with 2 mM NAD⁺, NR or NMN for the indicated hours.

5. The authors did not interpret Fig. 1m clearly. Even if high levels of NAD⁺ do not affect RRM activity, why would addition of dNTPs reduce DNA synthesis? Can this be explained by unbalanced nucleotide pools? Why?

We apologize since, due to space restrictions of the concise letter format, we left some results insufficiently described in our original manuscript.

For the deoxy-nucleoside supplementation, we wanted to test whether NAD^+ treatment could affect directly the activity of the Ribonucleotide reductase (RNR). Because we did not know specifically which deoxy-nucleotides were the most affected by NAD^+ treatment, we initially supplemented cells with all four deoxy-nucleosides. This supplementation reduced even further the DNA synthesis when it was combined with NAD^+ treatment. This showed that the deoxy-nucleosides and the NAD^+ treatment triggered different pathways that resulted in further inhibition of DNA replication. It turned out that the exogenous thymidine was responsible for this additive effect. An excess of thymidine has been used historically to arrest the cell cycle in the border G1/S of the cell cycle. In a cell-dependent manner, the excess of thymidine expands the dTTP pools and inhibits the activity of RNR with the consequent depletion of dCTP (O'Dwyer et al., 1987). We then confirmed the mutually independent nature of the molecular mechanisms triggered by NAD^+ treatment and the inhibition of the RNR, respectively, using hydroxyurea (HU). Hydroxyurea phenocopied the effect of thymidine in the reduction of EdU incorporation (These new results are shown in Fig. 4h-k).

6. In Fig. 2, the authors found that 24 h of NAD^+ treatment reduced pyrimidines but increased purines, among many other metabolic changes. They speculated that NAD^+ induced imbalance of nucleotides is the cause of reduced DNA synthesis and genomic instability. However, this idea is never directly tested. This is a major weakness of the current study.

Together with new data sets, we showed that NAD^+ treatment induces the accumulation of some purines and the depletion of pyrimidines. This phenomenon is time- and concentration-dependent.

Changes in pyrimidine and purine nucleotide levels in response to increasing NAD^+ concentrations. Shown in the manuscript as Fig. 2d. Mean nucleotide levels (bars) from metabolomics analysis and replication fork speed (boxplots) relative to the non-treated (NT) condition. HeLa cells were treated with the indicated NAD^+ concentrations for 24h.

The lack of pyrimidines is followed by a reduction in the replication fork speed, moreover, the treatment with oligomycin or BPTES alleviated the nucleotide imbalances induced by NAD^+ treatment and restored DNA replication.

Co-treatment with oligomycin or BPTES alleviates NAD^+ -induced nucleotide and replication fork impairments. Shown in the manuscript as Fig. 3g. Mean nucleotide levels (bars) from metabolomics analysis and replication fork speed (boxplots) relative to the non-treated (NT) condition. HeLa cells were treated with 5 μM oligomycin (Oligom), 10 μM BPTES and/or 2 mM NAD^+ as indicated for 24h.

In the new version of our manuscript, we avoid using the wording imbalance and use instead “depletion of pyrimidines” and “accumulation of purines”. Furthermore, to test directly whether pyrimidine depletion was the cause of compromised genomic DNA replication, cells were also co-treated with NAD^+ and uridine to replenish pyrimidine pools and restore nucleotide balance independently of the DHODH activity, which was impaired by the treatment with NAD^+ . Consistently, uridine supplementation fully rescued genomic DNA replication in NAD^+ -treated cells (Fig. 4f and below for convenience).

Uridine supplementation restores DNA synthesis in NAD^+ -treated cells. Shown in the manuscript as Fig. 4f. QIBC analysis of $\gamma H2AX$ foci during the cell cycle in cells treated for 24h without (NT) or with 100 μM uridine (Urd) and/or 2 mM NAD^+ .

We hope that the new analyses of metabolites, together with our response to the previous concern #5 of this reviewer will answer this point.

7. In Fig. 3i and 3j, the rescuing effects of oligomycin on NAD⁺ induced fork slowing is impressive. Can the authors confirm these effects by directly disrupting ETC?

Thank you very much for this comment.

Indeed, the result was impressive, and we supported it with the new analysis of metabolites (see our answer to the previous comment no. 6).

Inspired by this comment of the reviewer, we disrupted the ETC by using FCCP, and to confirm the role of the ETC in NAD⁺-mediated regulation of pyrimidine metabolism, we inhibited the DHODH complex involved in *de novo* pyrimidine synthesis.

The inhibition of DHODH by brequinar (BRQ) phenocopied the effect of NAD⁺ and, is highly relevant for cancer treatment, in this way, we also found a physiological indirect inhibitor of the activity of DHODH.

Effects of NAD⁺ treatment phenocopy DHODH inhibition. Shown in the manuscript as Fig. 4g. Analysis of γ H2AX foci during cell cycle with QIBC in cells treated for 24h without (NT) or with NAD⁺ as indicated and/or 2 μ M brequinar (BRQ).

We dedicated almost the entire new Fig. 4 to answering this and the following concerns.

8. The effects of FCCP are confusing. Since FCCP should disrupt ATP synthesis, should it also rescue fork speed? Why would FCCP alone reduce DNA synthesis in Fig. 3k? Why are the effects of FCCP different from Oligomycin?

We hope our answer below will clarify the confusion about this point.

Carbonyl cyanide-p-trifluoromethoxyphenylhydrazone (FCCP) ablates the membrane potential, these results are included in Fig. 4c.

Effects of mitochondrial OXPHOS inhibitors on NAD⁺-induced elevation of the mitochondrial membrane potential. Shown in the manuscript as Fig. 4c. Ratios of TMRM and mitotracker green (MTG) dye-intensities normalized to the non-treated (NT) condition. HeLa and U2OS cells were treated as indicated for 24h with 2 mM NAD⁺, 5 μM oligomycin (Oligom), 10 μM BPTES, and/or 10 μM FCCP.

Referee #2 is correct, as a consequence of FCCP treatment, the mitochondrial membrane potential and hence ATP synthesis is inhibited. FCCP reduced genomic DNA synthesis by itself in both cell types but only robustly prevented effects by NAD⁺ treatment in U2OS cells, in contrast to only partial impact in HeLa cells (see below). Consistently with a distinct, AMP-mediated regulation in HeLa cells, treatment with 2 mM of NAD⁺ further decreased the incorporation of EdU.

These results showed that to affect DNA synthesis, NAD⁺ requires mitochondrial membrane proficiency.

NAD⁺ treatment effects depend on mitochondrial membrane potential proficiency. Shown in the manuscript as Fig. 4d. Analysis of γH2AX foci during cell cycle with QIBC in cells treated for 24h without (NT) or with NAD⁺ as indicated and/or 10 μM FCCP.

Oligomycin inhibits the ATP synthase (complex V) but maintains the membrane potential. The inhibition of ATP synthesis by oligomycin reduces the electron flow of the ETC. NADH remains high and the NAD⁺ becomes low to feed the TCA. Each of these effects was sufficient to alleviate the effect of NAD⁺ treatment on DNA synthesis.

9. At the end of Fig. 3, the authors speculated that ETC is important for the maintenance of balanced nucleotide pools. Again, this idea is based on correlations of changes and is never tested directly. This is another major weakness of the current study.

For this answer, please see our answers to the previous two concerns (no. 7 & 8), specifying how we have now directly tested and defined this notion.

10. In Fig. 4c, the rescuing effects of BPTES on fork speed are clear. If BPTES affects TCA and indirectly affects ETC, it should have similar effects on mitochondrial membrane potential as Oligomycin and FCCP. The authors should test this directly.

We measured the mitochondrial membrane potential for those conditions. Altogether, our results showed that to accelerate or inhibit pyrimidine anabolism, NAD⁺ treatment requires proficient mitochondrial membrane potential. BPTES reduces the flow of the TCA cycle and the rate to the ETC, by slowing down the TCA, BPTES alleviated the effect of NAD⁺ in DNA synthesis.

Oligomycin, BPTES and FCCP impacted differently the membrane potential or the ETC flux (Fig. 4c).

11. The point on AMP in Fig. 4 is confusing. The authors tried to explain why HeLa cells are sensitive to NAD⁺ than U2OS by showing that HeLa cells accumulate more AMP and AMP inhibits DNA synthesis. However, even in HeLa cells, BPTES fully rescued DNA synthesis in the presence of exogenous NAD⁺ (Fig. 4c), even though fork speed was only partially rescued (Fig. 4a). Does this mean that the effects of endogenous AMP in HeLa cells are not detected by overall EdU incorporation? The effects of exogenous AMP in Fig. 4d could be artificial.

The former of the two scenarios suggested by the reviewer is correct. The DNA fibre technique can detect small variations in fork speed and stability. A reduction of fork speed can be compensated by dormant origin activation to complete the S phase at a constant pace (Blow et al., 2011). A reduction of fork speed below 30-50% will trigger DNA damage response and it cannot be compensated by dormant-origin activation (Koundrioukoff et al., 2013; Maya-Mendoza et al., 2018). A less extreme reduction of fork speed above this threshold will be compensated by dormant origin activation and resulting in no changes in global DNA synthesis. In other words, the mean intensity of EdU incorporation could remain the same. Both approaches complement each other in the analysis of DNA synthesis and provide different information.

Regarding the regulation by AMP, we provide more evidence to support this regulation in our model.

AMP accumulation can inhibit the activity of the UMPS. HeLa cells treated with NAD⁺ accumulated IMP and AMP in a concentration-dependent manner. AMP could originate from the degradation of exogenous NAD⁺, which can be hydrolyzed to NMN and AMP, thereby fuelling the endogenous levels of AMP, alternatively, originate from passive accumulation of purine nucleotides upon the NAD⁺-induced depletion of pyrimidines, given DNA synthesis being halted due to pyrimidine shortage.

AMP levels in response to treatment with increasing NAD⁺ concentrations. Relative quantification of AMP from metabolomics analysis in HeLa cells treated with the indicated NAD⁺ concentrations for 24h, $n=4$, mean \pm SD shown.

Upon NAD⁺ treatment, U2OS accumulated hypoxanthine, which itself is a degradation product of AMP and has been proposed to protect against AMP-induced toxicity due to pyrimidine levels.

Changes in purine metabolites after NAD⁺ treatment in HeLa and U2OS cells. Shown in the manuscript as Fig. 2g. Metabolite levels detected by metabolomics analysis in cells treated without (NT) or with 2 mM NAD⁺ for the indicated hours. The y-axis displays relative metabolite quantification, bars represent means, $n=4$.

Because treatment with 80 μ M of NAD⁺ had already a negative impact on EdU incorporation and fork speed, if there was another mechanism for the regulation of pyrimidine biosynthesis in HeLa cells, and it was dependent on AMP levels, it could be tested experimentally.

Treatment with 2 mM of NAD⁺ had an additive effect on reducing the EdU incorporation together with FCCP in HeLa cells but not in the U2OS cells.

The regulation by AMP is also active in U2OS cells, however, they are not as sensitive as HeLa cells owing to the differential expression of enzymes involved in purine metabolism, the scenario which is now clearly described in the new version of the manuscript. And because the regulation of pyrimidine anabolism by AMP is downstream of the activity of DHODH, it was insensitive to the alleviation by oligomycin or BPTES.

Effects of AMP treatment are independent of mitochondrial metabolic proficiency. Shown in the manuscript as Fig. 4e. QIBC analysis of γ H2AX foci during cell cycle in HeLa and U2OS cells treated without (NT) or with 2 mM AMP, 5 μ M oligomycin (Oligom), and/or 10 μ M BPTES for 24h.

The AMP-dependent regulation in HeLa cells was also evident when we depleted the mitochondrial NAD⁺ transporter (Fig. 5c).

12. In Fig. 5a, it seems that SLC25A51 does not have any impact on the NAD⁺ induced replication inhibition in HeLa cells. This is not consistent with the other results that ETC has a major contribution to this NAD⁺ effect. AMP should be only partially responsible for the effect.

The depletion of SLC25A51 has a mild effect on the DNA synthesis in HeLa and a stronger effect on the U2OS. We do notice that the knockdown of the gene has a deeply negative effect on DNA synthesis, particularly in U2OS. This could explain why only in some cell lines the knockout of this gene was compatible with viability and hence feasible (Luongo et al., 2020). Consistently with an AMP-mediated regulation, 80 μ M of AMP slightly enhanced the inhibition of EdU incorporation in HeLa cells but we did not see any effect on U2OS cells. And 2 mM of NAD⁺ had a strong effect on HeLa cells but mildly affected U2OS which were depleted to SCL25A51.

In agreement with the referee's comment, AMP was only partially responsible for the effect of NAD⁺.

Effect of SLC25A51 depletion on NAD⁺ treatment effects. Shown in the manuscript as Fig. 5c. QIBC analysis of γ H2AX foci during cell cycle. Cells transfected with control siRNA (CTRL) or si-SLC25A51 were treated for 24h with NAD⁺ as indicated.

13. The interpretations of Fig. 5e and 5f are confusing. The effects of PARPi and NAD⁺ are clearly distinct (Fig. 5f) and additive (Fig. 5e). This argues that PARPi and NAD⁺ are not epistatic. The results of these experiments actually suggest that the effects of PARPi are independent of its impact on NAD(H).

Yes, the referee interpreted correctly our results. Our mistake if we described them in a confusing manner.

The effect of PARP inhibition in DNA synthesis is independent of the increase of NAD(H). To extend this observation we now include new results testing the combination of the PARP1 inhibitor at different concentrations with also different concentrations of NAD⁺. We used HeLa and normal fibroblasts, and three conclusions can be drawn, i) PARPi and NAD⁺ are not epistatic, ii) MRC5 normal fibroblasts are more sensitive to the inhibition of PARP1, iii) the use of PARP1 inhibitors to boost intracellular levels of NAD(H) and thereby alleviate some aspects of aging may have other adverse effects.

Effects of NAD⁺ and PARP inhibitor treatment combinations on cell proliferation. Shown in the manuscript as Fig. 5e. Relative cell number to non-treated cells. HeLa and MRC5 cells were treated with 1 μM PARPi and/or the indicated NAD⁺ concentration for 72h. Mean+SD, n=6, Welch's t-test.

14. The authors idea that NAD⁺ precursors are safer because they affect mitochondria more slowly does not make sense. They should analyze the effects of these precursors on RS and DDR at later time points.

We removed this sentence and performed a new set of experiments to better elucidate this issue.

NR and NMN did not affect the global DNA synthesis even after prolonged incubations (see below), however, a DDR activation was seen after 48h with the accumulation of phosphorylated H2AX, however without compromising cell viability (See below and Extended Data Figure 2l, m).

These results pointed out that the use of precursors of NAD⁺ may induce DNA damage in the long term. We discussed this on Pag. 17, lines 4002-406.

Treatment with NAD⁺ but not its precursors impair DNA replication. Shown in the manuscript as Fig. 1m. QIBC analysis of γ H2AX foci during cell cycle in HeLa cells treated without (NT) or with 2 mM NAD⁺, NR or NMN for the indicated hours.

Differential DDR activation after treatment with NAD⁺ and its precursor NR. Shown in the manuscript as Extended Data Fig. 2m. Western blot of DNA damage response proteins in HeLa cells treated without (NT) or for the indicated hours with 2 mM NAD⁺ or NR.

Reviewer #3 (Remarks to the Author): NAD⁺, aging

Apolinar Maya-Mendoza's group and colleagues presented a research article entitled 'NAD⁺ regulates nucleotide levels and genomic DNA replication'. By the application of different cell lines (Human U2OS, HeLa, BJ, BJ-5ta (BJ-tert), T98G, and U87 MG cell lines) and a series of biochemical and omics approaches, the authors manipulated cellular NAD⁺ levels and investigated changes in metabolism and genomic DNA replication. The authors presented a large amount of well-organized data which could be informative to the NAD⁺ and DNA repair fields. However, my enthusiasm was dramatically dampened due to the below major concerns:

- **Low novelty on much of the data.** While impressive amount of data are presented in this paper, a significant amount of 'discoveries' claimed in this paper are not new. E.g., the statement 'Here we show that high levels of NAD⁺ rapidly boost Krebs's cycle, increase the ATP level, and accelerate the speed of genomic DNA synthesis' is not new as there are mountains of data in the fields from last century enabled the writings in our text book that NAD⁺ is essential for glycolysis, TCA cycle, and OXPHOS (thus more ATP) (see review summaries from the Guarente, Auwerx and Bohr groups). The importance of NAD⁺ in DNA replication was evidenced from 1960s and has been well-studied onwards (PMID: 4310516).

We thank referee#3 for his/her comments.

It was not our intention to claim as new findings the well-known and established roles of NAD(H) in glycolysis, Kreb's cycle or OXPHOS. Perhaps we did not highlight enough the main new findings of our manuscript and did not put them in a proper context. We performed a substantial number of new experiments to validate and test our concept of how an excess of NAD⁺ can modulate genomic DNA synthesis. This has not been described in the literature yet, and here we also propose the new mechanism behind the observed cellular effects.

The doubts based on the seemingly lacking novelty of our findings can be due to two factors: i) Our poor description of our results and potentially confusing interpretation (aspects that we tried to remedy in this new version of the manuscript); ii) A misunderstanding by the reviewer, possibly due to overlook of some of our results. For instance, there is no published evidence that an increase in ATP results in an accelerated speed of DNA synthesis and it was not our intention to imply that. The increase in ATP was a consequence of the treatment with NAD⁺, and we find that high ATP levels correlate inversely with the speed of DNA synthesis and the inhibition of cell proliferation. Furthermore, the main findings and novelty of our study include the observation that treatment with NAD⁺ in the micromolar range induces depletion of pyrimidine nucleotides, with the concomitant accumulation of purine nucleotides, in a manner directly related to mitochondrial metabolic activities. This central and new concept that we present is now further tested and validated in this revised manuscript, improved by additional sets of experimental results.

Changes in pyrimidine and purine nucleotide levels in response to increasing NAD⁺ concentrations. Shown in the manuscript as Fig. 2d. Mean nucleotide levels (bars) from metabolomics analysis and replication fork speed (boxplots) relative to the non-treated (NT) condition. HeLa cells were treated with the indicated NAD⁺ concentrations for 24h.

Effects of increasing NAMPT inhibitor or NAD⁺ concentrations on cell proliferation and ATP levels. Shown in the manuscript as Fig. 2a. Relative ATP level and cell number to non-treated cells for each cell line. The indicated cell lines were treated with the indicated concentrations of NAMPTi or NAD⁺ for 48h (HeLa and U2OS) or 72h (BJ and BJ-T). Smoothed conditional means and CI95 are shown, $n > 4$.

Those findings are some of our new contributions to the topic of how basal metabolism modulates cell proliferation and DNA synthesis.

Referee #3 cited a very interesting publication PMID: 4310516 from the 60s that he/she thought may question the novelty of our manuscript. The article PMID: 4310516 entitled “Replication and properties of DNA in nicotinamide adenine dinucleotide deficiency of Escherichia coli cells” from 1969 is very important but not directly related to our manuscript. In that paper, Nozawa and Mizuno described that the DNA ligase from E. coli needs NAD⁺ as a cofactor to ligate replication intermediates. Most eukaryotic DNA ligases, together with archaeal and bacteriophage enzymes, use ATP as a cofactor (Martin and MacNeill, 2002), which is distinct from E. coli bacteria studied in the above article mentioned by the reviewer.

The requirement of NAD⁺ to ligate replication intermediates has likely evolved to satisfy the requirement of PARylation in the maturation of Okazaki fragments in eukaryotic cells. Due to its historical value, we cited Nozawa and Muzino’s publication in the new version of our manuscript, however, it does not diminish the value or novelty of our dataset.

We have now addressed all the concerns of referee#3, and we very much hope that she/he would agree with the relevance of our study and the advance it provides to the field.

• Much of the data are descriptive, lacking in-depth mechanistic studies. E.g, the authors have noticed that NAD⁺ regulates nucleotide levels, but the underlying molecular mechanisms are elusive. One possibility could be a feed-forward loop between NAD⁺ and amino acids generation via autophagy (PMIDs: 18296641; PMID: 24813611); a recent paper also showed that autophagy regulates NAD⁺ (PMID: 36413951).

We welcome this comment from referee #3. In response, we now provide an extensive amount of new results to support our model(s) (Extended Data Fig. 10).

The induction of autophagy by treating cells with rapamycin did not show an effect on DNA synthesis and did not modify the effect of NAD⁺ treatment. Confirming unrelated mechanisms, the combination of BPTES and rapamycin did not affect the incorporation of EdU, however, there was a slight increase

in the gammaH2AX foci formation, indicating a modestly enhanced DNA damage under such combined treatment.

Effects of NAD⁺ treatment are independent of mTOR activity. QIBC analysis of EdU incorporation (left) and γH2AX levels (right) in EdU-positive cells. HeLa cells were treated for 24h without (NT) or with 1 μM rapamycin (Rapa), 10 μM BPTES and/or the indicated NAD⁺ concentrations.

We then assessed by immunoblotting the LC3 activation, but, we did not observe differences in cells that were treated with NAD⁺. These experiments suggested that autophagy is not involved in the phenotypes triggered by NAD⁺ treatment. We included these results in this rebuttal but not in the current version of our manuscript due to space limitations.

LC3 detection following NAD⁺ treatment. Cells were treated with 2 mM of NAD⁺ for the indicated times. LC3 was detected using the method described in our previous paper by Vanzo et al. (2020).

Another interesting idea suggested by Referee #3 is that autophagy could serve as a feedback loop to provide amino acids. Our results suggested that amino acids are not replenished after treating the cells with NAD⁺, on the contrary, they may be depleted (Fig. 3a). However, and conclusively, our unbiased metabolomics analyses did not suggest amino acid metabolism to be the targeted mechanism by NAD⁺ treatment, in contrast to the observed pyrimidine depletion and purine accumulation.

Autophagy, specifically mitophagy, is required to maintain healthy mitochondria and if this organelle is damaged, it should be cleared. An early step in the defective mitochondria is the drop in NAD(H) and some of the precursors of NAD(H) or the inhibition of PARP can restore the mitochondrial levels of NAD(H) (Fang et al., 2014; Kataura et al., 2022). The cellular changes triggered by the NAD⁺, at least at the time we tested, seem not related to the autophagic processes. We find that NAD⁺ directly boosts mitochondrial activity, but not its precursors, and we provide experimental evidence and discuss this phenomenon.

• When working on metabolism (especially mitochondrial metabolism) and DNA replication, it is necessary to distinguish cancer-like-immortalized cell system and the normal cellular system. The majority of the data of this paper were from cancer (cancer-like) lines (although a small portion of data were repeated in primary cells), one should be extremely cautious in data interpretation. The used cell lines are majorly cancer cells: U2OS is from a moderately differentiated sarcoma, HeLa is from cervical cancer cells, while U87 MG is original from malignant gliomas. BJ (CRL-2522) is from skin taken from normal foreskin from a neonatal male. This cell line, although from a normal individual, shows signs of ‘immortalization’ (and likely cancer-like feature), including telomerase negative (although could lead to accelerated ageing) and ‘the capacity to proliferate to a maximum of 72 population doublings before the onset of senescence’ (<https://www.atcc.org/products/crl-2522>); for normal human fibroblasts, the average passages numbers are around 30-40 when entering to senescence (see papers from Campisi lab etc). Although BJ-tert cell line is used, it is important to use multiple primary cell lines to validate the data from cancer cell lines, if the authors want to generalize their findings to the normal cells.

We fully agree with the referee on this point. It seems the effect of NAD⁺ treatment on the replication fork speed and stability is a general feature of mammalian cells. We have now included also more primary cell types, MRC5 and MEFs, and also more examples of cervical cancer cell lines (Fig. 1f and results included only in this rebuttal).

Replication sensitivity to NAD⁺ treatment. Shown in the manuscript as Fig. 1f. Replication fork speed in the indicated cell lines treated without or with 2 mM NAD⁺ for 24h. Mean±SD, n>500 fibers per condition per cell line.

DNA synthesis sensitivity to NAD⁺ treatment and NAMPT inhibition in different cervical cancer cell lines. CaSki, SiHa and HeLa cells were treated as indicated for 24h without (Ctrl) or with 10 nM NAMPT inhibitor or 2 mM NAD⁺ followed by EdU pulse-labelling for 30 min. Flow cytometry analysis was then used for quantifying EdU incorporation (y-axis) and Hoechst intensity (DNA content, x-axis).

Collectively, the authors are applauded for a large amount of informative data generated. This paper may fit better for a cancer journal. In view of the importance of NAD⁺ in proliferation and chemotherapy-induced resilience, data from this paper may be more interested to cancer researchers.

We believe that NCB is the right choice to publish our findings, considering the multiple improvements that we have made in the revised manuscript. NCB has published multiple articles with a similar scope to ours, as well as many studies of cancer cells.

Reviewer #4 (Remarks to the Author): NAD⁺ homeostasis

Summary:

Munk et. al., aimed to address if the levels of NAD(H) had an impact on DNA damage response and/or genomic DNA synthesis. These processes are critical for maintaining homeostasis and genomic integrity. Most importantly, NAD⁺ is a key redox cofactor that serves as a key substrate for PARP enzymes, which regulate genomic DNA synthesis. The authors were able to show that increasing NAD⁺ concentrations led to increased mitochondrial activity and DNA synthesis. However, prolonged NAD⁺ elevation triggered a reduction in pyrimidine biosynthesis and cell cycle arrest. This observation is critical as we continue to understand how to utilize NAD⁺ supplements for various aged-related diseases, and how increase NAD⁺ can have negative feedback on organismal homeostasis. The authors proposed a model that showcases and reconcile

how NAD⁺ controls mitochondrial function, DNA replication, and cell proliferation—with the hope of explaining how NAD⁺ metabolism modulates basal metabolism and DNA synthesis.

We are grateful to Referee #4 for his/her support and enthusiasm for our work.

Comments/Questions:

- Key literature is not cited and discussed in the current state of the manuscript. This is misleading to the audiences who will read and use this work as a resource. Please be sure to include relevant citations that span across the field.

We have now checked to make sure that the references are correctly used and now have added some additional relevant references. Given a concise letter format, we had to reduce the number of references, we apologize for those that were not included in the previous version of our manuscript.

- Figures throughout the paper can be labeled better and more properly to help the reader follow along with the authors. Recommend adding titles to each figure for feasibility.

We have added titles to each figure and re-formatted the figure configuration accordingly to address all concerns pointed out by the 4 referees.

- For the global metabolomics—the authors should preform FDR correction on their results to ensure no false discoveries are identified

The FDR correction has now been performed, as suggested by the reviewer, in all metabolomics analyses and significance was indicated by asterisks in the heatmaps to aid the reader (see M&M).

- One limitation is that this paper only represents cell lines. It's understood that NAD⁺ metabolism drastically changes between in vitro and in vivo conditions (PMID: 29685734). It would be of great interest to test this hypothesis/model in vivo to confirm or discuss the limitations in the discussion if unable to perform this key experiment.

We agree with this comment from referee #4. We only characterized the effect of a surplus of NAD⁺ using cellular models and in vivo experiments could be highly informative. Intravenous injections of NAD⁺ are available for clinical use, nevertheless, this anti-aging treatment has not received FDA approval. Direct infusions of NAD⁺ also are offered as an alternative to treat many addictions including those to alcohol and heroin.

Norris Mestayer, Paula. **Addiction: The Dark Night of the Soul/ NAD⁺: The Light of Hope.** Balboa Press. 2019.

Because an injectable solution of NAD⁺ is already available in the market, it is feasible to think that the real test is to characterize its effect directly in humans.

We found a promising combination between NAD⁺ and 5-FU to be used in cancer therapy. We are currently about to initiate preclinical studies on XPD-cancer models and plan clinical trials to treat highly proliferative cancers, such as glioblastoma.

We have chosen to publish our data in a letter format in NCB now to inspire more clinical trials and to test our concepts in several different animal models of cancer and aging. These experiments are time-

consuming and costly, therefore, we will prefer to communicate our research as soon as possible to have the most impact as possible in the related research field.

- The authors are relying on NAD/NADH assays kit measurements to quantify NAD⁺. This is extremely limiting for their observations, their findings, and many additional reasons. Combining liquid chromatography and mass spectrometry is key for accurately and precisely quantitating NAD(H) levels. The conclusions are too strong for using a kit and not LCMS.

We have used both approaches to measure NAD(H). NAD(H) levels were quantified as mentioned by the reviewer in all metabolomics data sets. This is an example included in Extended Data Figure 2a, where the measured relative areas have been normalized to that of the non-treated condition and presented as log₂ fold changes.

Metabolomics quantification of NAD⁺ and NADH levels after NAD⁺ treatment. Shown in the manuscript as Extended Data Fig. 2a-b. Metabolomics analysis of NAD⁺ and NADH levels relative to non-treated (NT) cells. HeLa and U2OS cells were treated without or with 2 mM NAD⁺ for the indicated time. n=4, Welch's ANOVA test.

In summary, the authors have revealed potential mechanisms that link NAD to modulating basal metabolism and DNA synthesis. However, prolonged NAD⁺ elevation does trigger negative downstream events. Interestingly, the authors propose how NAD-supplements could indeed be a safer option for increasing intracellular NAD⁺ without the negative impacts (triggering DDR, etc.).

We thank this comment from referee#4 and we hope our answers will convince other referees about the relevance and quality of our data sets.

References mentioned in this rebuttal document

Alano, C.C., Ying, W., and Swanson, R.A. (2004). Poly(ADP-ribose) polymerase-1-mediated cell death in astrocytes requires NAD⁺ depletion and mitochondrial permeability transition. *J Biol Chem* 279, 18895-18902.

Bartkova, J., Horejsi, Z., Koed, K., Kramer, A., Tort, F., Zieger, K., Guldborg, P., Sehested, M., Nesland, J.M., Lukas, C., *et al.* (2005). DNA damage response as a candidate anti-cancer barrier in early human tumorigenesis. *Nature* 434, 864-870.

Bartkova, J., Rezaei, N., Lontos, M., Karakaidos, P., Kletsas, D., Issaeva, N., Vassiliou, L.V., Kolettas, E., Niforou, K., Zoumpourlis, V.C., *et al.* (2006). Oncogene-induced senescence is part of the tumorigenesis barrier imposed by DNA damage checkpoints. *Nature* 444, 633-637.

Bester, A.C., Roniger, M., Oren, Y.S., Im, M.M., Sarni, D., Chaoat, M., Bensimon, A., Zamir, G., Shewach, D.S., and Kerem, B. (2011). Nucleotide deficiency promotes genomic instability in early stages of cancer development. *Cell* 145, 435-446.

Billington, R.A., Travelli, C., Ercolano, E., Galli, U., Roman, C.B., Grolla, A.A., Canonico, P.L., Condorelli, F., and Genazzani, A.A. (2008). Characterization of NAD uptake in mammalian cells. *J Biol Chem* 283, 6367-6374.

Blow, J.J., Ge, X.Q., and Jackson, D.A. (2011). How dormant origins promote complete genome replication. *Trends Biochem Sci* 36, 405-414.

Braidy, N., Villalva, M.D., and van Eeden, S. (2020). Sobriety and Satiety: Is NAD⁺ the Answer? *Antioxidants (Basel)* 9.

Bruzzone, S., Guida, L., Zocchi, E., Franco, L., and De Flora, A. (2001). Connexin 43 hemi channels mediate Ca²⁺-regulated transmembrane NAD⁺ fluxes in intact cells. *FASEB J* 15, 10-12.

Bruzzone, S., Moreschi, I., Guida, L., Usai, C., Zocchi, E., and De Flora, A. (2006). Extracellular NAD⁺ regulates intracellular calcium levels and induces activation of human granulocytes. *Biochem J* 393, 697-704.

Buonvicino, D., Ranieri, G., Pittelli, M., Lapucci, A., Bragliola, S., and Chiarugi, A. (2021). SIRT1-dependent restoration of NAD⁺ homeostasis after increased extracellular NAD⁺ exposure. *J Biol Chem* 297, 100855.

Cambronne, X.A., Stewart, M.L., Kim, D., Jones-Brunette, A.M., Morgan, R.K., Farrens, D.L., Cohen, M.S., and Goodman, R.H. (2016). Biosensor reveals multiple sources for mitochondrial NAD(+). *Science* 352, 1474-1477.

Canto, C., Houtkooper, R.H., Pirinen, E., Youn, D.Y., Oosterveer, M.H., Cen, Y., Fernandez-Marcos, P.J., Yamamoto, H., Andreux, P.A., Cettour-Rose, P., *et al.* (2012). The NAD(+) precursor nicotinamide riboside enhances oxidative metabolism and protects against high-fat diet-induced obesity. *Cell Metab* 15, 838-847.

Canto, C., Menzies, K.J., and Auwerx, J. (2015). NAD(+) Metabolism and the Control of Energy Homeostasis: A Balancing Act between Mitochondria and the Nucleus. *Cell Metab* 22, 31-53.

Davila, A., Liu, L., Chellappa, K., Redpath, P., Nakamaru-Ogiso, E., Paoletta, L.M., Zhang, Z., Migaud, M.E., Rabinowitz, J.D., and Baur, J.A. (2018). Nicotinamide adenine dinucleotide is transported into mammalian mitochondria. *Elife* 7.

Diehl, F.F., Miettinen, T.P., Elbashir, R., Nabel, C.S., Darnell, A.M., Do, B.T., Manalis, S.R., Lewis, C.A., and Vander Heiden, M.G. (2022). Nucleotide imbalance decouples cell growth from cell proliferation. *Nat Cell Biol* 24, 1252-1264.

Fang, E.F., Scheibye-Knudsen, M., Brace, L.E., Kassahun, H., SenGupta, T., Nilsen, H., Mitchell, J.R., Croteau, D.L., and Bohr, V.A. (2014). Defective mitophagy in XPA via PARP-1 hyperactivation and NAD(+)/SIRT1 reduction. *Cell* 157, 882-896.

Kataura, T., Sedlackova, L., Otten, E.G., Kumari, R., Shapira, D., Scialo, F., Stefanatos, R., Ishikawa, K.I., Kelly, G., Seranova, E., *et al.* (2022). Autophagy promotes cell survival by maintaining NAD levels. *Dev Cell* 57, 2584-2598 e2511.

Kory, N., Uit de Bos, J., van der Rijt, S., Jankovic, N., Gura, M., Arp, N., Pena, I.A., Prakash, G., Chan, S.H., Kunchok, T., *et al.* (2020). MCART1/SLC25A51 is required for mitochondrial NAD transport. *Sci Adv* 6.

Koundrioukoff, S., Carignon, S., Techer, H., Letessier, A., Brison, O., and Debatisse, M. (2013). Stepwise activation of the ATR signaling pathway upon increasing replication stress impacts fragile site integrity. *PLoS Genet* 9, e1003643.

Li, J., K, M.S., Ibrahim, M., Zeng, X., McClellan, S., Angajala, A., Beiser, A., Andrews, J.F., Sun, M., Koczor, C.A., *et al.* (2021). NAD(+) bioavailability mediates PARG inhibition-induced replication arrest, intra S-phase checkpoint and apoptosis in glioma stem cells. *NAR Cancer* 3, zcab044.

Luongo, T.S., Eller, J.M., Lu, M.J., Niere, M., Raith, F., Perry, C., Bornstein, M.R., Oliphint, P., Wang, L., McReynolds, M.R., *et al.* (2020). SLC25A51 is a mammalian mitochondrial NAD(+) transporter. *Nature* 588, 174-179.

Martin, I.V., and MacNeill, S.A. (2002). ATP-dependent DNA ligases. *Genome Biol* 3, REVIEWS3005.

Maya-Mendoza, A., Moudry, P., Merchut-Maya, J.M., Lee, M., Strauss, R., and Bartek, J. (2018). High speed of fork progression induces DNA replication stress and genomic instability. *Nature* 559, 279-284.

Maynard, S., Hall, A., Galanos, P., Rizza, S., Yamamoto, T., Gram, H.H., Munk, S.H.N., Shoaib, M., Sorensen, C.S., Bohr, V.A., *et al.* (2022). Lamin A/C impairments cause mitochondrial dysfunction by attenuating GGC1alpha and the NAMPT-NAD+ pathway. *Nucleic Acids Res* 50, 9948-9965.

Nikiforov, A., Dolle, C., Niere, M., and Ziegler, M. (2011). Pathways and subcellular compartmentation of NAD biosynthesis in human cells: from entry of extracellular precursors to mitochondrial NAD generation. *J Biol Chem* 286, 21767-21778.

O'Dwyer, P.J., King, S.A., Hoth, D.F., and Leyland-Jones, B. (1987). Role of thymidine in biochemical modulation: a review. *Cancer Res* 47, 3911-3919.

Pirinen, E., Canto, C., Jo, Y.S., Morato, L., Zhang, H., Menzies, K.J., Williams, E.G., Mouchiroud, L., Moullan, N., Hagberg, C., *et al.* (2014). Pharmacological Inhibition of poly(ADP-ribose) polymerases improves fitness and mitochondrial function in skeletal muscle. *Cell Metab* 19, 1034-1041.

Szczepanowska, K., Senft, K., Heidler, J., Herholz, M., Kukat, A., Hohne, M.N., Hofsetz, E., Becker, C., Kaspar, S., Giese, H., *et al.* (2020). A salvage pathway maintains highly functional respiratory complex I. *Nat Commun* 11, 1643.

Weisman, G.A., Lustig, K.D., Lane, E., Huang, N.N., Belzer, I., and Friedberg, I. (1988). Growth inhibition of transformed mouse fibroblasts by adenine nucleotides occurs via generation of extracellular adenosine. *J Biol Chem* 263, 12367-12372.

Ying, W., Garnier, P., and Swanson, R.A. (2003). NAD+ repletion prevents PARP-1-induced glycolytic blockade and cell death in cultured mouse astrocytes. *Biochem Biophys Res Commun* 308, 809-813.

Decision Letter, first revision:

Message: Dear Professor Bartek,

Thank you for your email asking us to reconsider our decision on your manuscript, "NAD+ regulates nucleotide metabolism and genomic DNA replication". We are always willing to hear the authors' perspective, but we must first prioritize decisions on new submissions. We appreciate your patience while we considered this appeal.

I have now discussed your manuscript, the referees' comments, and your rebuttal in detail with my colleagues, and we are willing to seek input from the reviewers again (the original experts if available), provided that nothing similar is accepted for publication at Nature Cell Biology or published elsewhere in the meantime. For re-review, please upload your revised manuscript along with the following files:

- On resubmission, please provide the completed Editorial Policy Checklist (found here <https://www.nature.com/documents/nr-editorial-policy-checklist.pdf>), and Reporting Summary (found here <https://www.nature.com/documents/nr-reporting-summary.pdf>).

This is essential for reconsideration of the manuscript and these documents will be available to editors and referees in peer review. For more information, see below. Please also ensure that the presentation of statistical information in the revised submission complies with Nature Cell Biology's statistical guidelines (see below).

Please use the link below to submit the complete manuscript files and please include a point-by-point response to the complete reviewer comments, verbatim as provided in their reports.

[redacted]

Please let us know how you wish to proceed and when we can expect your revised manuscript. Thank you for considering NCB for this work,

With kind regards,

Melina

Melina Casadio, PhD
Senior Editor, Nature Cell Biology

ORCID ID: <https://orcid.org/0000-0003-2389-2243>

GUIDELINES FOR EXPERIMENTAL AND STATISTICAL REPORTING

REPORTING REQUIREMENTS – To improve the quality of methods and statistics reporting in our papers we have recently revised the reporting checklist we introduced in 2013. We are now asking all life sciences authors to complete two items: an Editorial Policy Checklist (found here <https://www.nature.com/documents/nr-editorial-policy-checklist.pdf>) that verifies compliance with all required editorial policies and a reporting summary (found here <https://www.nature.com/documents/nr-reporting-summary.pdf>) that collects information on experimental design and reagents. These documents are available to referees to aid the evaluation of the manuscript. Please note that these forms are dynamic 'smart pdfs' and must therefore be downloaded and completed in Adobe Reader. We will then flatten them for ease of use by the reviewers. If you would like to reference the guidance text as you complete the template, please access these flattened versions at <http://www.nature.com/authors/policies/availability.html>.

Author Rebuttal, first revision:

Responses to Reviewers' comments:

Before providing our specific point-by-point responses to the comments raised by each of the four reviewers, we wish to express our appreciation for the time and effort that you used to provide your expert opinion and guidance for our further work on this study. As you will see from the revised version of our manuscript and this rebuttal document, we have addressed every point raised, in the majority of cases by new sets of experiments inspired by the constructive criticisms and suggestions from the reviewers. Given their complementary expertise and the fact that four reviewers were consulted, it is natural that a large number of comments needed to be addressed, and despite we have also involved some new authors to help revise the manuscript, the revision has taken us some 6 months to complete. The vast majority of your comments were fully justified and very helpful to guide our new experimental efforts. As a result, we very much hope that our answers and the new data presented in this revised version of the manuscript, along with the explanation of a few misunderstandings mainly reflecting our original insufficient formulations of rationale or interpretation of some results or published literature, due to the space restrictions of the letter format, will convince the reviewers of the novelty and conclusive results, that we believe much better support and extend our original conclusions and model. Based on our own long-term experience with reviewing for NCB, Nature and other top journals, we do realize such review efforts take a lot of valuable time, so we hope the improved dataset provided in this revised manuscript will be of interest and maybe even provide some inspiration for your own research.

Among the new results and changes included in our revised manuscript in response to the reviewers' comments are the following (just to list a few important ones):

- Data on a range of NAD⁺ concentrations to show that also the much lower, more physiologically relevant concentrations recapitulate the effects presented in the original manuscript;
- Substantially deeper mechanistic insights including the role of NAD⁺ in modulation of nucleotide metabolism and activity of Dihydroorotate dehydrogenase, the key enzyme that links mitochondrial bioenergetics and pyrimidine synthesis;
- An extended spectrum of cell lines to further support the concept;
- Better insights into how the mechanism we describe here affects the fundamental process of DNA replication;
- Extended full metabolomics analyses to justify the focus on nucleotides;
- Explanations of why the published literature does not undermine the novelty of our findings, and what is the conceptual advance that our study provides to the field;

The major changes in the text addressing referees' comments have been highlighted in green.

Reviewer #1 (Remarks to the Author): nucleotide metabolism

In this manuscript, Munk et al., set out to define how NAD levels impact metabolism and DNA damage response (DDR). The authors show that supplementation of cells with high levels of exogenous NAD results in an increase in ATP levels and DNA synthesis short term, while sustained high levels of NAD reduce cell proliferation, possibly due to unbalanced nucleotide pools. While the authors have performed a comprehensive analysis of the metabolic outputs and DDR in response to exogenous NAD supplementation, the latter represents a major flaw in this study given that NAD is not cell-

permeable, and to date there are no known NAD transporters at the plasma membrane. Therefore, it becomes difficult to interpret the results not knowing how NAD regulates NAD or ATP levels, or nucleotide imbalance. Also, the manuscript does not present a major advance in the field due to published work illustrating links between nucleotide imbalance or NAD levels and replication stress (PMIDs: 35927450; 34806016).

We appreciate this referee's comments and we address all his/her concerns below, at the same time explaining the changes we have made in the revised version of our manuscript.

*Referee #1 is correct, NAD⁺ is not cell membrane-permeable, however, while the specific membrane transporter(s) have remained largely enigmatic, mammalian cells can transport exogenous NAD⁺ into the cytoplasm (Billington et al., 2008; Bruzzone et al., 2001; Bruzzone et al., 2006; Buonvicino et al., 2021). NAD⁺ can be imported through the purinergic receptor P2X7 (Alano et al., 2004) or the connexin 43 channels (Bruzzone et al., 2001). We included these important references in our manuscript. Given that the access of NAD⁺ to intracellular space is known, we aimed to study its impact on important cellular processes, rather than identifying the precise mechanism of its cell entry, which is another topic, not investigated in our present study. Experimentally, we tested whether exogenous NAD⁺ or some of its precursors could influence mitochondrial functions and we found that only intact NAD⁺ could rapidly alter mitochondrial activity.

Our results are consistent with the notion that the mitochondrial NAD⁺ pool can be established through the direct import of NAD⁺ (Davila et al., 2018), using the mitochondrial-specific NAD⁺ transporter SLC25A51 (Kory et al., 2020; Luongo et al., 2020). We interpreted our results as the most likely explanation for how NAD⁺ regulates mitochondrial functions that impact nucleotide biosynthesis. In the revised version we incorporated multiple new results to better justify our conclusions.

The mitochondrial pool of NAD(H), of course, can be synthesized from precursors which were shown before and reported by Ziegler's laboratory some years ago (Nikiforov et al., 2011). Of relevance, over a decade later, Zeigler co-authored a related paper showing the existence of the NAD⁺ transporter in the mitochondria (Luongo et al., 2020). Both models are correct and co-exist, and our new data support this concept. We integrated into our model both of these possibilities (see Extended Data Fig. 10a in the new version of our manuscript).

*We do not share the view that our dataset does not present a major advance in the field. Indeed, our present work is likely to considerably advance the current knowledge of how cells regulate genomic DNA synthesis and its interplay with mitochondrial functions. Furthermore, since treatment with NAD⁺ and its precursors are used in the literature interchangeably to manipulate intracellular NAD⁺ levels in cultured cells, our new findings, that NAD⁺ treatment impairs the synthesis of pyrimidine nucleotides with the concomitant accumulation of purine nucleotides, will lead to re-interpretation of published studies in which exogenous NAD⁺ was used as a mean to increase cellular NAD⁺: This important shift in viewing those previous studies is inevitable given that the authors of those papers were unaware of the previously

uncharacterized effect of NAD⁺ treatment on nucleotide levels (discovered and reported in our present manuscript) when drawing their conclusions about NAD⁺ effects.

* At the end of his/her comments, Referee #1 included two very good recent publications, which are however not closely overlapping with the cellular events studied in our present research and our manuscript. Vander Heiden laboratory PMID: 35927450 (Diehl et al., 2022) reported that nucleotide imbalance decouples cell growth from proliferation: it is indeed a good manuscript that strengthens the relevance of our new, complementary results, and we cite this work in the revised manuscript. Nucleotide imbalances are known as a source of replication stress (Bester et al., 2011), and the Diehl et al. study shows that specifically the imbalances of nucleosides are sensed by the ATR-dependent signalling pathway, and this observation is probably the main aspect of the novelty of their manuscript. In the Diehl et al manuscript, there is no experimental evidence of how nucleotide imbalances can be naturally achieved, nor any molecular mechanistic insights into the cross-regulation between the nucleotide pools. Furthermore, Diehl et al. showed that supra-physiological levels of guanine reduced the levels of adenylate nucleotides and inhibited cell proliferation. After the addition of guanine, GTP was accumulated. In our case, the addition of NAD⁺ did not induce the accumulation of GTP, indicating that the new mechanism of regulation of the nucleotide metabolism that we report here is different from that reported by Diehl et al.

Furthermore, our manuscript addresses an entirely different set of important questions, such as How do the levels of NAD⁺ impact RNA/DNA synthesis. While the Diehl et al paper studied the effects of treatment with nucleotide precursors, we demonstrate that extended NAD⁺ treatment directly modulates the mitochondrial metabolic activity with downstream consequences for DHODH functionality, causing the impairment of dihydroorotate to orotate conversion in *de novo* pyrimidine synthesis and, ultimately, depletion of pyrimidine nucleotides and accumulation of purines. This mechanism is not described in the aforementioned study or any other published work, and thereby represents our new, original findings that we now present in this NCB manuscript. We hope that these newly generated mechanistic results will also help to highlight the novelty of our revised manuscript. We also describe potential metabolic vulnerabilities in cancer cells that could be exploited in therapy, again an important issue not addressed by the papers mentioned by the reviewer.

The article PMID: 34806016 published in NAR cancer (Li et al., 2021) is even less related to our present manuscript, but it is more related to our work that we published in Nature some 5 years ago (Maya-Mendoza et al., 2018) where we reported the impact of PARylation on speed of replication fork progression and the different scenarios under which replication stress occurs, overall extending the concept of the major role of replication stress in genomic instability and tumorigenesis, a research direction that we pioneered almost 20 years ago in our two Nature papers by Bartkova et al. (Bartkova et al., 2005; Bartkova et al., 2006) and which has since then grown into an entire new field of biomedical research pursued by many laboratories. The Li et al. study is related to PARG inhibition and the accumulation of Poly-ADP-ribosylation (PARylation) in glioma cells. The authors combine PARG-inhibitors with dihydronicotinamide riboside (NRH), not surprisingly, cells treated with this combination accumulate PARylation that, as we discovered and reported in 2018 (Maya-Mendoza et al) impairs replication fork

progression. Again, this work by Li et al. is unrelated to our present manuscript and we hope that the revised version of our manuscript and this rebuttal help to clarify this point.

Major comments:

1) The concentrations of NAD used throughout the study are supraphysiological. Moreover, there are no known NAD transporters at the plasma membrane, raising the question of whether NAD signals through the purinergic receptors, similar to other purine nucleotides. Does exogenous NAD enter the cells after being converted to NMN or NR?

We thank the referee for pointing out this previous weakness in our story. We agree that 2 mM is a supraphysiological concentration. We initially supplemented cells with 2 mM of NAD⁺ for two reasons; first, NAD⁺ at the millimolar range has been reported to boost intracellular NAD⁺ levels (Bruzzone et al., 2001; Buonvicino et al., 2021; Maynard et al., 2022; Ying et al., 2003). Second, when we tested the effect of different concentrations of NAD⁺ on cell viability, these results were included in Fig 2a and Extended Data Fig. 3a in the first submission, 2 mM of NAD⁺ caused a notable peak of ATP increase with the arrest of cell proliferation.

Inspired by this reviewer's comment, we now present extensive evaluations of the effects of NAD⁺ treatment at a wide range of concentrations, at 16 μ M – 10 mM, on cell physiology. Interestingly, the effects of NAD⁺ treatment seem to be induced already at around 50 μ M, which is a physiologically relevant concentration, taking into account that the concentration of NAD⁺ in mitochondria ranges between 50-250 μ M (Cambronne et al., 2016) and that the Km for NAD⁺ transport by SLC25A51 is approximately 200 μ M (Luongo et al., 2020).

For an easier evaluation of this newly added large dataset, we include here below the results of the metabolomics analysis for different concentrations of exogenous NAD⁺. Consistent with our previous findings the pyrimidine depletion is achieved in experiments with the physiologically relevant 50-100 μ M concentrations of NAD⁺.

NAD⁺ treatment induces nucleotide imbalances. Shown in manuscript as Fig. 2c. Heatmap of metabolite levels from metabolomics analysis in HeLa cells treated with the indicated NAD⁺ concentrations for 24h. Asterisks indicate significant changes ($p\text{-adj.} < 0.05$) to the non-treated (NT) condition.

NAD⁺ degradation products, such as nicotinamide riboside and AMP, did not accumulate in the cell at these concentrations. However, since we observed differential effects by treatment with NAD⁺ and its precursors on genomic DNA synthesis, this is consistent with the notion that such effects of NAD⁺ are direct and not downstream of NAD⁺ degradation into its precursors. Furthermore, since the effects of AMP treatment did not phenocopy those of NAD⁺ treatment, this also indicates that extracellular degradation of NAD⁺ to NMN and AMP does not activate purinergic receptors responsible for the observed effects of NAD⁺ treatment on mitochondria.

2) Fig. 1: NR or NAD⁺ treatment increases cellular NAD(H) level, but only NAD⁺ treatment induces DDR response and inhibits DNA synthesis. Why is this response NAD specific? Are purinergic receptors or CD38 involved in metabolizing NAD and inducing downstream signaling pathways (Ca²⁺ or C-ADPR)?

Indeed, one of the main findings presented in our manuscript is the differential response of the cell to treatment with NAD⁺ or its precursors. The conversion to NAD⁺ from precursors is determined by the level of expression of the relevant enzymes and the availability of the substrates. NAD⁺ supplementation directly alters mitochondrial activity. Certainly, the treatment of cells with NR did not increase the mitochondrial membrane potential (Extended Data Figure 8b) in contrast to NAD⁺ treatment, and only treatment with NAD⁺ was able to induce an increase in ATP levels. The increase in the concentration of intracellular NAD⁺ from the precursors such as NR or NMN seems not to alter mitochondrial functions. The excess of NAD⁺ originating from precursors could be buffered by the increase of mitochondrial mass (Canto et al., 2012) or possibly by some additional regulatory mechanism(s) that are currently unknown.

Here, we have tested experimentally the potential involvement of CD38 and intracellular calcium signaling. Effects of NAD⁺ treatment were unaltered by co-treatment with 1 μ M CD38 inhibitor for 24h, indicating an involvement of CD38-independent independent mechanism(s). We appreciate this comment from referee#1 and include these results in the revised version of our manuscript (see below and Extended Data Fig. 2n, o).

Effects of NAD⁺ treatment are independent of CD38 activity. Shown in the manuscript as Extended Data Fig. 2n-o. **Left panel:** QIBC analysis of γ H2AX foci during cell cycle in cells treated for 24h without (NT) or with 1 μ M CD38i and/or 2 mM NAD⁺. **Right panel:** Relative intracellular calcium changes measured with the Fluo-8 indicator in cells after treatment as in the left panel, mean+SD, n=3.

3) How do NAD⁺ levels increase the levels of purines, but decrease pyrimidines (Fig 2)??. Since the metabolite patterns between HeLa and U2OS are different, more cellular contexts are required to understand whether these phenotypes are generalizable.

We appreciate that the reviewer finds these results new and surprising, and we fully agree that the examination of more cellular models is helpful in this context. We have now tested a panel of different cell types and all of them exhibit replication sensitivity to NAD⁺ treatment (Fig. 1f). A natural cause for the variation in sensitivity between cell types could be due to the pre-existent levels of nucleotide pools and the potential for anaplerosis of TCA cycle intermediates. Indeed, we observed a higher potential for anaplerosis in metabolomics analysis of U2OS cells compared to HeLa cells (Fig. 3a), and the higher pyrimidine levels maintained in U2OS cells compared to HeLa cells (Fig. 2e) could further provide them with the observed increased resistance to DHODH inhibitor-induced pyrimidine synthesis impairment (Fig. 4g).

The accumulation of purines after the incubation of cells treated with NAD⁺ is a later event that follows pyrimidine depletion. There can be two possible scenarios for the accumulation of purines; 1) NAD⁺ treatment only depletes pyrimidines, thus purines will accumulate since they are not used for DNA

synthesis and 2) NAD⁺ treatment depletes pyrimidines and, e.g. through its hydrolysis to NMN and AMP, elevates AMP levels, which can also be converted to GMP, and can thereby also function as a source to increase the pool of purine nucleotides. As suggested by this reviewer, In this new version of our manuscript, we have included additional cell types (Fig. 1f) and we have improved the analysis and presentation of the new and old metabolomic data (Figs. 2, 3).

4) Line 253: How does NAD⁺ regulate the activity of the TCA cycle and ETC to regulate DNA replication? In extended fig 3b, the TCA cycle metabolites fumarate and malate decrease in HeLa cells upon NAD treatment. There is not sufficient evidence to conclude that NAD regulates the DNA synthesis through “the NAD⁺ mediated-high activity of TCA cycle and ETC”

Our apology for not expressing our idea clearly in the previous version of the manuscript. We also provide more experimental evidence to support this conclusion. We stated that NAD⁺ treatment increased the activity of the TCA and ETC based on the following results:

*Treatment with NAD⁺ depleted most of the intermediates of the TCA in HeLa cells, presumably because the low endogenous levels of glutamine and aspartate in these cells were unable to sustain anaplerosis.

*After 1h of exposure to NAD⁺, the levels of UMP and uridine increased, and there was an increase in the mitochondrial activity in both U2OS and HeLa cell lines.

*The overactivation of mitochondrial function(s), specifically the TCA cycle and ETC, lead to the arrest of pyrimidine biosynthesis. Without pyrimidines, DNA synthesis cannot be sustained and exhibits higher vulnerability to such condition than the general transcription.

*Importantly, impairing TCA cycle anaplerosis and ETC proficiency using the inhibitors BPTES and oligomycin, respectively, rescued the replication defects caused by NAD⁺ treatment, directly demonstrating that TCA cycle and ETC activity are necessary for NAD⁺ induced i) hyperpolarization of the mitochondrial membrane potential, and ii) depletion of pyrimidines that are essential for genomic DNA replication and transcription.

In this revised version, we included new results showing that the DHODH inhibitor brequinar (BRQ), which caused inhibition of the *de novo* pyrimidine synthesis, phenocopies the effect of treatment with NAD⁺, demonstrating the NAD⁺ treatment impairs DHODH activity and that proficiency of the mitochondrial membrane potential is required for NAD⁺ effects on genomic DNA replication as observed by co-treatment with FCCP. Some of these results are shown below (and see Fig. 4 in the manuscript).

Effects of NAD⁺ treatment phenocopy DHODH inhibition and depend on mitochondrial membrane potential proficiency. Shown in the manuscript as Fig. 4d,g. **Upper panel:** Analysis of γ H2AX foci during cell cycle with QIBC in cells treated for 24h without (NT) or with NAD⁺ as indicated and/or 2 μ M brequinar (BRQ). **Lower panel:** Analysis of γ H2AX foci during cell cycle with QIBC in cells treated for 24h without (NT) or with NAD⁺ as indicated and/or 10 μ M FCCP.

5) Line 254: Extend fig 7.g, what is the evidence for the conclusions that high AMP level inhibits pyrimidine synthesis?

It has been shown and is well-established that AMP could inhibit the activity of UMPS, resulting in the depletion of UTP (Weisman et al., 1988). Our results showed that the sensitivity to AMP is also cell-type dependent. The results were previously shown in Extended Data Fig. 7. To clarify this issue, in this new version of the manuscript, we show some of the results regarding sensitivity to AMP as a part of Fig. 2 (here shown after treatment with 100 μ M and 2 mM AMP).

AMP treatment impairs DNA synthesis. Shown in the manuscript as Fig. 2h. QIBC quantification of EdU incorporation in EdU-positive cells. HeLa and U2OS cells were treated without (NT) or with the indicated AMP concentrations for 24h. $n > 1,600$ cells per condition.

To further validate the aforementioned published effects of AMP treatment for the reviewer, we co-treated cells with 2 mM of AMP and uridine for 24h, which would rescue the AMP-induced pyrimidine deficiency, and evaluated the effect thereof by cell cycle profiling, and the rescue effect of uridine on eliminating the S-phase accumulation was indeed observed (shown below and Fig. 4f).

Uridine supplementation rescues AMP-impaired S phase progression. HeLa and U2OS cells were treated for 24h without or with 2 mM AMP and/or 100 μ M uridine as indicated. Hoechst was then used to stain the DNA of the cells for cell cycle analysis by QIBC.

In this manuscript version, we included the manipulation of AMP levels (at 2 mM) in combination with the inhibitors oligomycin or BPTES, two inhibitors we found to rescue the effects of NAD^+ treatment on DNA replication. These and previous results showed that the regulation of DNA synthesis, via regulation of pyrimidine synthesis, by AMP occurs downstream of the reaction catalyzed by DHODH.

Effects of AMP treatment are independent of mitochondrial metabolic proficiency. Shown in the manuscript as Fig. 4e. QIBC analysis of γ H2AX foci during cell cycle in HeLa and U2OS cells treated without (NT) or with 2 mM AMP, 5 μ M oligomycin (Oligom), and/or 10 μ M BPTES for 24h.

Moreover, we agreed with the referee's concern and modify the sentence by saying that our experimental evidence shows that a high level of AMP reduced the incorporation of EdU incorporation (Page 10, line 213).

6) Line: 288-290: The presented data are not sufficient to conclude that the “effects of NAD⁺ depend on mitochondrial NAD⁺ transport in U2OS cells, while HeLa cells remain sensitive to NAD⁺ in the absence of SLC25A51 owing to AMP-regulated mechanisms”.

We thank the reviewer for this comment. We performed a new set of experiments to better support this conclusion.

First, our metabolome analysis, using different concentrations of NAD⁺ to treat the cells, showed that AMP starts to accumulate when NAD⁺ treatment at concentrations above 80 μ M is used.

AMP levels in response to treatment with increasing NAD⁺ concentrations. Relative quantification of AMP from metabolomics analysis in HeLa cells treated with the indicated NAD⁺ concentrations for 24h, n=4, mean±SD shown.

Therefore, we hypothesized that the effect of AMP on inhibiting the incorporation of EdU should only be seen at NAD⁺ concentrations higher than 80 μM in HeLa cells, while U2OS cells should exhibit reduced sensitivity. Indeed, this was the case (see results below). These results are presented as part of the Fig. 5c.

Furthermore, if the potential AMP-regulated mechanism is independent of the mitochondrial regulation of pyrimidine anabolism, the ablation of the mitochondrial transporter SLC25A51, and hence ablation of elevation of mitochondrial NAD⁺ following NAD⁺ treatment (see results below), should not interfere with its regulation downstream of the mitochondrial step in pyrimidine synthesis. We confirmed this prediction, with HeLa cells remaining sensitive to NAD⁺ treatment and U2OS cells losing their sensitivity after si-SLC25A51 treatment and found that the knockdown of SLC25A51 was also instrumental for maintaining genomic DNA replication.

Effect of SLC25A51 depletion on NAD⁺ treatment effects and mitochondrial NAD(H) levels. Shown in the manuscript as Fig. 5c and Extended Data Fig. 9d. **Upper panel:** QIBC analysis of γ H2AX foci during cell cycle. Cells transfected with control siRNA (CTRL) or si-SLC25A51 were treated for 24h with NAD⁺ as indicated. **Lower panel:** Quantification of total mitochondrial NAD(H) in cells treated as in upper panel but with only 2 mM of NAD⁺, NT = non-treated. n=3 (HeLa) and n=2 (U2OS), mean+SD, Student's t-test (HeLa).

7) The biological or physiological relevance of high exogenous NAD levels is unclear, given that plasma levels of NAD are in micromolar rather than millimolar range. Does supplementation of high levels of NAD *in vivo* have similar effects as presented in the two cell culture models?

Our new results showed that even a slight increase in exogenous NAD⁺ changes cell physiology. At the lowest concentration tested, 16 μ M NAD⁺, we already observed metabolic changes and altered dynamics of genomic DNA synthesis.

Replication sensitivity to low concentrations of NAD⁺ treatment. Shown in the manuscript as Fig. 1h-i. Replication fork speed in HeLa and U2OS cells treated without (NT) or with the indicated NAD⁺ concentrations for 24h. Mean \pm SD, n>500 fibers per condition per cell line.

Regarding the relevance in vivo, it seems that normal cells are more resistant to any variation in NAD⁺ levels (Fig. 5) while highly proliferative tumors that rely on the *de novo* pyrimidine synthesis can be targeted in a combined therapeutical setting.

The next phase of our research will involve clinical trials, most likely without the otherwise recommended intermediate step of testing in animal models first, because NAD⁺ is used as anti-aging therapy in humans and offered by clinics around the world. Worryingly, the concentration of NAD⁺ used in these therapies is at the millimolar level and it is injected intravenously. The typical treatment commonly uses 500–1000mg of NAD daily over 4–10 days.

<https://resetiv.com/products/nad>

<https://nadclinic.com/injections/#>

Moreover, millimolar concentrations of NAD⁺ are also used currently to treat diverse types of addictions (Braidy et al., 2020).

<https://www.springfieldwellnesscenter.com/about-nad/>

Further careful research on the effects of NAD⁺ at the cellular level is imperative to optimize the various treatments in humans, and our results contribute to the understanding of the biological impact of this fascinating molecule.

Reviewer #2 (Remarks to the Author): DNA replication stress, PARP

In this manuscript, the authors investigated how NAD⁺ levels affect DNA synthesis and genomic stability. They found that exogenous NAD⁺ inhibits DNA synthesis and induces DNA damage. By metabolic profiling, they showed that exogenous NAD⁺ induces an imbalance of nucleotides, reduces overall transcription, and activates mTOR. Their RNA-seq analysis of NAD⁺ treated cells uncovered that ETC activity is enhanced by NAD⁺ in mitochondria. Blocking ETC by oligomycin rescued the effects of NAD⁺ on DNA synthesis. Inhibition of the conversion of glutamine to glutamate and the TCA cycle by BPTES also rescued the effects of NAD⁺ on DNA synthesis. These results suggest that the TCA cycle and ETC are important for the effects of NAD⁺ on DNA synthesis, which may be mediated by unbalanced nucleotides. The authors also noticed differences between HeLa and U2OS cells in their responses to exogenous NAD⁺. HeLa cells accumulate higher levels of AMP after NAD⁺ treatment, which may also contribute to the inhibition of DNA synthesis. Finally, the authors proposed that NAD⁺ can be combined with drugs that alter pyrimidine levels in cancer therapy.

The main finding of this study that high NAD⁺ can alter nucleotide balance and inhibit DNA synthesis is interesting. However, almost of the experiments supporting this conclusion are based on the use of a high concentration of exogenous NAD⁺. Whether this approach is relevant to any physiological or pathological conditions in aging, cancer, and therapy is unclear. Whether NAD⁺ induced changes in nucleotide balance are the actual cause of reduced DNA synthesis and induction of DNA damage is not tested directly. Whether and how enhanced ETC is responsible for the unbalanced nucleotides is also untested. The overall model of this study is still too preliminary, with major gaps to be filled. Although this study is focused on the effects of NAD⁺ on DNA synthesis and genomic stability, the metabolomics data from this study show that NAD⁺ affects many other pathways and processes. It seems to me that the impacts of NAD⁺ in cells are much more complex than what is proposed in the model, and I am not sure if this study can establish general principles as claimed.

We appreciate the comments from referee#2, they have helped us in the last 6 months to improve our manuscript. The referee raised several valid concerns, most of which reflected our poor introduction and description of our results. In this new version of our manuscript, we provide numerous new experiments and amend the text to validate and support our proposed model. Aligned with the reviewer's comments, we now demonstrate that the previously observed effects of NAD⁺ are also induced by much lower, physiologically relevant concentrations of exogenous NAD⁺. The NAD⁺ treatment-induced nucleotide imbalances, namely depletion of pyrimidines with concomitant accumulation of purines, that we also demonstrate are balanced and alleviated by impairing the activity of the TCA cycle and ETC as well as co-treating cells with uridine to fuel pyrimidine synthesis independent of mitochondrial DHODH activity. We very much hope that referee#2 will find our revised manuscript sufficiently improved.

1. In Fig. 1b and 1c, the use of SIRT1i and PARPi to study the effects of increased NAD⁺ is not completely justified. Both SIRT1 and PARP1 have many functions independent of their effects on NAD(H) levels. In Fig. 1b, it is clear that SIRT1i and PARPi only modestly increased NAD(H) levels. However, it is unclear whether the modest increase of NAD(H) in PARPi treated cells is relevant to the effects of PARPi on DNA synthesis and genomic stability.

We appreciate this comment from referee#2. Our apology for not being clear to introduce the rationale behind these experiments.

Referee #2 is correct in stating that SIRT and PARP inhibitors alter diverse processes in the cell, as one of the downstream effects of their inhibition is the increase of intracellular NAD(H). The levels of NAD(H) drop with advancing age in humans, therefore, the use of PARP inhibitors might alleviate some of the aging features (Canto et al., 2015; Pirinen et al., 2014). This may be true, however, given the role of PARPs in DNA replication and genome integrity maintenance, such beneficial anti-aging effects seem unlikely.

Our results showed that the increase in intracellular NAD(H) after the inhibition of PARP1 or SIRT1 is independent of their roles in DNA synthesis or repair. Most likely due to the reduction of PARylated proteins or altered chromatin acetylation. Indeed, the levels of intracellular NAD(H) after the inhibition of SIRT1 increased only mildly with no statistical significance, after PARP1-inhibition there was a significant increase of ~50 % (Fig. 1b), despite their shared increase of intracellular NAD(H) after the inhibition of PARP1 or SIRT1, their respective effects on DNA synthesis were entirely different as we showed by EdU incorporation (Fig. 1d in the new version of our manuscript) and DNA fibre assay (Extended Data Figure 1d,e). Interestingly, the inhibition of NAMPT depleted 90% of the total NAD(H), with a significant negative impact in HeLa cells but not in U2OS cells.

Next, in our experiments, we tried to rescue the effect of low NAD(H) by supplementing cells with NAD⁺. These experiments revealed the unexpected results that we then further characterized and mechanistically elucidated in the subsequent parts of our manuscript.

b

Manipulation of intracellular NAD(H) levels by inhibitor treatment. Shown in the manuscript as Fig. 1b. NAD(H) quantification in cells treated without (NT) or with 10 nM NAMPTi, 2 μ M SIRT1i or 10 μ M PARPi for 24h. Mean+SD, n=3, Student's t-test.

d

Differential DNA synthesis defects induced by NAD⁺ treatment and PARP or NAMPT inhibition. Shown in the manuscript as Fig. 1d. QIBC analysis of γ H2AX foci during cell cycle in HeLa cells treated for 24h without (NT) or with 10 nM NAMPTi, 10 μ M PARPi and/or 2 mM NAD⁺.

d

e

Replication forks in U2OS cells are not affected by NAMPT inhibition. Shown in the manuscript as Extended Data Fig. 1d-e. Replication fork speed (**left panel**, mean \pm SD) and symmetry (**right panel**) in U2OS cells treated for 24h without (NT) or with 10 nM NAMPTi, 2 μ M SIRT1i or 10 μ M PARPi. n>425 forks per condition.

We corrected and extended the text to explain better the rationale behind all those experiments (Page. 4, lines 80-82).

2. In Fig. 1b, NAMPTi drastically reduced NAD(H) in both HeLa and U2OS cells, but it only reduced DNA synthesis in HeLa cells. This difference between HeLa and U2OS is never explained in the following experiments. The 1.5 fold higher NAD⁺ in U2OS is unlikely sufficient to explain the difference. The

authors completed switched to using exogenous NAD^+ to inhibit DNA replication, which is a completely different phenomenon.

Again, our apology because we did not discuss enough the differences in sensitivity to NAMPT inhibitors between HeLa and U2OS. In this new version of our manuscript, we show more experimental evidence to explain the differences in sensitivity to NAMPT inhibitors between HeLa and U2OS cells (Fig. 3).

Our results showed that in general, cells tolerate low intracellular levels of NAD(H) . They may shift to use some other redox molecules to support their basic metabolism. Our analysis of metabolites showed that HeLa cells under untreated conditions have a low level of certain TCA intermediates, specifically, upstream of the complex II-mediated oxidation of succinate (*succinic acid* below). Fumarate (*fumaric acid* below) is depleted after NAD^+ treatment in these cells (Fig. 3a and below). Because the level of glutamine and glutamate is very low in HeLa cells, the anaplerosis of the TCA via α -ketoglutarate (also known as 2-oxoglutaric acid) is also limited.

Changes in the TCA cycle intermediates succinate and fumarate after NAD^+ treatment. Crop of Fig. 3a in the manuscript. Metabolite levels from metabolomics analysis in HeLa and U2OS cells treated without (NT) or with 2 mM NAD^+ for the indicated hours. The y-axis displays relative metabolite quantification. Bars represent means, $n=4$.

Together, these metabolic features suggested that HeLa cells but not U2OS cells could have impaired complex II activity, and therefore, be more sensitive to complex I inhibition as well as the depletion of NAD(H) induced by NAMPT inhibitor. Indeed, the replication forks of HeLa cells were more sensitive to metformin, an inhibitor of complex I.

Differential replication sensitivity to metformin in HeLa and U2OS cells. Shown in the manuscript as Fig. 3c-d. Replication fork speed in HeLa and U2OS cells treated without (NT) or with 2 mM NAD⁺, 10 μM BPTES, and/or 1 mM metformin (Met) for 24h. Mean±SD, n>500 fibers per condition.

3. The NAD⁺ rescue experiments in Fig. 1d and 1e are confusing. NAD⁺ partially rescued the effects of NAMPTi in HeLa, but it acted independently of NAMPTi in U2OS cells. Since NAMPTi reduced endogenous NAD(H) in U2OS (Fig. 1b), should the effects of exogenous NAD⁺ be at least reduced by NAMPT1i?

This is a very interesting point raised by referee#2. We thought the same way as referee #2, however, our data showed that the effect of NAD⁺ in U2OS cells is dominant over the inhibition of NAMPT.

We explain this by the fact that U2OS cells tolerate a low level of NAD(H), also supported by the well-tolerated inhibition of complex I by metformin, and due to their high potential for anaplerosis via the glutamine-glutamate-αKetoglutarate axis, they can feed the ETC by complex II. When we added NAD⁺ to U2OS cells, because they rely on ETC flow to regulate the activity of the DHODH, they accumulate carbamoyl aspartate and dihydroorotate (Extended Data Figure 6a).

The partial alleviation of the effect of NAMPT inhibitors by NAD⁺ in HeLa cells is likely due to the differential regulation of nucleotide anabolism.

We clarify this phenomenon on Pag. 11.

Changes in intermediates of de novo pyrimidine synthesis after NAD^+ treatment. Crop of Extended Data Fig. 6a in the manuscript. Metabolite levels from metabolomics analysis of HeLa and U2OS cells treated without (NT) or with 2 mM NAD^+ for the indicated hours. In metabolite plots, the y-axis displays relative metabolite quantification. Bars represent means, $n=4$.

4. Adding 2mM exogenous NAD^+ to cells is a very artificial manipulation. It is not clear whether this high concentration of NAD^+ is relevant to any physiological or pathological conditions in aging, cancer, or therapy. The effects of this extremely high concentration of exogenous NAD^+ may or may not be achievable in cells under any physiological or pathological conditions. This is a major concern on the experimental approach of the current study. The authors should confirm their findings using other strategies to increase endogenous NAD^+ .

We acknowledge this comment and, as suggested by the reviewer, we have performed an extensive set of new experiments to validate our model of how NAD^+ treatment, even at lower, physically relevant concentrations, regulates nucleotide biosynthesis, genomic DNA synthesis, and cell proliferation.

We agree with this concern, we did not emphasize enough some of our results regarding different treatment concentrations of NAD^+ in the previous version of our manuscript where we tested several concentrations of NAD^+ to test its effect on cell viability and energy production (Fig. 2a). We have now revised and updated our manuscript to provide evidence that NAD^+ treatment blocks pyrimidine anabolism, inhibiting genomic DNA synthesis at concentrations as low as 80 μM , and that this effect did not change further at concentrations beyond 400 μM (Fig. 1g).

DNA synthesis sensitivity to increasing concentrations of NAD⁺ treatment. Shown in the manuscript as Fig. 1g. EdU incorporation in EdU-positive cells quantified by QIBC. HeLa cells were treated without (NT) or with the indicated NAD⁺ concentrations for 24h, n>6,000 cells per condition. P-values were calculated from comparison to the NT condition.

To allow for a stronger, better-validated interpretation of these results, we also performed the full metabolomic analysis of this experimental condition (shown now in Fig. 2c).

NAD⁺ treatment induces nucleotide imbalances. Shown in the manuscript as Fig. 2c. Heatmap of metabolite levels from metabolomics analysis in HeLa cells treated with the indicated NAD⁺ concentrations for 24h. Asterisks indicate significant changes ($p\text{-adj.} < 0.05$) to the non-treated (NT) condition.

In the previous version of our manuscript, we mainly used 2 mM of NAD⁺ because it is a concentration that has been used in recent studies to boost intracellular NAD(H) (Luongo et al., 2020; Maynard et al., 2022; Szczepanowska et al., 2020), and because our results showing that cells reach the maximum ATP production with a decrease in cell viability, a metabolic link that has never been explored in the literature.

Effects of increasing NAMPT inhibitor or NAD⁺ concentrations on cell proliferation and ATP levels. Shown in the manuscript as Fig. 2a. Relative ATP level and cell number to non-treated cells for each cell line. The indicated cell lines were treated with the indicated concentrations of NAMPTi or NAD⁺ for 48h (HeLa and U2OS) or 72h (BJ and BJ-T). Smoothed conditional means and CI95 are shown, $n > 4$.

We are confident that this revised version of our manuscript is substantially improved by the added data from the new set of experiments performed in response to the constructive criticisms and suggestions from the reviewers.

We also confirmed our results using known precursors of NAD⁺ that boost intracellular NAD⁺ levels. An interesting new result shows that only treatment with NAD⁺ directly was able to rapidly change mitochondrial activity, impact DNA replication and arrest the cell cycle.

Treatment with NAD⁺ but not its precursors impair DNA replication. Shown in the manuscript as Fig. 1l-m. **Left panel:** Replication fork speed in HeLa cells treated without (NT) or with 2 mM NAD⁺, NR or NMN for 24h. Means ± SD are shown, $n > 400$ fibers per condition. **Right panel:** QIBC analysis of γH2AX foci during cell cycle in HeLa cells treated without (NT) or with 2 mM NAD⁺, NR or NMN for the indicated hours.

5. The authors did not interpret Fig. 1m clearly. Even if high levels of NAD⁺ do not affect RRM activity, why would addition of dNTPs reduce DNA synthesis? Can this be explained by unbalanced nucleotide pools? Why?

We apologize since, due to space restrictions of the concise letter format, we left some results insufficiently described in our original manuscript.

For the deoxy-nucleoside supplementation, we wanted to test whether NAD⁺ treatment could affect directly the activity of the Ribonucleotide reductase (RNR). Because we did not know specifically which deoxy-nucleotides were the most affected by NAD⁺ treatment, we initially supplemented cells with all four deoxy-nucleosides. This supplementation reduced even further the DNA synthesis when it was combined with NAD⁺ treatment. This showed that the deoxy-nucleosides and the NAD⁺ treatment triggered different pathways that resulted in further inhibition of DNA replication. It turned out that the exogenous thymidine was responsible for this additive effect. An excess of thymidine has been used historically to arrest the cell cycle in the border G1/S of the cell cycle. In a cell-dependent manner, the excess of thymidine expands the dTTP pools and inhibits the activity of RNR with the consequent depletion of dCTP (O'Dwyer et al., 1987). We then confirmed the mutually independent nature of the molecular mechanisms triggered by NAD⁺ treatment and the inhibition of the RNR, respectively, using hydroxyurea (HU). Hydroxyurea phenocopied the effect of thymidine in the reduction of EdU incorporation (These new results are shown in Fig. 4h-k).

6. In Fig. 2, the authors found that 24 h of NAD⁺ treatment reduced pyrimidines but increased purines, among many other metabolic changes. They speculated that NAD⁺ induced imbalance of nucleotides is the cause of reduced DNA synthesis and genomic instability. However, this idea is never directly tested. This is a major weakness of the current study.

Together with new data sets, we showed that NAD⁺ treatment induces the accumulation of some purines and the depletion of pyrimidines. This phenomenon is time- and concentration-dependent.

Changes in pyrimidine and purine nucleotide levels in response to increasing NAD^+ concentrations. Shown in the manuscript as Fig. 2d. Mean nucleotide levels (bars) from metabolomics analysis and replication fork speed (boxplots) relative to the non-treated (NT) condition. HeLa cells were treated with the indicated NAD^+ concentrations for 24h.

The lack of pyrimidines is followed by a reduction in the replication fork speed, moreover, the treatment with oligomycin or BPTES alleviated the nucleotide imbalances induced by NAD^+ treatment and restored DNA replication.

Co-treatment with oligomycin or BPTES alleviates NAD^+ -induced nucleotide and replication fork impairments. Shown in the manuscript as Fig. 3g. Mean nucleotide levels (bars) from metabolomics

analysis and replication fork speed (boxplots) relative to the non-treated (NT) condition. HeLa cells were treated with 5 μM oligomycin (Oligom), 10 μM BPTES and/or 2 mM NAD^+ as indicated for 24h.

In the new version of our manuscript, we avoid using the wording imbalance and use instead “depletion of pyrimidines” and “accumulation of purines”. Furthermore, to test directly whether pyrimidine depletion was the cause of compromised genomic DNA replication, cells were also co-treated with NAD^+ and uridine to replenish pyrimidine pools and restore nucleotide balance independently of the DHODH activity, which was impaired by the treatment with NAD^+ . Consistently, uridine supplementation fully rescued genomic DNA replication in NAD^+ -treated cells (Fig. 4f and below for convenience).

Uridine supplementation restores DNA synthesis in NAD^+ -treated cells. Shown in the manuscript as Fig. 4f. QIBC analysis of γH2AX foci during the cell cycle in cells treated for 24h without (NT) or with 100 μM uridine (Urd) and/or 2 mM NAD^+ .

We hope that the new analyses of metabolites, together with our response to the previous concern #5 of this reviewer will answer this point.

7. In Fig. 3i and 3j, the rescuing effects of oligomycin on NAD^+ induced fork slowing is impressive. Can the authors confirm these effects by directly disrupting ETC?

Thank you very much for this comment.

Indeed, the result was impressive, and we supported it with the new analysis of metabolites (see our answer to the previous comment no. 6).

Inspired by this comment of the reviewer, we disrupted the ETC by using FCCP, and to confirm the role of the ETC in NAD⁺-mediated regulation of pyrimidine metabolism, we inhibited the DHODH complex involved in *de novo* pyrimidine synthesis.

The inhibition of DHODH by brequinar (BRQ) phenocopied the effect of NAD⁺ and, is highly relevant for cancer treatment, in this way, we also found a physiological indirect inhibitor of the activity of DHODH.

Effects of NAD⁺ treatment phenocopy DHODH inhibition. Shown in the manuscript as Fig. 4g. Analysis of γ H2AX foci during cell cycle with QIBC in cells treated for 24h without (NT) or with NAD⁺ as indicated and/or 2 μ M brequinar (BRQ).

We dedicated almost the entire new Fig. 4 to answering this and the following concerns.

8. The effects of FCCP are confusing. Since FCCP should disrupt ATP synthesis, should it also rescue fork speed? Why would FCCP alone reduce DNA synthesis in Fig. 3k? Why are the effects of FCCP different from Oligomycin?

We hope our answer below will clarify the confusion about this point.

Carbonyl cyanide-p-trifluoromethoxyphenylhydrazone (FCCP) ablates the membrane potential, these results are included in Fig. 4c.

Effects of mitochondrial OXPHOS inhibitors on NAD⁺-induced elevation of the mitochondrial membrane potential. Shown in the manuscript as Fig. 4c. Ratios of TMRM and mitotracker green (MTG) dye-intensities normalized to the non-treated (NT) condition. HeLa and U2OS cells were treated as indicated for 24h with 2 mM NAD⁺, 5 μM oligomycin (Oligom), 10 μM BPTES, and/or 10 μM FCCP.

Referee #2 is correct, as a consequence of FCCP treatment, the mitochondrial membrane potential and hence ATP synthesis is inhibited. FCCP reduced genomic DNA synthesis by itself in both cell types but only robustly prevented effects by NAD⁺ treatment in U2OS cells, in contrast to only partial impact in HeLa cells (see below). Consistently with a distinct, AMP-mediated regulation in HeLa cells, treatment with 2 mM of NAD⁺ further decreased the incorporation of EdU.

These results showed that to affect DNA synthesis, NAD⁺ requires mitochondrial membrane proficiency.

NAD⁺ treatment effects depend on mitochondrial membrane potential proficiency. Shown in the manuscript as Fig. 4d. Analysis of γ H2AX foci during cell cycle with QIBC in cells treated for 24h without (NT) or with NAD⁺ as indicated and/or 10 μ M FCCP.

Oligomycin inhibits the ATP synthase (complex V) but maintains the membrane potential. The inhibition of ATP synthesis by oligomycin reduces the electron flow of the ETC. NADH remains high and the NAD⁺ becomes low to feed the TCA. Each of these effects was sufficient to alleviate the effect of NAD⁺ treatment on DNA synthesis.

9. At the end of Fig. 3, the authors speculated that ETC is important for the maintenance of balanced nucleotide pools. Again, this idea is based on correlations of changes and is never tested directly. This is another major weakness of the current study.

For this answer, please see our answers to the previous two concerns (no. 7 & 8), specifying how we have now directly tested and defined this notion.

10. In Fig. 4c, the rescuing effects of BPTES on fork speed are clear. If BPTES affects TCA and indirectly affects ETC, it should have similar effects on mitochondrial membrane potential as Oligomycin and FCCP. The authors should test this directly.

We measured the mitochondrial membrane potential for those conditions. Altogether, our results showed that to accelerate or inhibit pyrimidine anabolism, NAD⁺ treatment requires proficient mitochondrial membrane potential. BPTES reduces the flow of the TCA cycle and the rate to the ETC, by slowing down the TCA, BPTES alleviated the effect of NAD⁺ in DNA synthesis.

Oligomycin, BPTES and FCCP impacted differently the membrane potential or the ETC flux (Fig. 4c).

11. The point on AMP in Fig. 4 is confusing. The authors tried to explain why HeLa cells are sensitive to NAD⁺ than U2OS by showing that HeLa cells accumulate more AMP and AMP inhibits DNA synthesis.

However, even in HeLa cells, BPTES fully rescued DNA synthesis in the presence of exogenous NAD^+ (Fig. 4c), even though fork speed was only partially rescued (Fig. 4a). Does this mean that the effects of endogenous AMP in HeLa cells are not detected by overall EdU incorporation? The effects of exogenous AMP in Fig. 4d could be artificial.

The former of the two scenarios suggested by the reviewer is correct. The DNA fibre technique can detect small variations in fork speed and stability. A reduction of fork speed can be compensated by dormant origin activation to complete the S phase at a constant pace (Blow et al., 2011). A reduction of fork speed below 30-50% will trigger DNA damage response and it cannot be compensated by dormant-origin activation (Koundrioukoff et al., 2013; Maya-Mendoza et al., 2018). A less extreme reduction of fork speed above this threshold will be compensated by dormant origin activation and resulting in no changes in global DNA synthesis. In other words, the mean intensity of EdU incorporation could remain the same. Both approaches complement each other in the analysis of DNA synthesis and provide different information.

Regarding the regulation by AMP, we provide more evidence to support this regulation in our model.

AMP accumulation can inhibit the activity of the UMPS. HeLa cells treated with NAD^+ accumulated IMP and AMP in a concentration-dependent manner. AMP could originate from the degradation of exogenous NAD^+ , which can be hydrolyzed to NMN and AMP, thereby fuelling the endogenous levels of AMP, alternatively, originate from passive accumulation of purine nucleotides upon the NAD^+ -induced depletion of pyrimidines, given DNA synthesis being halted due to pyrimidine shortage.

AMP levels in response to treatment with increasing NAD^+ concentrations. Relative quantification of AMP from metabolomics analysis in HeLa cells treated with the indicated NAD^+ concentrations for 24h, $n=4$, mean \pm SD shown.

Upon NAD⁺ treatment, U2OS accumulated hypoxanthine, which itself is a degradation product of AMP and has been proposed to protect against AMP-induced toxicity due to pyrimidine levels.

Changes in purine metabolites after NAD⁺ treatment in HeLa and U2OS cells. Shown in the manuscript as Fig. 2g. Metabolite levels detected by metabolomics analysis in cells treated without (NT) or with 2 mM NAD⁺ for the indicated hours. The y-axis displays relative metabolite quantification, bars represent means, n=4.

Because treatment with 80 μM of NAD⁺ had already a negative impact on EdU incorporation and fork speed, if there was another mechanism for the regulation of pyrimidine biosynthesis in HeLa cells, and it was dependent on AMP levels, it could be tested experimentally.

Treatment with 2 mM of NAD⁺ had an additive effect on reducing the EdU incorporation together with FCCP in HeLa cells but not in the U2OS cells.

The regulation by AMP is also active in U2OS cells, however, they are not as sensitive as HeLa cells owing to the differential expression of enzymes involved in purine metabolism, the scenario which is now clearly described in the new version of the manuscript. And because the regulation of pyrimidine anabolism by AMP is downstream of the activity of DHODH, it was insensitive to the alleviation by oligomycin or BPTES.

Effects of AMP treatment are independent of mitochondrial metabolic proficiency. Shown in the manuscript as Fig. 4e. QIBC analysis of γH2AX foci during cell cycle in HeLa and U2OS cells treated without (NT) or with 2 mM AMP, 5 μM oligomycin (Oligom), and/or 10 μM BPTES for 24h.

The AMP-dependent regulation in HeLa cells was also evident when we depleted the mitochondrial NAD⁺ transporter (Fig. 5c).

12. In Fig. 5a, it seems that SLC25A51 does not have any impact on the NAD⁺ induced replication inhibition in HeLa cells. This is not consistent with the other results that ETC has a major contribution to this NAD⁺ effect. AMP should be only partially responsible for the effect.

The depletion of SLC25A51 has a mild effect on the DNA synthesis in HeLa and a stronger effect on the U2OS. We do notice that the knockdown of the gene has a deeply negative effect on DNA synthesis, particularly in U2OS. This could explain why only in some cell lines the knockout of this gene was compatible with viability and hence feasible (Luongo et al., 2020). Consistently with an AMP-mediated regulation, 80 μM of AMP slightly enhanced the inhibition of EdU incorporation in HeLa cells but we did not see any effect on U2OS cells. And 2 mM of NAD⁺ had a strong effect on HeLa cells but mildly affected U2OS which were depleted to SCL25A51.

In agreement with the referee's comment, AMP was only partially responsible for the effect of NAD⁺.

Effect of SLC25A51 depletion on NAD⁺ treatment effects. Shown in the manuscript as Fig. 5c. QIBC analysis of γH2AX foci during cell cycle. Cells transfected with control siRNA (CTRL) or si-SLC25A51 were treated for 24h with NAD⁺ as indicated.

13. The interpretations of Fig. 5e and 5f are confusing. The effects of PARPi and NAD⁺ are clearly distinct (Fig. 5f) and additive (Fig. 5e). This argues that PARPi and NAD⁺ are not epistatic. The results of these experiments actually suggest that the effects of PARPi are independent of its impact on NAD(H).

Yes, the referee interpreted correctly our results. Our mistake if we described them in a confusing manner.

The effect of PARP inhibition in DNA synthesis is independent of the increase of NAD(H). To extend this observation we now include new results testing the combination of the PARP1 inhibitor at different concentrations with also different concentrations of NAD⁺. We used HeLa and normal fibroblasts, and three conclusions can be drawn, i) PARPi and NAD⁺ are not epistatic, ii) MRC5 normal fibroblasts are more sensitive to the inhibition of PARP1, iii) the use of PARP1 inhibitors to boost intracellular levels of NAD(H) and thereby alleviate some aspects of aging may have other adverse effects.

Effects of NAD⁺ and PARP inhibitor treatment combinations on cell proliferation. Shown in the manuscript as Fig. 5e. Relative cell number to non-treated cells. HeLa and MRC5 cells were treated with 1 μM PARPi and/or the indicated NAD⁺ concentration for 72h. Mean+SD, n=6, Welch's t-test.

14. The authors idea that NAD⁺ precursors are safer because they affect mitochondria more slowly does not make sense. They should analyze the effects of these precursors on RS and DDR at later time points.

We removed this sentence and performed a new set of experiments to better elucidate this issue.

NR and NMN did not affect the global DNA synthesis even after prolonged incubations (see below), however, a DDR activation was seen after 48h with the accumulation of phosphorylated H2AX, however without compromising cell viability (See below and Extended Data Figure 2l, m).

These results pointed out that the use of precursors of NAD⁺ may induce DNA damage in the long term. We discussed this on Pag. 17, lines 4002-406.

Treatment with NAD⁺ but not its precursors impair DNA replication. Shown in the manuscript as Fig. 1m. QIBC analysis of γ H2AX foci during cell cycle in HeLa cells treated without (NT) or with 2 mM NAD⁺, NR or NMN for the indicated hours.

Differential DDR activation after treatment with NAD⁺ and its precursor NR. Shown in the manuscript as Extended Data Fig. 2m. Western blot of DNA damage response proteins in HeLa cells treated without (NT) or for the indicated hours with 2 mM NAD⁺ or NR.

Reviewer #3 (Remarks to the Author): NAD⁺, aging

Apolinar Maya-Mendoza’s group and colleagues presented a research article entitled ‘NAD⁺ regulates nucleotide levels and genomic DNA replication’. By the application of different cell lines (Human U2OS, HeLa, BJ, BJ-5ta (BJ-tert), T98G, and U87 MG cell lines) and a series of biochemical and omics approaches, the authors manipulated cellular NAD⁺ levels and investigated changes in metabolism and genomic DNA replication. The authors presented a large amount of well-organized data which could be informative to the NAD⁺ and DNA repair fields. However, my enthusiasm was dramatically dampened due to the below major concerns:

- Low novelty on much of the data. While impressive amount of data are presented in this paper, a significant amount of ‘discoveries’ claimed in this paper are not new. E.g., the statement ‘Here we

show that high levels of NAD⁺ rapidly boost Krebs's cycle, increase the ATP level, and accelerate the speed of genomic DNA synthesis' is not new as there are mountains of data in the fields from last century enabled the writings in our text book that NAD⁺ is essential for glycolysis, TCA cycle, and OXPHOS (thus more ATP) (see review summaries from the Guarente, Auwerx and Bohr groups). The importance of NAD⁺ in DNA replication was evidenced from 1960s and has been well-studied onwards (PMID: 4310516).

We thank referee#3 for his/her comments.

It was not our intention to claim as new findings the well-known and established roles of NAD(H) in glycolysis, Krebs's cycle or OXPHOS. Perhaps we did not highlight enough the main new findings of our manuscript and did not put them in a proper context. We performed a substantial number of new experiments to validate and test our concept of how an excess of NAD⁺ can modulate genomic DNA synthesis. This has not been described in the literature yet, and here we also propose the new mechanism behind the observed cellular effects.

The doubts based on the seemingly lacking novelty of our findings can be due to two factors: i) Our poor description of our results and potentially confusing interpretation (aspects that we tried to remedy in this new version of the manuscript); ii) A misunderstanding by the reviewer, possibly due to overlook of some of our results. For instance, there is no published evidence that an increase in ATP results in an accelerated speed of DNA synthesis and it was not our intention to imply that. The increase in ATP was a consequence of the treatment with NAD⁺, and we find that high ATP levels correlate inversely with the speed of DNA synthesis and the inhibition of cell proliferation. Furthermore, the main findings and novelty of our study include the observation that treatment with NAD⁺ in the micromolar range induces depletion of pyrimidine nucleotides, with the concomitant accumulation of purine nucleotides, in a manner directly related to mitochondrial metabolic activities. This central and new concept that we present is now further tested and validated in this revised manuscript, improved by additional sets of experimental results.

Changes in pyrimidine and purine nucleotide levels in response to increasing NAD^+ concentrations. Shown in the manuscript as Fig. 2d. Mean nucleotide levels (bars) from metabolomics analysis and replication fork speed (boxplots) relative to the non-treated (NT) condition. HeLa cells were treated with the indicated NAD^+ concentrations for 24h.

Effects of increasing NAMPT inhibitor or NAD^+ concentrations on cell proliferation and ATP levels. Shown in the manuscript as Fig. 2a. Relative ATP level and cell number to non-treated cells for each cell line. The indicated cell lines were treated with the indicated concentrations of NAMPTi or NAD^+ for 48h (HeLa and U2OS) or 72h (BJ and BJ-T). Smoothed conditional means and CI95 are shown, $n > 4$.

Those findings are some of our new contributions to the topic of how basal metabolism modulates cell proliferation and DNA synthesis.

Referee #3 cited a very interesting publication PMID: 4310516 from the 60s that he/she thought may question the novelty of our manuscript. The article PMID: 4310516 entitled “Replication and properties of DNA in nicotinamide adenine dinucleotide deficiency of Escherichia coli cells” from 1969 is very important but not directly related to our manuscript. In that paper, Nozawa and Mizuno described that the DNA ligase from E. coli needs NAD^+ as a cofactor to ligate replication intermediates. Most eukaryotic DNA ligases, together with archaeal and bacteriophage enzymes, use ATP as a cofactor (Martin and MacNeill, 2002), which is distinct from E. coli bacteria studied in the above article mentioned by the reviewer.

The requirement of NAD^+ to ligate replication intermediates has likely evolved to satisfy the requirement of PARylation in the maturation of Okazaki fragments in eukaryotic cells. Due to its historical value, we cited Nozawa and Muzino’s publication in the new version of our manuscript, however, it does not diminish the value or novelty of our dataset.

We have now addressed all the concerns of referee#3, and we very much hope that she/he would agree with the relevance of our study and the advance it provides to the field.

• Much of the data are descriptive, lacking in-depth mechanistic studies. E.g, the authors have noticed that NAD⁺ regulates nucleotide levels, but the underlying molecular mechanisms are elusive. One possibility could be a feed-forward loop between NAD⁺ and amino acids generation via autophagy (PMIDs: 18296641; PMID: 24813611); a recent paper also showed that autophagy regulates NAD⁺ (PMID: 36413951).

We welcome this comment from referee #3. In response, we now provide an extensive amount of new results to support our model(s) (Extended Data Fig. 10).

The induction of autophagy by treating cells with rapamycin did not show an effect on DNA synthesis and did not modify the effect of NAD⁺ treatment. Confirming unrelated mechanisms, the combination of BPTES and rapamycin did not affect the incorporation of EdU, however, there was a slight increase in the gammaH2AX foci formation, indicating a modestly enhanced DNA damage under such combined treatment.

Effects of NAD⁺ treatment are independent of mTOR activity. QIBC analysis of EdU incorporation (left) and gammaH2AX levels (right) in EdU-positive cells. HeLa cells were treated for 24h without (NT) or with 1 µM rapamycin (Rapa), 10 µM BPTES and/or the indicated NAD⁺ concentrations.

We then assessed by immunoblotting the LC3 activation, but, we did not observe differences in cells that were treated with NAD⁺. These experiments suggested that autophagy is not involved in the phenotypes

triggered by NAD⁺ treatment. We included these results in this rebuttal but not in the current version of our manuscript due to space limitations.

LC3 detection following NAD⁺ treatment. Cells were treated with 2 mM of NAD⁺ for the indicated times. LC3 was detected using the method described in our previous paper by Vanzo et al. (2020).

Another interesting idea suggested by Referee #3 is that autophagy could serve as a feedback loop to provide amino acids. Our results suggested that amino acids are not replenished after treating the cells with NAD⁺, on the contrary, they may be depleted (Fig. 3a). However, and conclusively, our unbiased metabolomics analyses did not suggest amino acid metabolism to be the targeted mechanism by NAD⁺ treatment, in contrast to the observed pyrimidine depletion and purine accumulation.

Autophagy, specifically mitophagy, is required to maintain healthy mitochondria and if this organelle is damaged, it should be cleared. An early step in the defective mitochondria is the drop in NAD(H) and some of the precursors of NAD(H) or the inhibition of PARP can restore the mitochondrial levels of NAD(H) (Fang et al., 2014; Kataura et al., 2022). The cellular changes triggered by the NAD⁺, at least at the time we tested, seem not related to the autophagic processes. We find that NAD⁺ directly boosts mitochondrial activity, but not its precursors, and we provide experimental evidence and discuss this phenomenon.

• **When working on metabolism (especially mitochondrial metabolism) and DNA replication, it is necessary to distinguish cancer-like-immortalized cell system and the normal cellular system. The majority of the data of this paper were from cancer (cancer-like) lines (although a small portion of data were repeated in primary cells), one should be extremely cautious in data interpretation. The used cell lines are majorly cancer cells: U2OS is from a moderately differentiated sarcoma, HeLa is from cervical cancer cells, while U87 MG is original from malignant gliomas. BJ (CRL-2522) is from skin taken from normal foreskin from a neonatal male. This cell line, although from a normal individual, shows signs of ‘immortalization’ (and likely cancer-like feature), including telomerase negative (although could lead to accelerated ageing) and ‘the capacity to proliferate to a maximum of 72 population doublings before the onset of senescence’ (<https://www.atcc.org/products/crl-2522>); for normal human fibroblasts, the average passages numbers are around 30-40 when entering to senescence (see papers from Campisi lab etc). Although BJ-tert cell line is used, it is important to use multiple primary cell lines to validate the data from cancer cell lines, if the authors want to generalize their findings to the normal cells.**

We fully agree with the referee on this point. It seems the effect of NAD⁺ treatment on the replication fork speed and stability is a general feature of mammalian cells. We have now included also more

primary cell types, MRC5 and MEFs, and also more examples of cervical cancer cell lines (Fig. 1f and results included only in this rebuttal).

Replication sensitivity to NAD⁺ treatment. Shown in the manuscript as Fig. 1f. Replication fork speed in the indicated cell lines treated without or with 2 mM NAD⁺ for 24h. Mean±SD, n>500 fibers per condition per cell line.

DNA synthesis sensitivity to NAD⁺ treatment and NAMPT inhibition in different cervical cancer cell lines. CaSki, SiHa and HeLa cells were treated as indicated for 24h without (Ctrl) or with 10 nM NAMPT inhibitor or 2 mM NAD⁺ followed by EdU pulse-labelling for 30 min. Flow cytometry analysis was then used for quantifying EdU incorporation (y-axis) and Hoechst intensity (DNA content, x-axis).

Collectively, the authors are applauded for a large amount of informative data generated. This paper may fit better for a cancer journal. In view of the importance of NAD⁺ in proliferation and chemotherapy-induced resilience, data from this paper may be more interesting to cancer researchers.

We believe that NCB is the right choice to publish our findings, considering the multiple improvements that we have made in the revised manuscript. NCB has published multiple articles with a similar scope to ours, as well as many studies of cancer cells.

Reviewer #4 (Remarks to the Author): NAD⁺ homeostasis

Summary:

Munk et. al., aimed to address if the levels of NAD(H) had an impact on DNA damage response and/or genomic DNA synthesis. These processes are critical for maintaining homeostasis and genomic integrity. Most importantly, NAD⁺ is a key redox cofactor that serves as a key substrate for PARP enzymes, which regulate genomic DNA synthesis. The authors were able to show that increasing NAD⁺ concentrations led to increased mitochondrial activity and DNA synthesis. However, prolonged NAD⁺ elevation triggered a reduction in pyrimidine biosynthesis and cell cycle arrest. This observation is critical as we continue to understand how to utilize NAD⁺ supplements for various aged-related diseases, and how increase NAD⁺ can have negative feedback on organismal homeostasis. The authors proposed a model that showcases and reconcile how NAD⁺ controls mitochondrial function, DNA replication, and cell proliferation—with the hope of explaining how NAD⁺ metabolism modulates basal metabolism and DNA synthesis.

We are grateful to Referee #4 for his/her support and enthusiasm for our work.

Comments/Questions:

- Key literature is not cited and discussed in the current state of the manuscript. This is misleading to the audiences who will read and use this work as a resource. Please be sure to include relevant citations that span across the field.

We have now checked to make sure that the references are correctly used and now have added some additional relevant references. Given a concise letter format, we had to reduce the number of references, we apologize for those that were not included in the previous version of our manuscript.

- Figures throughout the paper can be labeled better and more properly to help the reader follow along with the authors. Recommend adding titles to each figure for feasibility.

We have added titles to each figure and re-formatted the figure configuration accordingly to address all concerns pointed out by the 4 referees.

- For the global metabolomics—the authors should preform FDR correction on their results to ensure no false discoveries are identified

The FDR correction has now been performed, as suggested by the reviewer, in all metabolomics analyses and significance was indicated by asterisks in the heatmaps to aid the reader (see M&M).

- One limitation is that this paper only represents cell lines. It's understood that NAD⁺ metabolism drastically changes between in vitro and in vivo conditions (PMID: 29685734). It would be of great interest to test this hypothesis/model in vivo to confirm or discuss the limitations in the discussion if unable to perform this key experiment.

We agree with this comment from referee #4. We only characterized the effect of a surplus of NAD⁺ using cellular models and in vivo experiments could be highly informative. Intravenous injections of NAD⁺ are available for clinical use, nevertheless, this anti-aging treatment has not received FDA approval. Direct infusions of NAD⁺ also are offered as an alternative to treat many addictions including those to alcohol and heroin.

Norris Mestayer, Paula. **Addiction: The Dark Night of the Soul/ NAD⁺: The Light of Hope**. Balboa Press. 2019.

Because an injectable solution of NAD⁺ is already available in the market, it is feasible to think that the real test is to characterize its effect directly in humans.

We found a promising combination between NAD⁺ and 5-FU to be used in cancer therapy. We are currently about to initiate preclinical studies on XPD-cancer models and plan clinical trials to treat highly proliferative cancers, such as glioblastoma.

We have chosen to publish our data in a letter format in NCB now to inspire more clinical trials and to test our concepts in several different animal models of cancer and aging. These experiments are time-consuming and costly, therefore, we will prefer to communicate our research as soon as possible to have the most impact as possible in the related research field.

- The authors are relying on NAD/NADH assays kit measurements to quantify NAD⁺. This is extremely

limiting for their observations, their findings, and many additional reasons. Combining liquid chromatography and mass spectrometry is key for accurately and precisely quantitating NAD(H) levels. The conclusions are too strong for using a kit and not LCMS.

We have used both approaches to measure NAD(H). NAD(H) levels were quantified as mentioned by the reviewer in all metabolomics data sets. This is an example included in Extended Data Figure 2a, where the measured relative areas have been normalized to that of the non-treated condition and presented as log₂ fold changes.

Metabolomics quantification of NAD⁺ and NADH levels after NAD⁺ treatment. Shown in the manuscript as Extended Data Fig. 2a-b. Metabolomics analysis of NAD⁺ and NADH levels relative to non-treated (NT) cells. HeLa and U2OS cells were treated without or with 2 mM NAD⁺ for the indicated time. n=4, Welch's ANOVA test.

In summary, the authors have revealed potential mechanisms that link NAD to modulating basal metabolism and DNA synthesis. However, prolonged NAD⁺ elevation does trigger negative downstream events. Interestingly, the authors propose how NAD-supplements could indeed be a safer option for increasing intracellular NAD⁺ without the negative impacts (triggering DDR, etc.).

We thank this comment from referee#4 and we hope our answers will convince other referees about the relevance and quality of our data sets.

References mentioned in this rebuttal document

- Alano, C.C., Ying, W., and Swanson, R.A. (2004). Poly(ADP-ribose) polymerase-1-mediated cell death in astrocytes requires NAD⁺ depletion and mitochondrial permeability transition. *J Biol Chem* 279, 18895-18902.
- Bartkova, J., Horejsi, Z., Koed, K., Kramer, A., Tort, F., Zieger, K., Guldborg, P., Sehested, M., Nesland, J.M., Lukas, C., *et al.* (2005). DNA damage response as a candidate anti-cancer barrier in early human tumorigenesis. *Nature* 434, 864-870.
- Bartkova, J., Rezaei, N., Liontos, M., Karakaidos, P., Kletsas, D., Issaeva, N., Vassiliou, L.V., Kolettas, E., Niforou, K., Zoumpourlis, V.C., *et al.* (2006). Oncogene-induced senescence is part of the tumorigenesis barrier imposed by DNA damage checkpoints. *Nature* 444, 633-637.
- Bester, A.C., Roniger, M., Oren, Y.S., Im, M.M., Sarni, D., Chaoat, M., Bensimon, A., Zamir, G., Shewach, D.S., and Kerem, B. (2011). Nucleotide deficiency promotes genomic instability in early stages of cancer development. *Cell* 145, 435-446.
- Billington, R.A., Travelli, C., Ercolano, E., Galli, U., Roman, C.B., Grolla, A.A., Canonico, P.L., Condorelli, F., and Genazzani, A.A. (2008). Characterization of NAD uptake in mammalian cells. *J Biol Chem* 283, 6367-6374.
- Blow, J.J., Ge, X.Q., and Jackson, D.A. (2011). How dormant origins promote complete genome replication. *Trends Biochem Sci* 36, 405-414.
- Braidy, N., Villalva, M.D., and van Eeden, S. (2020). Sobriety and Satiety: Is NAD⁺ the Answer? *Antioxidants (Basel)* 9.
- Bruzzone, S., Guida, L., Zocchi, E., Franco, L., and De Flora, A. (2001). Connexin 43 hemi channels mediate Ca²⁺-regulated transmembrane NAD⁺ fluxes in intact cells. *FASEB J* 15, 10-12.
- Bruzzone, S., Moreschi, I., Guida, L., Usai, C., Zocchi, E., and De Flora, A. (2006). Extracellular NAD⁺ regulates intracellular calcium levels and induces activation of human granulocytes. *Biochem J* 393, 697-704.
- Buonvicino, D., Ranieri, G., Pittelli, M., Lapucci, A., Bragliola, S., and Chiarugi, A. (2021). SIRT1-dependent restoration of NAD⁺ homeostasis after increased extracellular NAD⁺ exposure. *J Biol Chem* 297, 100855.
- Cambronne, X.A., Stewart, M.L., Kim, D., Jones-Brunette, A.M., Morgan, R.K., Farrens, D.L., Cohen, M.S., and Goodman, R.H. (2016). Biosensor reveals multiple sources for mitochondrial NAD(+). *Science* 352, 1474-1477.
- Canto, C., Houtkooper, R.H., Pirinen, E., Youn, D.Y., Oosterveer, M.H., Cen, Y., Fernandez-Marcos, P.J., Yamamoto, H., Andreux, P.A., Cettour-Rose, P., *et al.* (2012). The NAD(+) precursor nicotinamide riboside enhances oxidative metabolism and protects against high-fat diet-induced obesity. *Cell Metab* 15, 838-847.
- Canto, C., Menzies, K.J., and Auwerx, J. (2015). NAD(+) Metabolism and the Control of Energy Homeostasis: A Balancing Act between Mitochondria and the Nucleus. *Cell Metab* 22, 31-53.
- Davila, A., Liu, L., Chellappa, K., Redpath, P., Nakamaru-Ogiso, E., Paoletta, L.M., Zhang, Z., Migaud, M.E., Rabinowitz, J.D., and Baur, J.A. (2018). Nicotinamide adenine dinucleotide is transported into mammalian mitochondria. *Elife* 7.
- Diehl, F.F., Miettinen, T.P., Elbashir, R., Nabel, C.S., Darnell, A.M., Do, B.T., Manalis, S.R., Lewis, C.A., and Vander Heiden, M.G. (2022). Nucleotide imbalance decouples cell growth from cell proliferation. *Nat Cell Biol* 24, 1252-1264.

- Fang, E.F., Scheibye-Knudsen, M., Brace, L.E., Kassahun, H., SenGupta, T., Nilsen, H., Mitchell, J.R., Croteau, D.L., and Bohr, V.A. (2014). Defective mitophagy in XPA via PARP-1 hyperactivation and NAD(+)/SIRT1 reduction. *Cell* *157*, 882-896.
- Kataura, T., Sedlackova, L., Otten, E.G., Kumari, R., Shapira, D., Scialo, F., Stefanatos, R., Ishikawa, K.I., Kelly, G., Seranova, E., *et al.* (2022). Autophagy promotes cell survival by maintaining NAD levels. *Dev Cell* *57*, 2584-2598 e2511.
- Kory, N., Uit de Bos, J., van der Rijt, S., Jankovic, N., Gura, M., Arp, N., Pena, I.A., Prakash, G., Chan, S.H., Kunchok, T., *et al.* (2020). MCART1/SLC25A51 is required for mitochondrial NAD transport. *Sci Adv* *6*.
- Koundrioukoff, S., Carignon, S., Techer, H., Letessier, A., Brison, O., and Debatisse, M. (2013). Stepwise activation of the ATR signaling pathway upon increasing replication stress impacts fragile site integrity. *PLoS Genet* *9*, e1003643.
- Li, J., K, M.S., Ibrahim, M., Zeng, X., McClellan, S., Angajala, A., Beiser, A., Andrews, J.F., Sun, M., Koczor, C.A., *et al.* (2021). NAD(+) bioavailability mediates PARG inhibition-induced replication arrest, intra S-phase checkpoint and apoptosis in glioma stem cells. *NAR Cancer* *3*, zcab044.
- Luongo, T.S., Eller, J.M., Lu, M.J., Niere, M., Raith, F., Perry, C., Bornstein, M.R., Oliphint, P., Wang, L., McReynolds, M.R., *et al.* (2020). SLC25A51 is a mammalian mitochondrial NAD(+) transporter. *Nature* *588*, 174-179.
- Martin, I.V., and MacNeill, S.A. (2002). ATP-dependent DNA ligases. *Genome Biol* *3*, REVIEWS3005.
- Maya-Mendoza, A., Moudry, P., Merchut-Maya, J.M., Lee, M., Strauss, R., and Bartek, J. (2018). High speed of fork progression induces DNA replication stress and genomic instability. *Nature* *559*, 279-284.
- Maynard, S., Hall, A., Galanos, P., Rizza, S., Yamamoto, T., Gram, H.H., Munk, S.H.N., Shoaib, M., Sorensen, C.S., Bohr, V.A., *et al.* (2022). Lamin A/C impairments cause mitochondrial dysfunction by attenuating PGC1alpha and the NAMPT-NAD+ pathway. *Nucleic Acids Res* *50*, 9948-9965.
- Nikiforov, A., Dolle, C., Niere, M., and Ziegler, M. (2011). Pathways and subcellular compartmentation of NAD biosynthesis in human cells: from entry of extracellular precursors to mitochondrial NAD generation. *J Biol Chem* *286*, 21767-21778.
- O'Dwyer, P.J., King, S.A., Hoth, D.F., and Leyland-Jones, B. (1987). Role of thymidine in biochemical modulation: a review. *Cancer Res* *47*, 3911-3919.
- Pirinen, E., Canto, C., Jo, Y.S., Morato, L., Zhang, H., Menzies, K.J., Williams, E.G., Mouchiroud, L., Moullan, N., Hagberg, C., *et al.* (2014). Pharmacological inhibition of poly(ADP-ribose) polymerases improves fitness and mitochondrial function in skeletal muscle. *Cell Metab* *19*, 1034-1041.
- Szczepanowska, K., Senft, K., Heidler, J., Herholz, M., Kukat, A., Hohne, M.N., Hofsetz, E., Becker, C., Kaspar, S., Giese, H., *et al.* (2020). A salvage pathway maintains highly functional respiratory complex I. *Nat Commun* *11*, 1643.
- Weisman, G.A., Lustig, K.D., Lane, E., Huang, N.N., Belzer, I., and Friedberg, I. (1988). Growth inhibition of transformed mouse fibroblasts by adenine nucleotides occurs via generation of extracellular adenosine. *J Biol Chem* *263*, 12367-12372.
- Ying, W., Garnier, P., and Swanson, R.A. (2003). NAD+ repletion prevents PARP-1-induced glycolytic blockade and cell death in cultured mouse astrocytes. *Biochem Biophys Res Commun* *308*, 809-813.

Email Correspondence, second revision

Message: Dear Jiri,

We now have received all reviews from the original revs on your revision "NAD+ regulates nucleotide metabolism and genomic DNA replication" -- we have been discussing the reviewers' comments editorially and apologize for the delay in sending our decision to you.

I am writing because Reviewer #1 shared some comments regarding the revision that we found important. I am pasting Rev#1's comments below. At this stage, we were hoping to ask if you would be willing to please send us responses to the comments pasted below, ****focusing on points #2-3 that are of greater editorial concern****.

*****To be clear, we are not asking for a revised manuscript/new experiments/new data now. We are interested in hearing your thoughts on points #2-3 within 2 business days if possible - including if you tried these analyses/experiments and they could not be done, or if you may have these analyses/data, or if you think you would be able to provide analyses or minimal experiments along these lines in a reasonable time frame if needed or if you disagree with the need for these analyses, etc.*****

****This would be extremely informative to us as we continue the editorial process. Please note that we may discuss your response with the reviewers again before reaching our decision editorially.****

Please let me know if you have any questions and I look forward to hearing your thoughts on the points below. Thank you so much for your time and consideration, and please do let me know if you have any questions or concerns.

Best wishes,
Melina

--

Melina Casadio, PhD
Senior Editor, NCB

REVIEWER #1 COMMENTS TO THE AUTHORS

The new experimental data have strengthened the manuscript. However, while some of my initial concerns have been addressed, my enthusiasm is limited because of the unresolved concerns that are noted below.

- 1) The lack of in vivo evidence for the observed phenotypes. Does prolonged exposure to NAD results in pyrimidine depletion and, subsequently, replication stress in vivo? This is key to establishing the broader physiological relevance of the presented findings.
- 2) Failure to address the mechanism by which exogenous NAD depletes pyrimidine levels while increasing purine levels. Tracing experiments for pyrimidine and purine synthesis would clarify the author's proposed model.
- 3) The cell permeability of NAD remains undetermined. While the additional references provided by the authors are helpful and informative, I remain unpersuaded that NAD is effectively entering the cells. ¹³C-NAD tracing experiments could be performed to directly

monitor the fractional abundance of ^{13}C -NAD with respect to total NAD levels in the cytosol versus mitochondria.

Author Response to Review Comments, second revision

Subject: NCB-LE50081B

A point-by-point response to Reviewer 1's concerns related to the revised manuscript

Below, we first copy the text of each of the specific comments (1-3), followed by our response in bold. For the convenience of the Editors, our responses contain also some very relevant data from the references that we either used or can add if deemed helpful for the message of our paper, as well as complemented by several figure panels with legends from our revised manuscript, to support our mechanistic conclusions and model (Point 2) and the issue of NAD cell entry, the latter also supported by our unpublished data shown here (Point 3) respectively.

Point 1

- 1) The lack of in vivo evidence for the observed phenotypes. Does prolonged exposure to NAD results in pyrimidine depletion and, subsequently, replication stress in vivo? This is key to establishing the broader physiological relevance of the presented findings.

While we agree with the Reviewer that in vivo experiments are generally important to assess the physiological relevance of any newly discovered biological mechanism, none of the other reviewers raised this concern about our manuscript, likely because the majority of broadly analogous metabolism-focused mechanistic studies are first published in the context of cellular models, and only later followed by the time- and resources-consuming in vivo experiments to corroborate the initial discovery that provides a significant advance to the field in its own right. As we already explained in our response to the Reviewer's first round of comments, we chose to extend this initial dataset directly by organizing a human clinical trial, given the rather special situation of NAD⁺ being approved and available for human use without a requirement of a medical prescription. As an example of the trial we are considering, patients suffering from glioblastoma multiforme tumors might benefit from being treated with a combination of 5-FU and NAD⁺. The logistics behind such trials is always complicated, conclusive results are commonly reached only after several years and thus beyond the realistic scope of this first manuscript on the new mechanism that we report here.

- 2) Failure to address the mechanism by which exogenous NAD depletes pyrimidine levels while increasing purine levels. Tracing experiments for pyrimidine and purine synthesis would clarify the author's proposed model.

We have carefully considered Rev#1's concern regarding the mechanism underlying the depletion of pyrimidine and increase in purine levels upon exogenous NAD⁺ treatment. Below we address this query and elaborate on the pertinent sections within our manuscript that address these points. In general, we do acknowledge the possibility of utilizing isotope-labeled NAD⁺ for tracing experiments as a means to investigate the dynamics of pyrimidine and purine synthesis. However, we opted for a combination of several alternative experimental strategies, which though more intricate than conventional tracing experiments, present the advantage of circumventing the use of radioactivity and offer deeper functional insights into the cellular response to exogenous NAD⁺.

In this context, we would like to draw your attention to the experimental evidence presented in our revised manuscript. We detail the mechanistic effect of exogenous NAD⁺ on pyrimidine depletion through the impairment of mitochondrial DHODH enzyme activity which is essential for *de novo* pyrimidine synthesis. This effect is thoroughly demonstrated through multiple experiments, including mitochondrial metabolic activity assays (Fig. 4b), membrane potential analysis (Fig. 4c), and analyses utilizing the chemical DHODH inhibitor brequinar (Fig. 4g). Furthermore, our findings are reinforced by the ability to rescue the pyrimidine depletion effect through the restriction of mitochondrial metabolic activity (Fig. 3b-g) or uridine supplementation that bypasses DHODH dependency for pyrimidine synthesis (Fig. 4f). The mechanistic role of the mitochondrial transporter SLC25A51 in moderating the effect of exogenous NAD⁺ is also highlighted, supported by genetic and functional evidence (see also our response to Point 3 below).

Detailed insights into the metabolism of purines and the expression levels of associated enzymes from metabolomics and RNA-seq analysis, respectively, are provided in the Extended Data Fig. 5. Due to its placement as ED data, these important results might have escaped the attention of the reviewer. Indeed, one possibility to remedy this would be to place this display item as a separate main figure, provided such change would be requested/approved at the editorial level.

While we acknowledge the potential value of tracing experiments, we firmly believe that our current experimental framework, supported by the extensive data presented, sufficiently validates our proposed model. The complexity and extent of the results and the depth of the mechanistic insights we have obtained would likely not be fully captured by traditional tracing experiments using labeled NAD⁺ or other metabolites. As such, we hold

the view that further tracing experiments would not significantly enhance the robustness of our conclusions.

More specifically, our model is supported by rigorously controlled evidence that exogenous NAD⁺ induces depletion of pyrimidines (Fig 2c-e) by impairing the activity of the mitochondrial DHODH. We demonstrate that exogenous NAD⁺ increases mitochondrial NAD(H) levels (Fig. 2b and Extended Data Figure 3c) and the mitochondrial metabolic activity (Fig. 4b), leading to hyperpolarization of the mitochondrial membrane potential (Fig. 4c), which impairs DHODH activity to the same extent as the DHODH chemical inhibitor brequinar (Fig. 4g). Supporting this model, the effect of exogenous NAD⁺ can be rescued by restricting mitochondrial metabolic activity using the inhibitors oligomycin and BPTES (Fig. 3b-g) or by supplementation with uridine (Fig. 4f). Moreover, depletion of the mitochondrial transporter of NAD⁺, SLC25A51, blunts the effect of exogenous NAD⁺ (Fig. 5c).

Regarding the variation in purine nucleotide accumulation after exogenous NAD⁺ addition, we emphasize the cell type-specific response of HeLa and U2OS cells, where AMP and hypoxanthine accumulate, respectively. This nuanced response is intricately tied to cell-specific physiological regulation mediated by AMP levels (Fig. 2d-h).

Changes in nucleotide levels and replication fork speed in response to NAD^+ treatment. (Shown in the manuscript as Fig. 2d,e). **d**, Mean nucleotide levels (bars) from metabolomics analysis and replication fork speed (boxplots) relative to the non-treated (NT) condition. HeLa cells were treated with the indicated NAD^+ concentrations for 24h. **e**, Mean nucleotide levels (bars) from metabolomics analysis and replication fork speed (boxplots) relative to non-treated HeLa cells (HeLa NT). HeLa and U2OS cells were treated with 2 mM NAD^+ for the indicated hours.

Indeed, after the addition of exogenous NAD^+ , HeLa cells accumulate AMP in a concentration- and time-dependent manner, while U2OS cells accumulate the purine intermediate hypoxanthine. The degradation of adenine nucleotides to hypoxanthine protects cells from the negative effect of adenosine on cell proliferation (Weisman et al., 1988).

A detailed analysis of the metabolism of purines and the expression level of related enzymes is included as part of the Extended Data Fig 5.

Purine metabolite changes in response to NAD^+ treatment. (Shown in the manuscript as Extended Data Fig. 5b). b. Purine synthesis and metabolic pathways with assigned metabolite levels (left) and enzyme expression levels (right) as detected by metabolomics and RNA-seq analyses. HeLa and U2OS cells were treated without (NT) or with 2 mM NAD^+ for the

indicated hours. Dashed lines indicate multiple synthesis steps. In metabolite plots, the y-axis displays relative quantification and bars represent means of n=4. In boxplots of enzyme expression, the y-axis shows normalized counts, n=3. All boxplots show the median value and the lower and upper hinges correspond to the 25th and 75th percentiles.

We very much hope that this concise summary of the mechanistically-focused data helps to make the point that we do have sufficient evidence to support the model presented in our revised manuscript, and without additional tracing experiments that we think would not add any significant further mechanistic insights.

- 3) The cell permeability of NAD remains undetermined. While the additional references provided by the authors are helpful and informative, I remain unpersuaded that NAD is effectively entering the cells. ¹³C-NAD tracing experiments could be performed to directly monitor the fractional abundance of ¹³C-NAD with respect to total NAD levels in the cytosol versus mitochondria.

Overall to this point, we are entirely convinced (as were apparently also all the other reviewers who evaluated our manuscript) that together with thorough published evidence to this point, our own data document beyond any reasonable doubt, that mammalian cells are permeable for NAD, which also efficiently enters the mitochondria. Given the fact that this knowledge is well established, we did not even include our control experiment for fluorescence tracing the NAD's cell permeability – shown at the end of this response for information.

First, prompted by the reviewer, we can now provide further references demonstrating the transport of exogenous NAD⁺ across the plasma membrane. In Billington et al. (Billington et al., 2008) ([https://www.jbc.org/article/S0021-9258\(20\)57215-3/fulltext](https://www.jbc.org/article/S0021-9258(20)57215-3/fulltext)), uptake of exogenous NAD in human and murine cells was demonstrated using [³²P]NAD (see below, the figure is taken from Billington et al.).-i.e. through an isotope-based tracing approach that the reviewer recommends.

FIGURE 1. NAD transport in NIH-3T3 cells. *A*, time-dependent uptake of $[^{32}\text{P}]\text{NAD}$ (circles, solid line) and $[^{32}\text{P}]\text{NAADP}$ (squares, dashed line) in NIH-3T3 cells. Data presented as mean \pm S.E., $n = 8-12$. *B*, HPLC analysis of transported nucleotides in NIH-3T3 cells. The gray line represents the HPLC trace for authentic standards (NAD, peak 3 min; NAADP, peak 13 min). The other lines represent radioactivity remaining extracellularly (dashed line, squares) and recovered intracellularly (solid line, circles) after $[^{32}\text{P}]\text{NAD}$ incubation. *C*, kinetic plot of $[^{32}\text{P}]\text{NAD}$ uptake in NIH-3T3 cells. Data presented as mean \pm S.E., $n = 6-8$.

The experimental evidence presented by Pittelli et al. (Pittelli et al., 2010) (<https://pubmed.ncbi.nlm.nih.gov/21917911/>), showing that exogenous NAD is transported intact across the plasma membrane is convincing for us and supports our findings (see below, the figure taken is from Pittelli et al.)

Fig. 1. Effects of eNAD and metabolic precursors on iNAD contents in various cell lines. **A**, iNAD increase upon exposure of HeLa cells to eNAD is time-, concentration-, and temperature-dependent. **B**, increase of iNAD upon exposure of different cell lines (RAW mouse macrophages or C6 rat glioma cells) or primary astrocyte or neuronal cultures to 1 mM NAD for 6 h. Neuronal cultures of 7 or 15 DIV were used. **C**, effect of different NAD precursors added to the culture medium (1 mM for 6 h) on iNAD content in HeLa cells. **D**, effect of 1 mM eNAD on iNAD depletion induced by FK866 (100 μ M for 6 h). In **A**, each point represents the mean \pm S.E.M. of four experiments conducted in duplicate. In **B** to **D**, each column represents the mean of three experiments conducted in duplicate. *, $p < 0.05$; **, $p < 0.01$; ***, $p < 0.001$, versus control (i.e., without eNAD addition); analysis of variance + Tukey post hoc test. ADPR, ADP-ribose.

In Roh et al. (Roh et al., 2018) ([https://www.metabolismjournal.com/article/S0026-0495\(18\)30173-2/fulltext](https://www.metabolismjournal.com/article/S0026-0495(18)30173-2/fulltext)), exogenous NAD⁺ uptake into hypothalamic neuronal cells was shown to be dependent on and occur via connexin 43.

Furthermore, in our control experiments performed in the early phases of this project, we used fluorescence-based tracing to validate NAD cellular permeability: we used 2-Ethynyl-Adenosine-NAD⁺ (click-NAD⁺) which is subsequently detected by click chemistry (Zhang et al., 2019) to corroborate published observations on this matter. We detected a strong intracellular NAD signal after 24h of incubation with Click-NAD⁺ and a mild signal as early as after 1h (see the figure below). We did not include these results in our manuscript because we considered that the issue of NAD cell permeability was firmly established by others before us and accepted in the field.

In general, Denmark (and other countries) have substantially phased down the use of radioactive isotopes such as the experiments recommended by the referee. Instead, fluorescence tracing is preferred, also due to the serious environmental and health concerns about radioactivity. We could add this unpublished control result in the ED section of the results if deemed useful/necessary.

Detection of 2-Ethynyl-Adenosine-NAD⁺ in cells. HeLa cells were incubated in a full cell culture medium containing 2-Ethynyl-Adenosine-NAD⁺ (Jena Bioscience Cat. CLK-043) at 10 µM for the indicated times. Cells were fixed and the 2-Ethynyl-Adenosine-NAD⁺ was detected by click chemistry using azide-AlexaFluor 555.

In his/her critical point 3, the Reviewer also asks about the NAD entry into mitochondria, an aspect that we believe was also convincingly documented already in our revised

manuscript. Specifically, we addressed the mitochondrial entry of exogenous NAD⁺. We show that treatment with exogenous NAD⁺ resulted in a rapid increase of intra-mitochondrial NAD(H) levels, and to a markedly greater extent than the NAD⁺ precursor nicotinamide riboside (NR) at 1h of the treatment.

C

Quantification of total mitochondrial NAD(H). (Shown in manuscript as Extended Data Fig. 3c). HeLa cells were treated without (NT) or with NAD⁺ or NR as indicated. Mean+SD, n=3, Student's t-test.

Upon depletion of the mitochondrial NAD⁺ transporter SLC25A51, treatment with exogenous NAD⁺ did not elevate mitochondrial NAD(H) levels, demonstrating the dependency on this transporter for increased mitochondrial NAD(H) in response to exogenous NAD⁺ treatment.

Effect of SLC25A51 depletion on mitochondrial NAD(H) levels. (Shown in the manuscript as Extended Data Fig. 9d). Quantification of total mitochondrial NAD(H) in cells transfected with control siRNA (siCTRL) or si-SLC25A51 treated for 24h without (NT) or with 2 mM NAD⁺. n=3 (HeLa) and n=2 (U2OS), mean+SD, Student's t-test (HeLa).

Furthermore, we show that depletion of SLC25A51 blunts the effect of exogenous NAD⁺ on genomic DNA replication.

Effect of SLC25A51 depletion on NAD⁺ treatment effects. (Shown in the manuscript as Fig. 5c). Quantitative image-based cytometry analysis of EdU incorporation and γH2AX foci during cell cycle. Cells transfected with control siRNA (CTRL) or si-SLC25A51 were treated

for 24h with NAD⁺ as indicated.

Together, we interpreted our results as evidence that NAD⁺, or at least a clearly detectable proportion of the NAD molecules, can reach the mitochondria and modify its function. We very much hope that together with the published studies on this phenomenon, these results presented in our manuscript sufficiently document both, the notion of the efficient NAD⁺ cellular uptake from the extracellular environment, as well as NAD⁺ entry into mitochondria from the cytoplasm (the latter also documented by our genetic and functional data based on manipulation of the mitochondrial NAD transporter SLC25A51).

References

- Billington, R.A., Travelli, C., Ercolano, E., Galli, U., Roman, C.B., Grolla, A.A., Canonico, P.L., Condorelli, F., and Genazzani, A.A. (2008). Characterization of NAD uptake in mammalian cells. *J Biol Chem* 283, 6367-6374.
- Pittelli, M., Formentini, L., Faraco, G., Lapucci, A., Rapizzi, E., Cialdai, F., Romano, G., Moneti, G., Moroni, F., and Chiarugi, A. (2010). Inhibition of nicotinamide phosphoribosyltransferase: cellular bioenergetics reveals a mitochondrial insensitive NAD pool. *J Biol Chem* 285, 34106-34114.
- Roh, E., Park, J.W., Kang, G.M., Lee, C.H., Dugu, H., Gil, S.Y., Song, D.K., Kim, H.J., Son, G.H., Yu, R., *et al.* (2018). Exogenous nicotinamide adenine dinucleotide regulates energy metabolism via hypothalamic connexin 43. *Metabolism* 88, 51-60.
- Weisman, G.A., Lustig, K.D., Lane, E., Huang, N.N., Belzer, I., and Friedberg, I. (1988). Growth inhibition of transformed mouse fibroblasts by adenine nucleotides occurs via generation of extracellular adenosine. *J Biol Chem* 263, 12367-12372.
- Zhang, X.N., Cheng, Q., Chen, J., Lam, A.T., Lu, Y., Dai, Z., Pei, H., Evdokimov, N.M., Louie, S.G., and Zhang, Y. (2019). A ribose-functionalized NAD(+) with unexpected high activity and selectivity for protein poly-ADP-ribosylation. *Nat Commun* 10, 4196.

Decision Letter, second revision:

Message: Our ref: NCB-LE50081B

11th September 2023

Dear Jiri,

Thank you very much for submitting your revised manuscript "NAD⁺ regulates nucleotide metabolism and genomic DNA replication" (NCB-LE50081B) and thank you for your patience with the re-review process and with the additional discussions. It has now been

seen by the original referees, as you know, and their comments are below.

As we discussed already, Rev#1 had three major remaining concerns: (1) the lack of in vivo data; (2) the lack of deeper investigations of the mechanism underlying the changes in pyrimidines and purines; and (3) the lack of direct evidence to support NAD⁺ entry into cells as is. We thank you again for providing responses to these points to help inform our decision-making process.

We have now discussed these remaining concerns editorially and with Rev#3 (NAD expert).

As you know, we editorially would not require in vivo data for this manuscript's publication, and we were most interested in Rev#3's thoughts on points #2-3 from Rev#1.

You will see that Rev#3 is in full agreement with Rev#1. We found their viewpoint very informative and discussed their comments (see them below with the reviews) in depth at the journal and with Rev#3.

First, we find their points regarding the potential for NAD⁺ to enter cells compelling. This is a controversial question in the field and we are not sure that tracing studies alone would fully be sufficient to address it (i.e., without accompanying mechanistic studies to explain the data - and these are beyond the scope of the paper). Therefore, we ask that you please tone down and rewrite the manuscript text at any instance related to NAD entry into cells and any description of the interpretation comparing effects of NAD⁺ to those of its precursors (because this comparison is not made systematically throughout the paper in each experiment, hence, we feel it's fair to tone it down). This is very important so as not to mislead potentially less expert readers. We encourage you to provide a more complete discussion of this issue in the paper. We feel it would be very constructive for the field.

Second, we agree with both experts that the mechanistic data could be more developed. Again however we are not fully convinced tracing studies alone would be sufficiently informative in this case. We do agree that direct evidence is lacking in the paper that DHODH activity itself changes (we believe this is inferred from other clever perturbation and inhibitor studies). In the absence of such direct evidence, we feel it is best to reword or tone down the title "Exogenous NAD⁺ impairs DHODH activity" and related discussion.

Overall, as two independent experts agree that the tracing studies would be valuable to address Rev#1's point #2, we encourage you to add them. However, if this is not possible due to regulations in Denmark, we are still willing to consider the study given the overall support from the panel.

Therefore, we'll be happy in principle to publish the manuscript in Nature Cell Biology, pending minor revisions to satisfy the referees' final requests [as described above] and to comply with our editorial and formatting guidelines.

We will now begin performing detailed checks on your paper and will send you a checklist detailing our editorial and formatting requirements in about 1-2 weeks. Please do not upload the final materials and make any revisions to the text/figures until you receive this

additional information from us.

Thank you again for your interest in Nature Cell Biology and for your patience. Please do not hesitate to contact me if you have any questions.

Sincerely,

Melina

Melina Casadio, PhD
Senior Editor, Nature Cell Biology
ORCID ID: <https://orcid.org/0000-0003-2389-2243>

Reviewer #1 (Remarks to the Author):

The new experimental data have strengthened the manuscript. However, while some of my initial concerns have been addressed, my enthusiasm is limited because of the unresolved concerns that are noted below.

- 1) The lack of in vivo evidence for the observed phenotypes. Does prolonged exposure to NAD results in pyrimidine depletion and, subsequently, replication stress in vivo? This is key to establishing the broader physiological relevance of the presented findings.
- 2) Failure to address the mechanism by which exogenous NAD depletes pyrimidine levels while increasing purine levels. Tracing experiments for pyrimidine and purine synthesis would clarify the author's proposed model.
- 3) The cell permeability of NAD remains undetermined. While the additional references provided by the authors are helpful and informative, I remain unpersuaded that NAD is effectively entering the cells. ¹³C-NAD tracing experiments could be performed to directly monitor the fractional abundance of ¹³C-NAD with respect to total NAD levels in the cytosol versus mitochondria.

Reviewer #2 (Remarks to the Author):

The authors have provided very thorough responses to my comments. I am satisfied by their responses.

Reviewer #3 (Remarks to the Author):

The authors took a big effort with a 6-month of work to address the comments raised by this reviewer and the other 3 reviewers. Key new data were: CD38 did not involved in NAD⁺-dependent DNA repair and NAD⁺ changed metabolisms of nucleotides. Especially, the authors have addressed the major comments raised by this reviewer and the quality of the paper is now much improved. There is no comments from this reviewer.

ADDITIONAL COMMENTS FROM REV#3 ON REV#1'S COMMENTS:

Rev#1 point #1: The lack of in vivo evidence for the observed phenotypes. Does prolonged exposure to NAD results in pyrimidine depletion and, subsequently, replication

stress in vivo? This is key to establishing the broader physiological relevance of the presented findings.

Rev#3: I agree that in vivo animal experiments are not absolutely needed in this paper, although with the animal data will reinforce the findings.

Rev#1 point #2: Failure to address the mechanism by which exogenous NAD depletes pyrimidine levels while increasing purine levels. Tracing experiments for pyrimidine and purine synthesis would clarify the author's proposed model.

Rev#3: The authors summarized a lot of data to response to this question, but none of them were direct and key pieces of evidence. It is likely acceptable that exogenous NAD⁺ induces high MMP which reducing DHODH activity, and finally could result in low pyrimidine; studies on why NAD⁺ reduces DHODH at molecular level in this paper is obscure as the target journal is a cell biology journal. Furthermore, the mechanism on why exogenous NAD⁺ increases purine is still not well answered.

Rev#1 point #3: The cell permeability of NAD remains undetermined. While the additional references provided by the authors are helpful and informative, I remain unpersuaded that NAD is effectively entering the cells. ¹³C-NAD tracing experiments could be performed to directly monitor the fractional abundance of ¹³C-NAD with respect to total NAD levels in the cytosol versus mitochondria.

Rev#3: It is clear that cytoplasmic NAD⁺ can transport to mitochondria via SLC25A51 (Baur lab, Nature). However, it is true that to my knowledge, most of the NAD⁺ researchers I know, including myself, we do not believe NAD⁺ can transport to the cells (it may enter into the cells in 1-2 specific cell types published by some Italian group, with data pending validation). While NR can be transported into the cells via ENT1, ENT2, ENT4 transporters, there is big debate on whether NMN (for structure = NR + P) can be imported into the cells through the Slc12a8 (PMID: 32694647; PMID: 32694650). Since NAD⁺ is much bigger than NMN, it is very unlikely it can be transported into the cells; however, this does not negatively affect the cellular data the authors presented, as extracellular NAD⁺ can be quickly and easily degraded into NR, NMN (can be further cut to NR and NAM), and NAM, which can enter the cells to synthesize more cytoplasmic NAD⁺.

Many of the sentences in the rebuttal are overstated: 'Overall to this point, we are entirely convinced (as were apparently also all the other reviewers who evaluated our manuscript) that together with thorough published evidence to this point, our own data document beyond any reasonable doubt, that mammalian cells are permeable for NAD⁺ ...' Given the fact that this knowledge is well established'.

In the imaging figures the authors provided, it may also suggest NAD⁺ can not transport to the cells directly, as after 1 h treatment with 2-Ethynyl-Adenosine-NAD⁺, there was very weak signal in the cells (compared to the signals after 24 h). The strong signal in 24 h could be due to degraded 2-Ethynyl-Adenosine-NAD⁺ (could be NR, NMN, or NAM) to enter the cells and re-synthesized. If NAD⁺ can enter the cells, 1 h is enough!!!

In conclusion, for point 2 and point 3, it would be ideal to perform ¹³C-NAD tracing experiments: other large amount experiments that authors have been done can not replace

the tracing experiments, but can definitely as a complimentary set of data. We understand the argument 'In general, Denmark (and other countries) have substantially phased down the use of radioactive isotopes such as the experiments recommended by the referee. Instead, fluorescence tracing is preferred, also due to the serious environmental and health concerns about radioactivity' and support environmental protection, but such experiments can be safely done in a collaborative lab or via a company which has strict ways on waste collection and treatment.

As this is the top journal of cell mechanism and biology, the authors should have expected the performance of additional necessary experiments. However, I would still be interested in the manuscript if the authors instead toned down the claim of direct entry of NAD+.

Reviewer #4 (Remarks to the Author):

Taken together, the authors of this manuscript were able to take the reviewers' comments and greatly improve the paper and strengthen their findings. I agree the letter format in NCB will inspire more clinical trials to investigate their concept being proposed here. I am happy with the rebuttal, experiments performed, explanations given, and the way the work is now presented. This report will have a strong impact on the field.

Decision Letter, author guidance:

Message: Our ref: NCB-LE50081B

20th September 2023

Dear Dr. Bartek,

Thank you for your patience as we've prepared the guidelines for final submission of your Nature Cell Biology manuscript, "NAD+ regulates nucleotide metabolism and genomic DNA replication" (NCB-LE50081B). Please carefully follow the step-by-step instructions provided in the attached file, and add a response in each row of the table to indicate the changes that you have made. Please also check and comment on any additional marked-up edits we have proposed within the text. Ensuring that each point is addressed will help to ensure that your revised manuscript can be swiftly handed over to our production team.

In recognition of the time and expertise our reviewers provide to Nature Cell Biology's editorial process, we would like to formally acknowledge their contribution to the external peer review of your manuscript entitled "NAD+ regulates nucleotide metabolism and genomic DNA replication". For those reviewers who give their assent, we will be publishing their names alongside the published article.

Nature Cell Biology offers a Transparent Peer Review option for new original research manuscripts submitted after December 1st, 2019. As part of this initiative, we encourage our authors to support increased transparency into the peer review process by agreeing to have the reviewer comments, author rebuttal letters, and editorial decision letters published as a Supplementary item. When you submit your final files please clearly state in your cover letter whether or not you would like to participate in this initiative. Please note that failure to state your preference will result in delays in accepting your manuscript for publication.

Cover suggestions

COVER ARTWORK: We welcome submissions of artwork for consideration for our cover. For more information, please see our guide for cover artwork.

Nature Cell Biology has now transitioned to a unified Rights Collection system which will allow our Author Services team to quickly and easily collect the rights and permissions required to publish your work. Approximately 10 days after your paper is formally accepted, you will receive an email in providing you with a link to complete the grant of rights. If your paper is eligible for Open Access, our Author Services team will also be in touch regarding any additional information that may be required to arrange payment for your article.

Please note that *Nature Cell Biology* is a Transformative Journal (TJ). Authors may publish their research with us through the traditional subscription access route or make their paper immediately open access through payment of an article-processing charge (APC). Authors will not be required to make a final decision about access to their article until it has been accepted. Find out more about Transformative Journals

Please use the following link for uploading these materials:
[redacted]

Best regards,

Kendra Donahue
Staff
Nature Cell Biology

On behalf of

Melina Casadio, PhD
Senior Editor, Nature Cell Biology
ORCID ID: <https://orcid.org/0000-0003-2389-2243>

Reviewer #1:

Remarks to the Author:

The new experimental data have strengthened the manuscript. However, while some of my initial concerns have been addressed, my enthusiasm is limited because of the unresolved concerns that are noted below.

- 1) The lack of in vivo evidence for the observed phenotypes. Does prolonged exposure to NAD results in pyrimidine depletion and, subsequently, replication stress in vivo? This is key to establishing the broader physiological relevance of the presented findings.
- 2) Failure to address the mechanism by which exogenous NAD depletes pyrimidine levels while increasing purine levels. Tracing experiments for pyrimidine and purine synthesis would clarify the author's proposed model.
- 3) The cell permeability of NAD remains undetermined. While the additional references provided by the authors are helpful and informative, I remain unpersuaded that NAD is effectively entering the cells. ¹³C-NAD tracing experiments could be performed to directly monitor the fractional abundance of ¹³C-NAD with respect to total NAD levels in the cytosol versus mitochondria.

Reviewer #2:

Remarks to the Author:

The authors have provided very thorough responses to my comments. I am satisfied by their responses.

Reviewer #3:

Remarks to the Author:

The authors took a big effort with a 6-month of work to address the comments raised by this reviewer and the other 3 reviewers. Key ndw data were: CD38 did not involved in NAD⁺-dependent DNA repair and NAD⁺ changed metabolisms of nucleotides. Especially, the authors have addressed the major comments raised by this reviewer and the quality of the paper is now much improved. There is no comments from this reviewer.

ADDITIONAL COMMENTS FROM REV#3 ON REV#1'S COMMENTS:

Rev#1 point #1: The lack of in vivo evidence for the observed phenotypes. Does prolonged exposure to NAD results in pyrimidine depletion and, subsequently, replication stress in vivo? This is key to establishing the broader physiological relevance of the presented findings.

Rev#3: I agree that in vivo animal experiments are not absolutely needed in this paper, although with the animal data will reinforce the findings.

Rev#1 point #2: Failure to address the mechanism by which exogenous NAD depletes pyrimidine levels while increasing purine levels. Tracing experiments for pyrimidine and purine synthesis would clarify the author's proposed model.

Rev#3: The authors summarized a lot of data to response to this question, but none of them were direct and key pieces of evidence. It is likely acceptable that exogenous NAD⁺ induces high MMP which reducing DHODH activity, and finally could result in low pyrimidine; studies on why NAD⁺ reduces DHODH at molecular level in this paper is obscure as the target journal is a cell biology journal. Furthermore, the mechanism on why exogenous NAD⁺ increases purine is still not well answered.

Rev#1 point #3: The cell permeability of NAD remains undetermined. While the additional references provided by the authors are helpful and informative, I remain unpersuaded that NAD is effectively entering the cells. 13C-NAD tracing experiments could be performed to directly monitor the fractional abundance of 13C-NAD with respect to total NAD levels in the cytosol versus mitochondria.

Rev#3: It is clear that cytoplasmic NAD⁺ can transport to mitochondria via SLC25A51 (Baur lab, Nature). However, it is true that to my knowledge, most of the NAD⁺ researchers I know, including myself, we do not believe NAD⁺ can transport to the cells (it may enter into the cells in 1-2 specific cell types published by some Italian group, with data pending validation). While NR can be transported into the cells via ENT1, ENT2, ENT4 transporters, there is big debate on whether NMN (for structure = NR + P) can be imported into the cells through the Slc12a8 (PMID: 32694647; PMID: 32694650). Since NAD⁺ is much bigger than NMN, it is very unlikely it can be transported into the cells; however, this does not negatively affect the cellular data the authors presented, as extracellular NAD⁺ can be quickly and easily degraded into NR, NMN (can be further cut to NR and NAM), and NAM, which can enter the cells to synthesize more cytoplasmic NAD⁺.

Many of the sentences in the rebuttal are overstated: 'Overall to this point, we are entirely convinced (as were apparently also all the other reviewers who evaluated our manuscript) that together with thorough published evidence to this point, our own data document

beyond any reasonable doubt, that mammalian cells are permeable for NAD' ...' Given the fact that this knowledge is well established'.

In the imaging figures the authors provided, it may also suggest NAD⁺ can not transport to the cells directly, as after 1 h treatment with 2-Ethynyl-Adenosine-NAD⁺, there was very weak signal in the cells (compared to the signals after 24 h). The strong signal in 24 h could be due to degraded 2-Ethynyl-Adenosine-NAD⁺ (could be NR, NMN, or NAM) to enter the cells and re-synthesized. If NAD⁺ can enter the cells, 1 h is enough!!!

In conclusion, for point 2 and point 3, it would be ideal to perform ¹³C-NAD tracing experiments: other large amount experiments that authors have been done can not replace the tracing experiments, but can definitely as a complimentary set of data. We understand the argument 'In general, Denmark (and other countries) have substantially phased down the use of radioactive isotopes such as the experiments recommended by the referee. Instead, fluorescence tracing is preferred, also due to the serious environmental and health concerns about radioactivity' and support environmental protection, but such experiments can be safely done in a collaborative lab or via a company which has strict ways on waste collection and treatment.

As this is the top journal of cell mechanism and biology, the authors should have expected the performance of additional necessary experiments. However, I would still be interested in the manuscript if the authors instead toned down the claim of direct entry of NAD⁺.

Reviewer #4:

Remarks to the Author:

Taken together, the authors of this manuscript were able to take the reviewers' comments and greatly improve the paper and strengthen their findings. I agree the letter format in NCB will inspire more clinical trials to investigate their concept being proposed here. I am happy with the rebuttal, experiments performed, explanations given, and the way the work is now presented. This report will have a strong impact on the field.

Author Rebuttal, second revision:

Responses to the remaining comments of the reviewers

- 1) Our responses are in **bold**
- 2) Comments from the individual reviewers are in standard letters and those raised by Reviewer nr. 1 are now shown in italics prior to the comments of Reviewer 3 on those of Reviewer 1 and on our responses to them.

Reviewer #1:

We have answered the 3 remaining concerns raised by referee #1 in our previous rebuttal.

Please see also our answers to referee #3 for the remaining concerns.

Reviewer #2:

Remarks to the Author:

The authors have provided very thorough responses to my comments. I am satisfied by their responses.

Many thanks to the reviewer #2 for supporting our manuscript.

Reviewer #3:

Remarks to the Author:

The authors took a big effort with a 6-month of work to address the comments raised by this reviewer and the other 3 reviewers. Key ndw data were: CD38 did not involved in NAD⁺-dependent DNA repair and NAD⁺ changed metabolisms of nucleotides. Especially, the authors have addressed the major comments raised by this reviewer and the quality of the paper is now much improved. There is no comments from this reviewer.

Our sincere thanks to referee #3 for recognizing our efforts and supporting our manuscript.

ADDITIONAL COMMENTS FROM REV#3 ON REV#1'S COMMENTS:

Ad:

Rev#1 point #1: The lack of in vivo evidence for the observed phenotypes. Does prolonged exposure to NAD results in pyrimidine depletion and, subsequently, replication stress in vivo? This is key to establishing the broader physiological relevance of the presented findings.

Rev#3: I agree that in vivo animal experiments are not absolutely needed in this paper, although with the animal data will reinforce the findings.

We agree and, as we previously explained, we intend to pursue this important line of in vivo investigation, mainly by initiating clinical trial(s) of cancer treatment. This effort might be somewhat facilitated in terms of ethical concerns, given that NAD⁺ is readily available as a food supplement and an injectable solution approved for human use.

Ad:

Rev#1 point #2: Failure to address the mechanism by which exogenous NAD depletes pyrimidine levels while increasing purine levels. Tracing experiments for pyrimidine and purine synthesis would clarify the author's proposed model.

Rev#3: The authors summarized a lot of data to response to this question, but none of them were direct and key pieces of evidence. It is likely acceptable that exogenous NAD⁺ induces high MMP which reducing DHODH activity, and finally could result in low pyrimidine; studies on why NAD⁺ reduces DHODH at molecular level in this paper is obscure as the target journal is a cell biology journal. Furthermore, the mechanism on why exogenous NAD⁺ increases purine is still not well answered.

First, there seems to be a slight misunderstanding on this point, in case the reviewer thought that we wished to imply a direct inhibitory impact of NAD⁺ on DHODH. Indeed, and consistent with what the reviewer regards as an acceptable indirect explanation for the decreased DHODH activity, we show (and concisely state on page 13, lines: 296-298) that exogenous NAD⁺ modulates the activity of the electron transport chain (ETC) and because of this the activity of DHODH was decreased. The evidence for this mechanism, i.e. acting through the primary impact of exogenous NAD⁺ on the mitochondrial Electron Transport Chain, is presented in figures 4-6, and further supported by the extensive metabolome analyses that we document in the re-revised manuscript.

Furthermore, we have now modified the relevant text related to AMP and UMPS, and their relevance for DHODH activity and impact on pyrimidine levels, to better present our findings and clarify the overall context, as follows:

previously, we wrote:

Interestingly, the accumulation of AMP has previously been reported to inhibit the growth of 3T6 mouse fibroblasts, presumably by inducing pyrimidine starvation, and AMP as well as IMP, a key intermediate in purine biosynthesis, can impair the activity of uridine monophosphate synthase (UMPS) in the pyrimidine biosynthesis pathway (lines 205-208).

In the current re-revised version we concisely state:

AMP can inhibit the UMPS enzymatic step (Ishii, et al. 1973. JCS 13:429-439), which occurs downstream of the mitochondrial conversion of dihydroorotate to orotate by DHODH in pyrimidine biosynthesis (see lines 268-269 in the revised version of our manuscript).

Overall on this point, we hope the textual changes have helped to better convey the message that while the activity of DHODH enzyme is key in the observed effects, this reflects the impact on the primary target of the NAD⁺, namely the Electron Transport Chain upstream of DHODH. We also hope that our comprehensive metabolome analysis, conducted across various cell lines and under diverse conditions, provides a sufficient level of detail to support our conclusions.

Ad:

Rev#1 point #3: The cell permeability of NAD remains undetermined. While the additional references provided by the authors are helpful and informative, I remain unpersuaded that NAD is effectively

entering the cells. 13C-NAD tracing experiments could be performed to directly monitor the fractional abundance of 13C-NAD with respect to total NAD levels in the cytosol versus mitochondria.

Rev#3: It is clear that cytoplasmic NAD⁺ can transport to mitochondria via SLC25A51 (Baur lab, Nature). However, it is true that to my knowledge, most of the NAD⁺ researchers I know, including myself, we do not believe NAD⁺ can transport to the cells (it may enter into the cells in 1-2 specific cell types published by some Italian group, with data pending validation). While NR can be transported into the cells via ENT1, ENT2, ENT4 transporters, there is big debate on whether NMN (for structure = NR + P) can be imported into the cells through the Slc12a8 (PMID: 32694647; PMID: 32694650). Since NAD⁺ is much bigger than NMN, it is very unlikely it can be transported into the cells; however, this does not negatively affect the cellular data the authors presented, as extracellular NAD⁺ can be quickly and easily degraded into NR, NMN (can be further cut to NR and NAM), and NAM, which can enter the cells to synthesize more cytoplasmic NAD⁺.

Here, reviewer 1 and reviewer 3, express their belief that exogenous NAD⁺ may not be able to enter the cells, despite some evidence on the contrary, including some of the data we present in this manuscript. The reviewer instead suggests an alternative interpretation, namely that exogenous NAD⁺ may first be degraded outside the cell, then the precursors generated through such degradation may enter the cell, where they are then used to synthesize NAD⁺ again, which then causes the biochemical and biological effects that we observe, including the newly identified impact on nucleotide metabolism and DNA replication that we highlight as our main findings.

First, we wish to say again that we are aware of this ongoing debate in the field on whether or not exogenous NAD⁺ can enter cells, and we pointed this out in both the Introduction and Discussion of our manuscript.

Second, we have directly tested the suggested alternative explanation via NAD⁺ degradation and the effects being caused by the precursors, however in the light of the data we obtained, this alternative interpretation seems unlikely. This statement is based on two types of experiments that we performed and presented in the revised manuscript:

- i) In parallel comparative settings, we treated cells with NAD⁺ and side by side also with the relevant degradation products/precursors NR and NMN, mentioned by the reviewer at this point. Notably, the precursors were unable to recapitulate the effects of exogenous intact NAD⁺ (see Figure 2f, g, top of page 7).**
- ii) Furthermore, in a complementary experiment with NAD⁺, we inhibited the activity of the extracellular enzyme CD38 that metabolizes the NAD⁺, however, the CD38 inhibition showed no detectable effect on the genomic DNA synthesis (see Extended Data Figure 2n-o, top of page 7)**

In summary, while we cannot completely exclude the possibility that the degradation products/precursors of NAD⁺ might contribute to subsequent intracellular synthesis of NAD⁺, our experimental data do not support this as a mechanism through which exogenous NAD⁺ impacts DNA synthesis and nucleotide levels leading to phenotypes that we observe. Nevertheless, we followed the recommendation of the reviewer no. 3 and toned down our interpretation in this regard (see page 16, lines 353-355, and our response to the rest of this comment below)

In this context, various tracing experiments, demonstrating conclusively that NAD⁺ can enter the cell as an intact molecule, have been conducted by numerous research groups, both within and outside of Italy. We have cited this body of evidence in our manuscript, with perhaps the most compelling support coming from the research published in "Proceedings of the National Academy of Sciences of the United States of America" (PNAS) in 1987 (Vol. 84, pp. 1286-1289). Please refer to the provided snapshot from the mentioned manuscript for further details. Additionally, as mentioned above, since the effects of NAD⁺ treatment are not phenocopied by its precursors NR and NMN in our study, this further supports our original interpretation that the effects are elicited by NAD⁺ directly and not following the conversion of its degradation products/metabolites.

1288 Cell Biology: Loetscher *et al.*

Proc. Natl. Acad. Sci. USA 84 (1987)

Table 2. Evidence for membrane transfer of intact NAD⁺, radiolabeled in two different positions, in cultured hepatocytes concomitant with the increase of intracellular NAD⁺

Incubation time, hr	Medium NAD ⁺ , μM	[adenine-2,8- ³ H]NAD ⁺ /[carbonyl- ¹⁴ C]NAD ⁺ isotope ratio		
		Extracellular	Intracellular	Extracellular/intracellular
1	100	0.71	0.82	0.87
1	500	0.68	0.69	0.98
2	100	0.68	0.80	0.85

Hepatocytes were incubated with double-labeled NAD⁺ ([adenine-2,8-³H]NAD⁺; specific activity, 5278 cpm/nmol. [carbonyl-¹⁴C]NAD⁺; specific activity, 7389 cpm/nmol). Following purifica-

Table 3. Consequences of the elevation of intracellular NAD⁺ concentrations on the level of poly(ADP-ribose) modification of hepatocellular chromatin and the ratio between ADP-ribosyl residues contained in NAD⁺ and poly(ADP-ribose)

	Intracellular NAD ⁺ , μM	Poly(ADP-ribose), nM	Poly(ADP-ribose)/NAD ⁺
Control cells	458.0 ± 34.6	66.0 ± 2.8	1.4 × 10 ⁻⁴
NAD ⁺ -loaded cells	786.0 ± 57.1	340.0 ± 60.8	4.3 × 10 ⁻⁴

The data represent the averaged values from two independent experiments involving separate cell preparations.

phosphoribosyl-AMP residues [i.e., internal (ADP-ribose)_n-

Many of the sentences in the rebuttal are overstated: ‘Overall to this point, we are entirely convinced (as were apparently also all the other reviewers who evaluated our manuscript) that together with thorough published evidence to this point, our own data document beyond any reasonable doubt, that mammalian cells are permeable for NAD’ ...’ Given the fact that this knowledge is well established’.

We apologize for any potential overstatements in our responses to the referees' comments. We have chosen to maintain a transparent review process, making all correspondence available to the readers

who can therefore judge to what extent are our responses and conclusions justified, based on the available experimental evidence.

In the imaging figures the authors provided, it may also suggest NAD⁺ can not transport to the cells directly, as after 1 h treatment with 2-Ethynyl-Adenosine-NAD⁺, there was very weak signal in the cells (compared to the signals after 24 h). The strong signal in 24 h could be due to degraded 2-Ethynyl-Adenosine-NAD⁺ (could be NR, NMN, or NAM) to enter the cells and re-synthesized. If NAD⁺ can enter the cells, 1 h is enough!!!

We believe it is only natural that we see increasing accumulation of the click-NAD⁺ signal over time, with lower signal at 1 hour and stronger signal after 24 hours of treatment. The reviewer suggests the kinetics might be due to the degraded precursors entering and forming new NAD⁺. Again, while the degradation/precursor route might take place over time, the main phenotypes that we saw cannot be induced by such precursors (see above in our response to this point). Here, we performed the click-NAD⁺ experiments as an alternative to the use of radioactivity and to confirm what had been published before. In the previous rebuttal, we specified that we used click-NAD⁺ at a concentration of 10 μ M, which at this low concentration could influence how quickly a sufficient detectable amount of NAD⁺ is taken up by the cell. Unfortunately, Click-NAD⁺ can be only purchased as a 10 mM solution in 20 μ l volume. Even using a high-numerical-aperture 63X magnification microscope objective, the noise-to-signal ratio allows us to see only a weak signal after 1 hr, and a strong signal of NAD⁺ accumulation inside the cell after 24h.

We agree with the fact that radioactivity methods are valuable, indeed, Loetscher, P. et. al. demonstrated the accumulation of exogenous intact NAD⁺ inside the cell after 1h, using such a radioisotope-based approach.

Loetscher, P., Alvarez-Gonzalez, R. & Althaus, F. R. Poly(ADP-ribose) may signal changing metabolic conditions to the chromatin of mammalian cells. Proc Natl Acad Sci U S A 84, 1286-1289, doi:10.1073/pnas.84.5.1286 (1987).

In conclusion, for point 2 and point 3, it would be ideal to perform ¹³C-NAD tracing experiments: other large amount experiments that authors have been done can not replace the tracing experiments, but can definitely as a complimentary set of data. We understand the argument 'In general, Denmark (and other countries) have substantially phased down the use of radioactive isotopes such as the experiments recommended by the referee. Instead, fluorescence tracing is preferred, also due to the serious environmental and health concerns about radioactivity' and support environmental protection, but such experiments can be safely done in a collaborative lab or via a company which has strict ways on waste collection and treatment.

As this is the top journal of cell mechanism and biology, the authors should have expected the performance of additional necessary experiments. However, I would still be interested in the manuscript if the authors instead toned down the claim of direct entry of NAD⁺.

As our work safety officers strongly discouraged us from using radioisotopes when we approached them after receiving this comment, we opted for the reviewer's 3 suggested solution to this issue and in the re-revised latest version of our manuscript, we toned down the relevant claim about NAD⁺ entering cells as an intact molecule:

lines 353-355 - At least a fraction of the exogenous intact NAD⁺ may reach mitochondrial cristae...-

As reviewer no 3 regards this solution as acceptable, we hope this textual modification may solve this issue.

Reviewer #4:

Remarks to the Author:

Taken together, the authors of this manuscript were able to take the reviewers' comments and greatly improve the paper and strengthen their findings. I agree the letter format in NCB will inspire more clinical trials to investigate their concept being proposed here. I am happy with the rebuttal, experiments performed, explanations given, and the way the work is now presented. This report will have a strong impact on the field.

Our sincere thanks to the referee #4 for her/his positive comments.

Final Decision Letter:

Message: Dear Dr Bartek,

I am pleased to inform you that your manuscript, "NAD+ regulates nucleotide metabolism and genomic DNA replication", has now been accepted for publication in Nature Cell Biology.

Please note that *Nature Cell Biology* is a Transformative Journal (TJ). Authors may publish their research with us through the traditional subscription access route or make their paper immediately open access through payment of an article-processing charge (APC). Authors will not be required to make a final decision about access to their article until it has been accepted. Find out more about Transformative Journals

Authors may need to take specific actions to achieve compliance with funder and

institutional open access mandates. If your research is supported by a funder that requires immediate open access (e.g. according to Plan S principles) then you should select the gold OA route, and we will direct you to the compliant route where possible. For authors selecting the subscription publication route, the journal's standard licensing terms will need to be accepted, including self-archiving policies. Those licensing terms will supersede any other terms that the author or any third party may assert apply to any version of the manuscript.

If you have not already done so, we strongly recommend that you upload the step-by-step protocols used in this manuscript to the Protocol Exchange (www.nature.com/protocolexchange), an open online resource established by Nature Protocols that allows researchers to share their detailed experimental know-how. All uploaded protocols are made freely available, assigned DOIs for ease of citation and are fully searchable through nature.com. Protocols and Nature Portfolio journal papers in which they are used can be linked to one another, and this link is clearly and prominently visible in the online versions of both papers. Authors who performed the specific experiments can act as primary authors for the Protocol as they will be best placed to share the methodology details, but the Corresponding Author of the present research paper should be included as one of the authors. By uploading your Protocols to Protocol Exchange, you are enabling researchers to more readily reproduce or adapt the methodology you use, as well as increasing the visibility of your protocols and papers. You can also establish a dedicated page to collect your lab Protocols. Further information can be found at www.nature.com/protocolexchange/about

With kind regards,

Melina Casadio, PhD
Senior Editor, Nature Cell Biology
ORCID ID: <https://orcid.org/0000-0003-2389-2243>